# Ferritinophagy mediates adaptive resistance to EGFR tyrosine kinase inhibitors in non-small cell lung cancer

Hui Wang[1,2,3,8], Qianfan Hu[1,2,3,8], Yuzhong Chen[2,4,8], Xing Huang[5,8], Yipeng Feng[2,3], Yuanjian Shi[2,3], Rutao Li[6], Xuewen Yin[2], Xuming Song[2,3], Yingkuan Liang[1], Te Zhang[2,3], Lin Xu[1,2,3,7], Gaochao Dong [2,3] ✉ & Feng Jiang [1,2,3] ✉

Osimertinib (Osi) is a widely used epidermal growth factor receptor tyrosine kinase inhibitor (EGFR-TKI). However, the emergence of resistance is inevitable, partly due to the gradual evolution of adaptive resistant cells during initial treatment. Here, we find that Osi treatment rapidly triggers adaptive resistance in tumor cells. Metabolomics analysis reveals a significant enhancement of oxidative phosphorylation (OXPHOS) in Osi adaptive-resistant cells. Mechanically, Osi treatment induces an elevation of NCOA4, a key protein of ferritinophagy, which maintains the synthesis of iron-sulfur cluster (ISC) proteins of electron transport chain and OXPHOS. Additionally, active ISC protein synthesis in adaptive-resistant cells significantly increases the sensitivity to copper ions. Combining Osi with elesclomol, a copper ion ionophore, significantly increases the efficacy of Osi, with no additional toxicity. Altogether, this study reveals the mechanisms of NCOA4-mediated ferritinophagy in Osi adaptive resistance and introduces a promising new therapy of combining copper ionophores to improve its initial efficacy.

Epidermal growth factor receptor tyrosine kinase inhibitors (EGFR-TKIs) have become the first-line treatment for patients with advanced EGFR mutation in non-small cell lung cancer (NSCLC)[1–3]. However, the clinical efficacy of EGFR-TKIs is limited by the emergence of drug resistance[4,5]. Studies have shown that the adaptive resistance of tumor cells to drugs at the initial treatment[6,7], the drug-tolerant persister (DTP)[8–10], and the genomic instability of tumor cells under the long-term drug stress are the bridges mediating the emergence of irreversible drug resistance[11,12]. The characteristics of DTP cells include their

ability to evade cell death during initial chemotherapy or EGFR-TKIs-based combination therapy, which represents a subset of cancer cells. These cells serve as a reservoir from which drug-resistant tumors emerge[9,13,14]. Although the adaptive resistance of tumor cells to drugs is a crucial step in the acquisition of drug resistance, little is known about how this process relates to EGFR-TKIs.

Metabolic reprogramming is a hallmark of cancer cells, required for tumor growth and survival in different environments[15,16]. Studies have demonstrated that metabolic reprogramming occurs in response

[1]Department of Thoracic Surgery, Affiliated Cancer Hospital of Nanjing Medical University and Jiangsu Cancer Hospital and Jiangsu Institute of Cancer Research, Xuanwu District Nanjing, China. [2]Jiangsu Key Laboratory of Molecular and Translational Cancer Research, Xuanwu District Nanjing, China. [3]The Fourth Clinical College of Nanjing Medical University, Nanjing, PR China. [4]Department of Oncology, Affiliated Cancer Hospital of Nanjing Medical University and Jiangsu Cancer Hospital and Jiangsu Institute of Cancer Research, Xuanwu District Nanjing, China. [5]Department of Pathology, Affiliated Cancer Hospital of Nanjing Medical University and Jiangsu Cancer Hospital and Jiangsu Institute of Cancer Research, Xuanwu District Nanjing, China. [6]Department of Thoracic Surgery, The First Affiliated Hospital of Soochow University, Suzhou, China. [7]Collaborative Innovation Center for Cancer Personalized Medicine, Nanjing Medical University, Jiangning District Nanjing, China. [8]These authors contributed equally: Hui Wang, Qianfan Hu, Yuzhong Chen, Xing Huang. ✉e-mail: gaochao_dong@njmu.edu.cn; fengjiang_nj@njmu.edu.cn

to drug-induced stress, with energy production metabolism being the most affected[17,18]. Some researchers, such as Otto Warburg, have suggested that mitochondrial respiration defects may be the underlying cause of cancer, since cancer cells often ferment glucose in the presence of oxygen[19,20]. However, despite the preference of tumor cells for aerobic glycolysis, mitochondrial function is often upregulated in cancers, with mitochondrial biogenesis and quality control being commonly observed[21,22]. Furthermore, drug-tolerant tumor cells prefer oxidative phosphorylation (OXPHOS), which takes place in the mitochondria and produces large amounts of ATP[13,23]. Several clinical trials have been carried out to test the effectiveness of inhibiting mitochondrial metabolism as an emerging cancer therapeutic treatment[24-26]. Therefore, it is essential to understand the metabolic reprogramming associated with adaptive resistance to EGFR-TKIs and to establish an integrative framework for identifying and targeting vulnerabilities arising from adaptive resistance and stress mitigation pathways involved in metabolic reprogramming.

In this study, we investigated the active OXPHOS in tumor cells following treatment with Osimertinib (Osi), a third-generation EGFR-TKI. We found that Osi mediates adaptive resistance of tumor cells and that adaptive-resistant tumor cells maintain the synthesis of iron-sulfur clusters (ISC) in mitochondrial respiratory chains through the upregulation of NCOA4 and ferritinophagy. However, the highly active ISC protein synthesis and dependence on OXPHOS make tumor cells more sensitive to copper ions. We tested the effectiveness of combining Osi with copper ionophores both in vitro and in vivo, and observed that the combination therapy significantly inhibited the production of EGFR-TKIs adaptive resistance. This offers a new possibility for combined therapy with Osi.

## Results

### Osi treatment rapidly induces adaptative resistance in tumor cells

To investigate the response of EGFR-mutated NSCLC cells to Osi therapy in vivo, we utilized an immunodeficient mouse model (BALB/c nude mice), where human EGFR-mutated tumor cells were in situ implanted into the left lung. The NCI-H1975 (H1975, EGFR Exon_21_L858R) or HCC827 (EGFR Exon_19_DEL) cells were labeled with luciferase to monitor growth by bioluminescence imaging (BLI) after confirming successful tumor vaccination five days after injection. The mice were randomly divided into two groups and received either vehicle (DMSO plus saline) or Osi therapy, and the tumor growth was measured every five days by BLI (Fig. 1a). We observed that short-term treatment with Osi (within one week after treatment) significantly inhibited tumor cell growth, but after ten days of treatment, tumor cells rapidly gained proliferative capacity (Fig. 1b, c). Statistical analysis of tumor doubling time demonstrated that Osi only significantly inhibited tumor proliferation at the initial stage, whereas the doubling time of tumor cells gradually shortened with continued treatment until there was almost no difference between the control group (DMSO treatment) (Fig. 1d). We analyzed the volume of lung tumors in mice and the expression level of Ki-67 in tumor cells (a marker of cell proliferative activity) following the completion of the treatment regimen. The results indicated that although Osi treatment resulted in a smaller tumor volume, there was no significant difference in tumor Ki-67 expression between the DMSO treatment group and the Osi treatment group (Fig. 1e, f). Additional mouse models, incorporating sequential Ki-67 expression analysis, further supported the observation that tumor cell proliferation was temporarily suppressed after initial treatment and subsequently experienced a rapid recovery (Fig. 1f-h, Supplementary Fig. 1a, b). This finding supports the idea that tumor cells can overcome proliferation inhibition through adaptive resistance to Osi therapy or through Osi treatment-induced DTP evolution to transition back to the cell cycle and resume growth in vivo.

The p27 is a marker that is highly expressed in cells in the resting state[27]. To further verify the effect of Osi treatment on cell proliferative activity, we used mVenus-p27K labeled tumor cells, which measure cell proliferative activity by the intensity of mVenus-p27K[28,29]. Our results confirmed a significant increase in mVenus-p27K in quiescent tumor cells, which demonstrated that mVenus-p27K exhibits a high sensitivity in accurately assessing the rate of cell proliferation (Supplementary Fig. 1c). The results of three-dimensional (3-D) culture experiments based on Matrigel gel also showed that Osi treatment in the maximum effect (Emax) concentration (1 μM, measured in our previous study, data not shown here) resulted in tumor cell cycle arrest within the initial three days. However, after eight days, the treatment induced adaptive resistance in tumor cells, allowing them to regain their proliferative capacity (Fig. 1i). Additionally, cells cultivated on plates exhibited rapid acquisition of adaptive resistance within 72 h after treatment with Osi. This was demonstrated through the 5-ethynyl-2-deoxyuridine (EdU) assay in four EGFR-mutant NSCLC cell lines, which reflects cellular DNA replication (Fig. 1j, Supplementary Fig. 1d), and cell cycle analysis, which reflects cell division (Fig. 1k, Supplementary Fig. 1e). Cell apoptosis was also significantly reduced at 48-72 h compared to 0-24 h under Osi treatment, indicating that Osi therapy could rapidly induce adaptive resistance of tumor cells in vitro and in vivo (Fig. 1l, m, Supplementary Fig. 1f). However, the downstream of EGFR, including phosphorylated-AKT, phosphorylated-ERK1/2, and phosphorylated-S6 showed that the inhibitory effect of Osi on EGFR was still present, indicating that the generation of adaptive resistance was not caused by the reactivation of the EGFR signaling pathway (Supplementary Fig. 1g-j).

### Adaptive-resistant cells rely on active OXPHOS

There is growing evidence suggesting that metabolic reprogramming could mediate drug tolerance or even the development of acquired drug resistance in tumor cells[17,30]. In this study, we investigated metabolic changes in adaptive-resistant cells by performing metabolomics profiling in H1975 cells treated with Osi for 0 h (parental), 24 h, and 72 h (adapted), respectively (Fig. 2a). Principal component analysis (PCA) of metabolites showed progressive changes and could be distinguished into three groups (Supplementary Fig. 2a, b). Metabolomics analysis indicated significant metabolite alterations in adapted cells compared to parental cells (Supplementary Fig. 2c). The Kyoto Encyclopedia of Genes and Genomes (KEGG) enrichment analysis was conducted to assess the differential metabolites, which indicates the changes in cellular energy metabolism, specifically OXPHOS, in the adapted tumor cells (Fig. 2b). Moreover, there are metabolites in the OXPHOS pathway that are elevated in adapted cells (Fig. 2c, d, Supplementary Fig. 2d). These results suggest a significant alteration of OXPHOS in adaptive-resistant cells.

Cells with high OXPHOS levels were characterized by elevated mitochondrial content. To further characterize the levels of OXPHOS, we analyzed mitochondrial content in parental and adapted tumor cells. As expected, the mitochondrial area per cell surface unit was significantly increased (Fig. 2e, f, Supplementary Fig. 2e). The elevated ratio of mitochondrial DNA to nuclear DNA, along with the increased citrate synthase activity, in adaptive cells provide additional support for the observed enhancements in mitochondrial features and activities (Fig. 2g, h). We also quantified the levels of five electron transport chain (ETC) proteins, including NADH:ubiquinone oxidoreductase core subunit S1 (NDUFS1, complex I component), NADH:Ubiquinone Oxidoreductase Subunit B8 (NDUFB8, complex I component), succinate dehydrogenase complex iron-sulfur subunit B (SDHB, complex II), Ubiquinol-Cytochrome C Reductase Core Protein 2 (UQCRC2, complex III component), Mitochondrially Encoded Cytochrome C Oxidase II (MT-CO2, complex IV component), and ATP Synthase F1 Subunit Alpha (ATP5A, complex V component), which confirmed high-ETC protein levels in

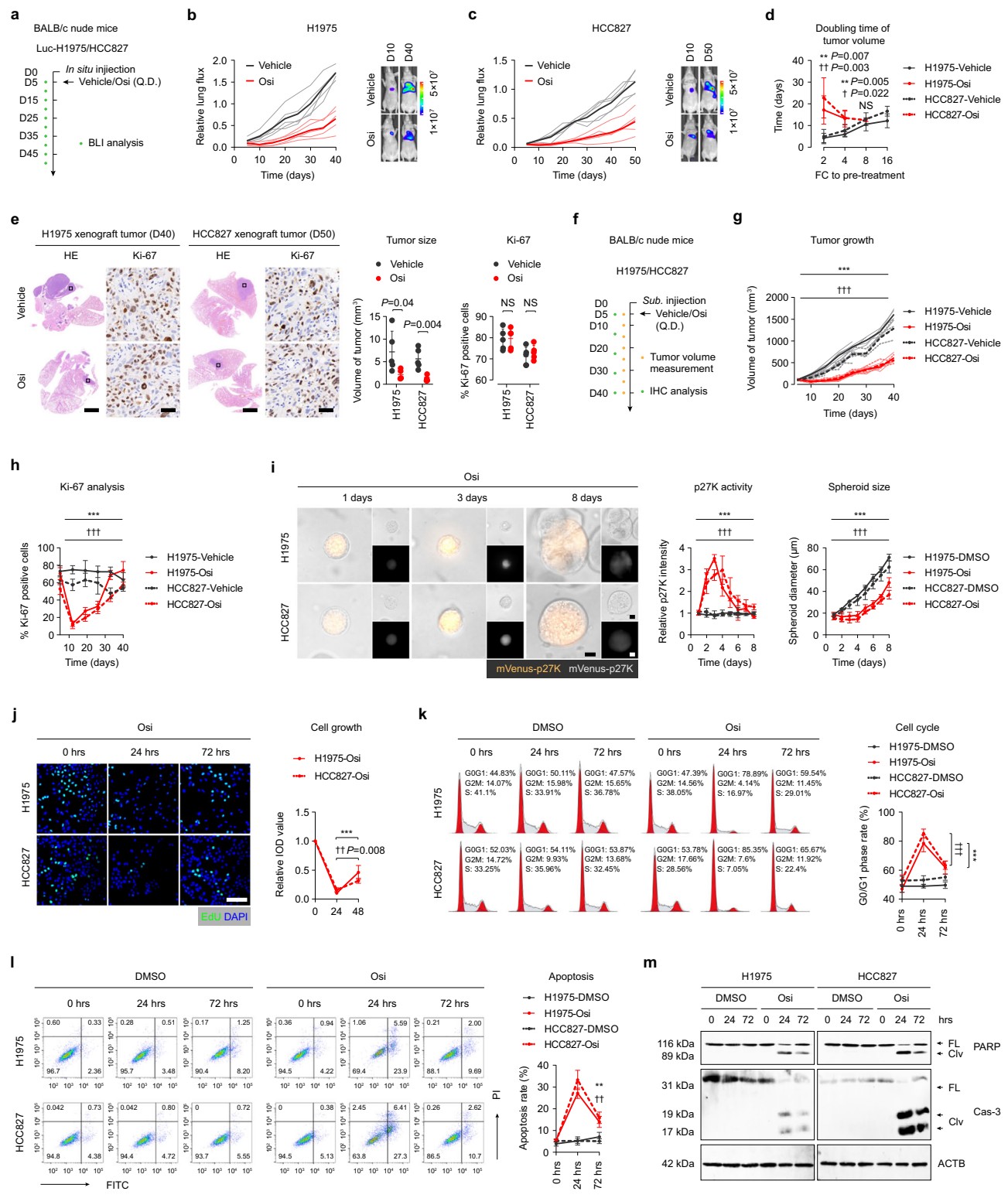

adapted cells (Fig. 2i, Supplementary Fig. 2f). Adapted cells also exhibited a higher oxygen consumption rate (OCR) at basal and maximal-uncoupled states relative to parental cells (Fig. 2j–l). Recent studies have demonstrated that the augmentation of OXPHOS through MAPK inhibition does not occur at the detriment of glycolysis[23]. We also evaluated extracellular acidification rate (ECAR) levels in both parental and adapted cells. The results indicated that the baseline ECAR level did not vary significantly between the parental and adaptive cells (Fig. 2m). However, there was a slight

decrease in the maximal ECAR level observed in the adaptive cells, suggesting a potential shift in the preferred energy metabolism pathway employed by these cells (Fig. 2n). To determine whether OXPHOS mediates the production of adaptive resistance or whether OXPHOS is essential for the survival of adaptive-resistant cells, parental and adapted cells were treated with the ATPase inhibitor Oligomycin A or ETC protein complex I inhibitor Rotenone, respectively. The results showed that inhibition of OXPHOS induced more cell death in the adapted cells than in the parental cells

**Fig. 1 | Osi treatment rapidly induces adaptative resistance in tumor cells.**
**a** BALB/c nude mouse lungs were orthotopically transplanted with Luc-H1975 or Luc-HCC827 cells and were given Vehicle or Osi and follow-up using bioluminescence imaging (BLI) (*n* = 5 mice each group). **b, c** Quantification of the relative fluorescence activity of the region of interest (ROI) in mice by BLI reflects the growth of mice lung tumors (*n* = 5 mice each group). **d** Doubling time of lung tumors reflects the fold-change (FC) to pre-treatment based on the relative strength of the ROI region in mice by BLI (*n* = 5 mice each group). **e** Left, Hematoxylin-eosin (H&E, 1×) and immunohistochemistry (IHC, 40×) analysis of mouse lungs from **a** to **c** on day 40 (H1975 tumors) and day 50 (HCC827 tumors). Scale bar, 1×, 2 mm; 40×, 50 μm. Right, statistics of lung tumor size and percent Ki-67 positive cells were obtained after Vehicle or Osi treatment, respectively (*n* = 5 tumors each group). **f–h** BALB/c nude mouse subcutaneously inoculated with H1975 or HCC827 cells were given Vehicle or Osi treatment. A subset of mice from each group were sacrificed, and their subcutaneous tumors were excised for IHC analysis (*n* = 9 mice each group). **i** H1975 and HCC827 cells treated with Osi in 3-D culture on day 1, day 3, and day 8. Representative experiment (n = 3 technical replicates; 3 independent experiments). Scale bar, 10 μm. **j** H1975 and HCC827 cells were treated with Osi for 0 h, 24 h, or 72 h and incubated with 5-ethynyl-2′-deoxyuridine (EdU) for 2 h. Representative experiment (*n* = 3 technical replicates; 3 independent experiments). Scale bar, 100 μm. **k** Flow cytometry analysis of cell cycles of H1975 and HCC827 cells treated with DMSO or Osi for 0 h, 24 h, or 72 h. Representative experiment (*n* = 3 technical replicates; 3 independent experiments). **l** Flow cytometry analysis of cell apoptosis in H1975 and HCC827 cells treated with DMSO or Osi for 0 h, 24 h, or 72 h. Representative experiment (*n* = 3 technical replicates; 3 independent experiments). **m** Immunoblotting analysis of apoptosis-related proteins PARP and Capase-3 (Cas-3) in H1975 and HCC827 cells treated with DMSO or Osi for 0 h, 24 h or 72 h (Representative images from 3 independent experiments). Data are shown as mean ± SD and were analyzed by a two-tailed unpaired t-test (**d,e**) or a two-way ANOVA **g–l**. NS no significance, \*$p < 0.05$, \*\*$p < 0.01$, \*\*\*$p < 0.001$. †$p < 0.05$, ††$p < 0.01$, †††$p < 0.001$, which was employed to specifically demonstrate the distinction in HCC827 cells. Source data are provided as a Source Data file.

(Fig. 2o, p, Supplementary Fig. 3a-d), indicating that active OXPHOS is crucial for adapted cells to survive.

## NCOA4-mediated ferritinophagy is activated in adaptative-resistant tumor cells

To understand the mechanism by which adaptive-resistant cells develop high levels of OXPHOS, RNA-sequencing was executed on parental and adapted cells (Fig. 3a). Analysis revealed that adapted tumor cells activated a common set of pathways, including cell cycle-related, MAPK, and HIPPO signaling pathways, which is consistent with prior studies[9,14,31] (Fig. 3b). GSEA indicated a significant activation of OXPHOS signaling in the adapted cells, in agreement with our earlier results (Fig. 3c). Additionally, autophagy and iron metabolism signaling pathways were found to be significantly elevated in the adapted cells relative to parental cells (Fig. 3d). Recent study showed that NCOA4-mediated ferritinophagy leads to resistance to chemotherapy or MAPK inhibition in pancreatic cancer cells[32,33]. Analysis of RNA-sequencing data showed that in addition to the overexpression of OXPHOS-associated genes in adapted cells, the levels of NCOA4 were also significantly increased, indicating the possibility of NCOA4-mediated ferritinophagy contribution to the development of Osi adaptive resistance (Fig. 3e). Quantitative PCR analysis of NCOA4 mRNA also verified a gradual increase in NCOA4 levels after Osi treatment (Fig. 3f).

Ferritinophagy refers to the intracellular degradation of ferritin into free ions through autophagy[34,35]. To further determine the extent of ferritinophagy in parental and adapted tumor cells, the levels of the ferritin heavy chain (FTH1) protein were detected, which showed a significant decrease in the levels of ferritin in the adapted cells (Fig. 3g). The number of autophagosomes in bilayer vesicles increased significantly in the adapted tumor cells (Fig. 3h). Autophagic flux was evaluated by employing the fluorescence reporter mCherry-EGFP-LC3B, as well as EGFP-LC3B, in both parental and adapted tumor cells. LC3B is an autophagy-related protein that undergoes post-translational modifications that lead to its lipidation and association with autophagic vesicles[36]. The analysis revealed a considerable upregulation in autophagic flux, as well as a substantial increase in the expression of lipidized LC3B-II, specifically in the adapted tumor cells (Fig. 3i, j, Supplementary Fig. 4a). Additionally, there was a higher degree of co-localization observed between ferritin and LC3 (Fig. 3k), as well as between ferritin and lysosomes in the adapted tumor cells (Fig. 3l), indicating activated ferritinophagy activity in adaptive-resistant tumor cells.

Studies indicate that high levels of free iron ions in cells have been linked to ferroptosis, the accumulation of intracellular lipid reactive oxygen species (ROS) that trigger cell death by depleting intracellular lipid peroxidation and glutathione peroxidase (GPXs)[37–39]. To determine whether activated ferritinophagy in the adapted cells induced ferroptosis, the expression of GPX4 and PTGS2 were examined, the key markers of cell ferroptosis, which showed that the expression levels of GPX4 protein and PTGS2 mRNA did not differ between parental and adaptive cells (Supplementary Fig. 4b, c). Moreover, there was no significant increase observed in the intracellular levels of ROS, lipid peroxidation, or malondialdehyde (MDA) in the adapted cells compared to the parental cells (Supplementary Fig. 3d–f). Furthermore, treatment with the lipid peroxidation inhibitor Ferrostatin-1 (Fer-1)[40] or Trolox (Tro)[41] did not reduce cell death in adapted cells (Supplementary Fig. 3g). Conversely, the combination of the iron chelating agent Deferoxamine (DFO) significantly induced adapted cell death, suggesting that free iron ions did not cause ferroptosis of adapted cells, rather, it is crucial to its survival (Supplementary Fig. 3g). To confirm the importance of ferritinophagy in the survival of adaptive-resistant cells, we employed hydroxychloroquine (HCQ), 3-methyladenine (3-MA), or the inhibition of Tax1 Binding Protein 1 (TAX1BP1), the adaptor protein of NCOA4 that is involved in ferritinophagy, to inhibit ferritinophagy activity. The results demonstrated that inhibiting ferritinophagy significantly increased cell death in the adapted cells (Fig. 3m, n, Supplementary Fig. 5a–e).

## Knockout of NCOA4 significantly reduces the synthesis of iron-sulfur cluster (ISC) proteins and OXPHOS under Osi stress

To discern the mechanism by which NCOA4 mediates adaptive resistance to Osi and maintains high-OXPHOS levels in tumor cells, CRISPR-cas9 was employed to generate two NCOA4 knockout (sg-#1 and sg-#2) cell lines (Fig. 4a). As anticipated, the adapted cells show decreased interaction between downstream autophagosomes and lysosomes after NCOA4 knockout, which is reflected in the restoration of ferritin levels and the obstruction of autophagic flux (Fig. 4b–d, Supplementary Fig. 6a, b). Simultaneously, the NCOA4-knockout adapted cells display an apoptotic morphology, accompanied by inhibition upstream of autophagy (Fig. 4b, c, e, Supplementary Fig. 6a, b), consistent with the characteristics of crosstalk between apoptosis and autophagy[42]. Additionally, NCOA4 knockout led to a significant decrease in ferritinophagy and mitochondrial free iron ions in the adapted cells (Fig. 4f–h).

To further investigate the impact of NCOA4 knockout on OXPHOS in adaptive-resistant cells, the protein levels of ETC components were evaluated. The analysis revealed that the levels of NDUFB8 (a component of complex I), UQCRC2 (a component of complex III), MT-CO2 (a component of complex IV), and ATP5A (a component of complex V) did not show significant differences between NCOA4-wildtype and the two NCOA4-knockout cell lines at both baseline (0 h Osi treatment) and adapted cells (72 h Osi treatment) (Fig. 4i, j, l–n). Nevertheless, the expression of SDHB (a component of complex II) was significantly

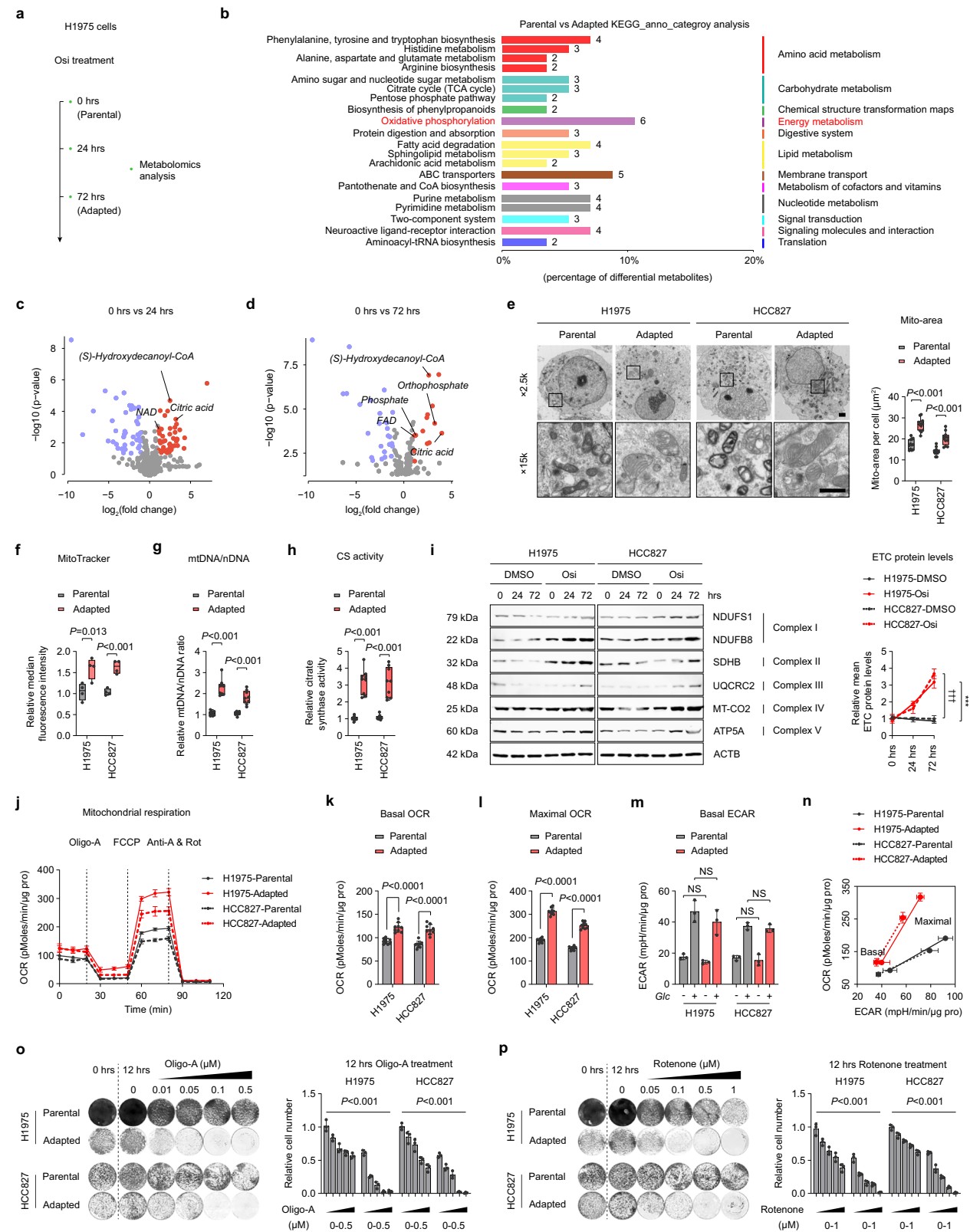

reduced in two NCOA4 knockout cells when compared to the control group both at baseline and after 72 h of Osi treatment (Fig. 4i, k). The complex II, also referred to as succinate dehydrogenase (SDH), which consists of succinic dehydrogenase and a series of iron-sulfur cluster (ISC) proteins that transfers electrons from succinic acid to coenzyme Q[43,44]. The ISC comprises iron and sulfur ions (Fe-S), which perform energy transfer functions as a cogroup of electron-transporting

proteins and participate in various biochemical reactions as an active group of some enzymes[45]. Based on our previous findings, we hypothesize that activated ferritinophagy in adaptive-resistant cells provides the iron ions essential for Fe-S synthesis that maintain respiratory chain activity.

To support this hypothesis, we confirmed the decreases in other ISC-containing ETC proteins in addition to complex II, including

**Fig. 2 | Adaptive resistance cells active oxidative phosphorylation (OXPHOS).**
**a** Metabolomics analysis of H1975 cells treated with Osi for 0 h (Parental), 24 h, and 72 h (Adapted) (*n* = 3 samples). **b** KEGG pathway enrichment analysis was performed on the metabolite comparing adapted cells versus parental cells. **c, d** Volcano plot displaying different metabolites after 24 h or 72 h Osi treatment in H1975 cells. **e** H1975 and HCC827 cells were harvested after 0 h (Parental) or 72 h (Adapted) Osi treatment and analyzed by electron microscopy for mitochondrial morphology (*n* = 4 technical replicates; 3 independent experiments). Scale bar, 1 μm. **f** Specific mean fluorescence intensity (MFI) of MitoTracker in Parental and Adapted cells. Representative experiment (*n* = 4 technical replicates; 3 independent experiments). **g** The ratio of mitochondrial DNA to nuclear DNA in Parental and Adapted cells. Representative experiment (n = 9 technical replicates; 3 independent experiments). **h** Relative citrate synthase (CS) activity in Parental and Adapted cells. Representative experiment (*n* = 9 technical replicates; 3 independent experiments). **i** Representative immunoblotting of ETC proteins in H1975 and HCC827 cells treated with DMSO or Osi for 0 h, 24 h, and 72 h (Representative images from 3 independent experiments). **j–l** The representative pattern of OCR over time **j**, basal OCR values **k** and maximal OCR values **l** normalized to total protein levels. Representative experiment (*n* = 3 technical replicates; 3 independent experiments). **m** Basal EACR in control conditions or the presence of 10 mM glucose (Glc) in Parental and Adapted cells. Representative experiment (*n* = 3 technical replicates; 3 independent experiments). **n** Cell Energy Phenotype analyses of the Parental and Adapted cells through real-time quantifications of ECAR and OCR at baseline or stressed with Oligo-A/FCCP. Representative experiment (*n* = 3 technical replicates; 3 independent experiments). **o, p** Colony formation assays of Parental and Adapted H1975 or HCC827 cells at 0 h, treated with DMSO for 12 h, and treated with an increasing concentration of oligo-A or rotenone for 12 h. Representative experiment (*n* = 3 technical replicates; 3 independent experiments). Data are shown as mean ± SD and were analyzed by a two-tailed unpaired t-test **e–l** or a two-way ANOVA **m, o, p**. NS no significance, ***$p < 0.001$. $^{†††}p < 0.001$, which was employed to specifically demonstrate the distinction in HCC827 cells. For box plots, the boxes extend from the 25th to 75th percentiles, with the median depicted by a horizontal line. Source data are provided as a Source Data file.

---

NADH:Ubiquinone Oxidoreductase Core Subunit S3 (NDUFS3, complex I component) and Ubiquinol-Cytochrome C Reductase, Rieske Iron-Sulfur Polypeptide 1 (UQCRFS1, complex III component) in NCOA4 knockout adapted cells (Fig. 5a). Additionally, we confirmed that Ferrochelatase (FECH; Heme synthesis pathway) and Dihydropyrimidine Dehydrogenase (DPYD; Pyrimidine catabolic pathway) containing ISC proteins were also significantly reduced at baseline and after 72 h of Osi treatment following NCOA4 knockout (Fig. 5a–e). Furthermore, we observed a significant increase in the levels of Iron Responsive Element Binding Protein 2 (IREB2) after the knockout of NCOA4, suggesting a significant decrease in ferritinophagy in both parental and adapted cells (Supplementary Fig. 6c). Decreased ISC protein activity, including respiratory chain complex II activity and FECH activity, was observed in response to NCOA4 depletion (Fig. 5f, g). We also confirmed the effect of NCOA4 knockout on OXPHOS activity in adapted cells. Consistent with expectations, the mitochondrial area per cell surface unit significantly diminished after NCOA4 depletion, and apoptotic morphology was observed (Fig. 5h). Furthermore, NCOA4 knockout resulted in significant reductions in mitochondrial content and activity, as indicated by decreased OCR and mitochondrial ATP content in the basal and maximum decoupling states of adaptive-resistant cells (Fig. 5i–r).

## Knockout of NCOA4 inhibits the formation of adaptative resistance by Osi in vitro and in vivo

We next assess the effects of NCOA4 deletion on formation adaptative resistance and explore the therapeutic relevance of NCOA4 as a target. Our findings showed that NCOA4 knockout cells were significantly more sensitive to Osi and did not exhibit adaptive resistance (Fig. 6a, b). Moreover, Osi treatment led to a notable increase in apoptosis in NCOA4 knockout tumor cells compared to NCOA4 wild-type tumor cells (Fig. 6c, d, Supplementary Fig. 7a, b). Additionally, 3-D culture pellet formation experiments demonstrated that Osi significantly inhibited the pellet formation of NCOA4 knockout cells and increased apoptosis (Fig. 6e).

To further validate whether NCOA4 knockout can avoid adaptive resistance and enhance Osi efficacy in vivo, we used a model of immunodeficient BALB/c nude mice implanted subcutaneously with NCOA4 wild-type or knockout tumor cells. Specifically, NCOA4 wild-type or NCOA4 knockout H1975/HCC827 tumor cells were subcutaneously transplanted into mice. Once the mice developed palpable tumors (about 100 mm³), they were randomly assigned into two treatment groups and administered either vehicle or Osi, and the subcutaneous tumor size was recorded every 5 days (Fig. 6f). Although NCOA4 knockout moderately inhibited the growth of subcutaneous tumors in the absence of Osi treatment for tumors derived by

H1975 sg-#1 cells, no significant inhibition was observed in H1975 sg-#2 and both two NCO14-knockout HCC827 cells (Fig. 6g, h, Supplementary Fig. 7d, e). However, the proliferation of tumors derived from two NCOA4 knockout cell lines treated with Osi significantly decreased compared with tumors derived from NCOA4 wildtype cell lines (Fig. 6g, h, Supplementary Fig. 7d, e).

Additionally, we investigated whether Osi could specifically eliminate a tumor containing a considerable fraction of NCOA4-knockout cells. We used immunodeficient mice with subcutaneous tumor models to examine this hypothesis. To simulate a clinically relevant situation in which cancer comprises both NCOA4-high and NCOA4-low tumor cells, H1975 cells stably transfected with EGFP and H1975 NCOA4 knockout cells stably transfected with mCherry were mixed at a 1:1 ratio and subcutaneously inoculated. When tumors became palpable (day 5), the mice were randomly assigned to receive either vehicle or Osi treatment (Fig. 6i). The tumors formed from a mixture of H1975 NCOA4 wild-type and NCOA4 knockout cells exhibited significantly reduced tumor growth under Osi treatment (Fig. 6j). Intriguingly, all mCherry-labeled NCOA4 knockout cells were eliminated specifically by Osi therapy at day 40, while only EGFP labeled wild-type tumor cells avoided Osi-induced death (Fig. 6k). These findings provide further evidence of the central role of NCOA4 in therapy adaptive resistance while demonstrating that NCOA4 knockout inhibits the formation of Osi adaptive resistance in vitro and in vivo.

## Active Fe-S protein synthesis in Osi adaptive-resistant tumor cells significantly reduced the threshold of cuproptosis

A recent study discovered that intracellular copper ions trigger the degradation of Fe-S protein, leading to cell death known as Cuproptosis[46]. Treatment that elevates intracellular copper ion levels and induces cell death via copper ionophores has been adopted in clinical trials[47,48]. Elesclomol, a copper ionophore, exhibits potent anticancer efficacy, and its combination with chemotherapy can enhance patient survival free from tumor[49,50]. Additionally, elesclomol proved to be more effective against malignant melanoma cells with high OXPHOS levels[51]. Given the higher OXPHOS levels in Osi adaptive-resistant cells, we evaluated whether elesclomol could specifically increase the death of adapted cells. As predicted, a low concentration (1 nM) of elesclomol plus $CuCl_2$ (1:1) considerably increased adaptive cell death, with no effect on parent cells (Fig. 7a). The same result was observed with two other copper ionophores, Disulfiram and NSC-319726, combined with $CuCl_2$ (1:1) (Fig. 7b, c).

These results strongly encourage us to explore options for treatment by combining Osi with copper ionophores at the beginning of treatment. The combination therapy combining Osi with low concentrations of copper ionophores was employed at the outset in H1975 and HCC827 cells, and surprisingly, it substantially inhibited

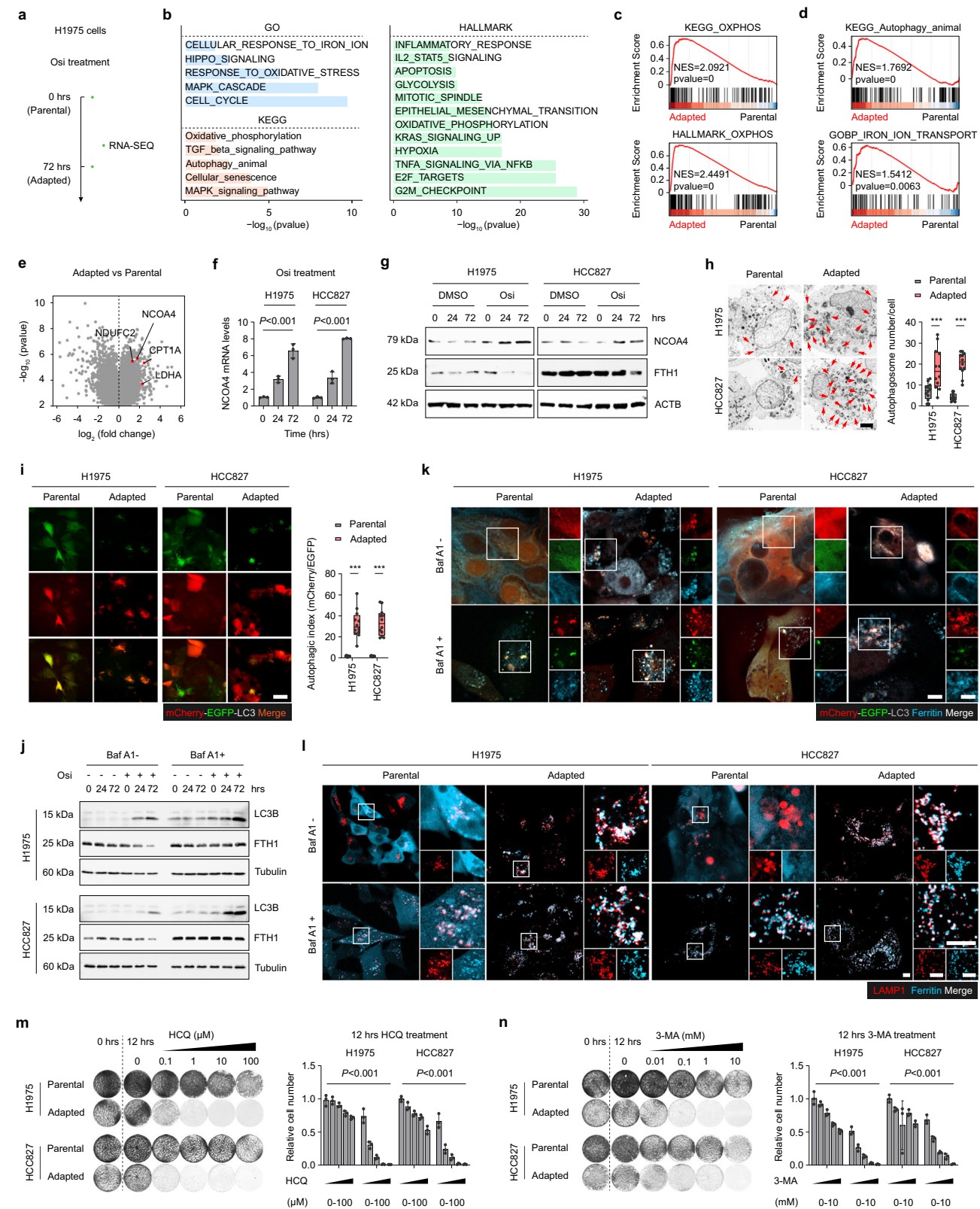

the development of adaptive resistance (Fig. 7d). A similar occurrence was observed in four other EGFR-mutated NSCLC cell lines (Fig. 7e, f). To investigate whether the differential sensitivity of parental cells and adapted cells to copper ions is based on the mechanism of cuproptosis, we evaluated key downstream effectors of cuproptosis. The expression levels of ISC proteins, as well as mitochondrial acyltransferase proteins, measured using a lipoic acid–specific antibody, were significantly reduced in adapted cells following co-treatment with copper ionophores (Fig. 7g, h). Furthermore, the combined treatment of Osi and copper ionophores effectively suppressed the activity of OXPHOS in adaptive-resistant cells (Fig. 7i–k).

We further sought to investigate whether the copper ionophore, in combination with Osi, could have a synergistic effect to

**Fig. 3 | NCOA4 mediates ferritinophagy and adaptative resistance to Osi. a** RNA-sequencing of Parental and Adapted H1975 cells (*n* = 3 samples). **b** Gene ontology (GO), KEGG pathway enrichment, and Hallmark analysis were performed comparing Adapted cells versus Parental cells. Barplot displaying the top 5 or top 12 pathways activated in adapted cells. **c, d** Gene Set Enrichment Analysis (GSEA) was performed comparing Adapted cells versus Parental cells. **e** Volcano plot displaying dysregulated genes in adapted cells. **f** NCOA4 mRNA expression levels were measured by qRT-PCR in H1975 and HCC827 cells treated with DMSO or Osi for 0 h, 24 h, or 72 h. Representative experiment (*n* = 3 technical replicates; 3 independent experiments). **g** Immunoblotting analysis of NCOA4 and FTH1 in H1975 and HCC827 cells treated with DMSO or Osi for 0 h, 24 h, or 72 h (Representative images from 3 independent experiments). **h** H1975 and HCC827 cells (Parental and Adapted) were harvested and analyzed by electron microscopy, the red arrow marks the autophagy vesicles (*n* = 4 technical replicates; 3 independent experiments). Scale bar, 2 μm. **i** Representative images of H1975 and HCC827 cells (Parental and Adapted) stably transfected with mCherry-EGFP-LC3B (*n* = 4 technical replicates; 3 independent experiments). Scale bar, 20 μm. **j** Immunoblotting analysis of cell lysates was performed to determine the levels of LC3B and FTH1 in H1975 and HCC827 cells treated with Osi for 0 h, 24 h, or 72 h followed by 2 h of treatment with DMSO or Baf A1 (Representative images from 3 independent experiments). **k** Immunofluorescence (IF) using an anti-Ferritin antibody (Cyan, false color) was performed in Parental and Adapted cells stably transfected with mCherry-EGFP-LC3B followed by 2 h of treatment with DMSO or Baf A1. Representative images (*n* = 4 technical replicates; 3 independent experiments). Scale bar, 10 μm. **l** IF using anti-Ferritin antibody (Cyan, false color) and anti-LAMP1 antibody (Red) was performed on Parental and Adapted cells followed by 2 h of treatment with DMSO or Baf A1. Representative images (*n* = 4 technical replicates; 3 independent experiments). Scale bar, 20 μm. **m, n** Colony formation assays of parental and adapted H1975 or HCC827 cells at 0 h, treated with DMSO for 12 h, and treated with an increasing concentration of HCQ or 3-MA for 12 h. Representative experiment (*n* = 3 technical replicates; 3 independent experiments). Data are shown as mean ± SD and were analyzed by a two-tailed unpaired t-test **h, i** or a two-way ANOVA **f–n**. ***$p < 0.001$. For box plots, the boxes extend from the 25th to 75th percentiles, with the median depicted by a horizontal line. Source data are provided as a Source Data file.

enhance its efficacy in vivo, the immunodeficient BALB/c nude mice were employed and implanted subcutaneously with H1975 and HCC827 cells. Specifically, H1975/HCC827 cells were subcutaneously transplanted into mice. After the mice developed palpable tumors (approximately 100 mm³), they were randomly assigned into three groups and administered vehicle, Osi, or Osi plus elesclomol, and the subcutaneous tumor's size was recorded every 5 days (Fig. 8a). While the use of elesclomol alone does not inhibit tumor growth (Supplementary Fig. 8a, b), it is encouraging to note that the Osi plus elesclomol regimen significantly limits tumor growth compared to Osi therapy (Fig. 8b-d). No increased toxicity was observed with the addition of elesclomol, as a low dose of elesclomol (5 mg/kg) was given to mice (Fig. 8e, f). Moreover, consistent results were obtained in subcutaneous tumor-bearing models using PC9 and HCC4006 cell lines (Supplementary Fig. 9a–h). Immunohistochemistry (IHC) analysis of the subcutaneous tumors in mice revealed that the combined treatment of Osi and elesclomol led to a significant reduction in intracellular ISC levels in the tumor cells (Fig. 8g).

Then, we implanted luciferase-labeled NCOA4 wild-type or NCOA4 knockout H1975 cells into the lungs of immunodeficient mice in situ and confirmed the inoculation by BLI after surgery. These mice were randomly assigned into three groups and treated with vehicle, Osi, or Osi plus elesclomol, and the tumor's size was measured using BLI every 10 days (Fig. 8h). The results showed that NCOA4 knockout significantly reduced tumor growth under Osi therapy, which was consistent with our previous findings (Fig. 8i, j). The combination therapy with Osi and elesclomol also significantly hindered the growth of both NCOA4 wild-type and NCOA4-knockout tumor cells (Fig. 8i, j). Moreover, mice undergoing Osi combination with elesclomol had the longest overall survival (Fig. 8k). Consistent with previous subcutaneous tumor-bearing animal models, no additional toxicity was observed in mice undergoing Osi and elesclomol combination therapy (Fig. 8l, m). IHC analysis of the lung tumors in mice demonstrated a significant reduction in intracellular ISC levels, along with increased apoptosis in the tumor cells with the combined treatment of Osi and elesclomol (Fig. 8n).

Finally, we validated the correlation between NCOA4 expression and decreased objective response rate (ORR) and shortened progression-free survival (PFS) in lung cancer patients receiving Osi treatment. To clinically validate our experimental findings, we performed NCOA4 immunostaining on tissue specimens that were obtained prior to Osi treatment from 40 EGFR-mutant lung cancer patients (Fig. 8o). An independent blinded pathological examination revealed a statistically significant association between NCOA4 expression and the decrease in ORR (Fig. 8p). Furthermore, a high expression of NCOA4 in cancer cells from pre-treatment samples was significantly correlated with a poorer PFS following Osi treatment, indicating that not only does NCOA4 expression induced by treatment matter, but also its high baseline expression prior to treatment can occur independently of Osi administration, potentially leading to resistance (Fig. 8q). Our data suggest that Osi treatment leads to NCOA4-mediated adaptive resistance, which activates ferritinophagy to sustain ISC protein synthesis and OXPHOS in adaptive-resistant cells. Nonetheless, it has been observed that adaptive-resistant cells exhibit heightened sensitivity to copper ions, and combination therapy with Osi and a low concentration of copper ionophores significantly inhibits the adaptive resistance of tumor cells.

## Discussion

In this study, we observed that Osi treatment can rapidly induce adaptive resistance in tumor cells, which is characterized by a reduction in apoptosis and the acquisition of rapid proliferation in the presence of the drug. We subsequently demonstrated that OXPHOS were significantly activated in adaptively resistant cells, and that adaptive-resistant tumor cells were significantly dependent on OXPHOS for survival. Furthermore, we showed that Osi treatment upregulates NCOA4, which mediates ferritinophagy and preserves ISC protein synthesis in the mitochondrial respiratory chain and OXPHOS. We identified that adaptive-resistant cells were sensitive to copper ions, which caused the mitochondria's degradation of Fe-S cluster proteins, leading to cuproptosis. Additionally, we found that Osi combined with copper ion ionophores significantly inhibits the production of adaptive resistance in tumor cells and improves the efficacy of Osi.

Acquisition of drug resistance in tumor cells is a long and complicated process that is believed to undergo resistance to treatment (adaptive resistance) at the beginning of treatment, as well as latent and transformation under drug pressure, leading to the acquisition of new mutations and irreversible resistance[5,6]. Understanding how adaptive resistance occurs in tumor cells is critical in defeating drug resistance. Cancer therapies that induce cellular stresses rapidly invoke non-genomic stress mitigation processes that sustain cell viability and thus represent key targetable resistance mechanisms. Our study found that tumors that adapt to Osi stress exhibit metabolic reprogramming, specifically a shift to OXPHOS. Our data suggest that metabolic reprogramming, especially changes in energy use propensity, mediates adaptive resistance to drug stress.

Many cancer cells prefer aerobic glycolysis for quick energy[52]. However, to survive metabolic stress associated with cancer treatment, cancer cells need to adjust their metabolism to unfavorable

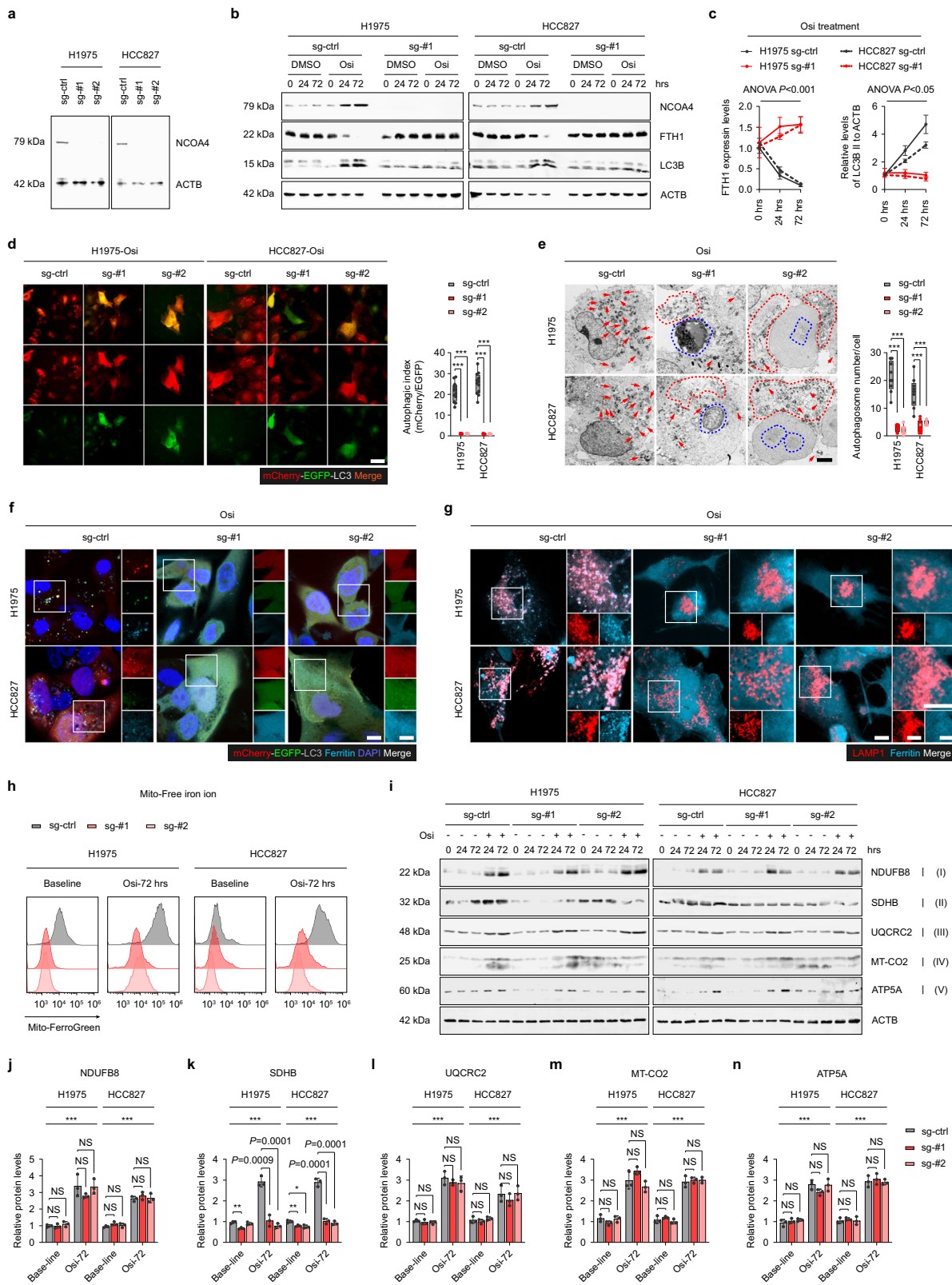

conditions by changing their energy consumption patterns[17]. OXPHOS is a highly efficient way of producing energy, which can enable tumor cells to obtain energy more efficiently under drug pressure and mediate adaptive resistance to drugs[23]. Although it has been suggested that following MAPK inhibition, DTP cells undergo a metabolic transition from glycolysis to OXPHOS, the drivers of DTP cell conversion to OXPHOS remain unknown[23]. Our study shows that

the activation of ferritinophagy is a key step in maintaining OXPHOS by adaptive-resistant cells, which provides free iron ions for the synthesis of key proteins in the TCA cycle and the ETC. However, how to maintain an appropriate iron ion concentration in adaptive cells to ensure the balance between ISC synthesis and avoidance of ferroptosis is indeed unknown, requiring our further exploration. Furthermore, in NCOA4-knockout cells that had not been treated

**Fig. 4 | Knockout of NCOA4 significantly reduces ferritinophagy under Osi stress. a** Immunoblotting analysis was performed to determine the protein levels of NCOA4 in NCOA4 wildtype (sg-ctrl, stably transfected with sg-ctrl), NCOA4 knockout-#1 (sg-#1) and NCOA4 knockout-#2 (sg-#2) H1975 and HCC827 cell lysates (Representative images from 3 independent experiments). **b, c** Immunoblotting analysis of cell lysates was performed to determine the levels of NCOA4, FTH1 and LC3B in NCOA4 sg-ctrl or sg-#1 H1975 and HCC827 cells treated with Osi for 0 h, 24 h, or 72 h (Representative images from 3 independent experiments). **d** Representative images of NCOA4 sg-ctrl, sg-#1 or sg-#2 H1975 and HCC827 cells stably transfected with mCherry-EGFP-LC3B under the treatments of Osi for 72 h ($n = 4$ technical replicates; 3 independent experiments). Scale bar, 20 µm. **e** NCOA4 sg-ctrl, sg-#1 or sg-#2 H1975 and HCC827 cells were harvested after 72 h of Osi treatment and analyzed by electron microscopy, the red arrow marks the autophagy vesicles. The red and blue dashed lines show broken cell membranes and crumpled nuclei of apoptotic cells ($n = 3$ technical replicates; 3 independent experiments). Scale bar, 2 µm. **f** Immunofluorescence using anti-Ferritin antibody (Cyan, false color) was performed in NCOA4 sg-ctrl, sg-#1 or sg-#2 H1975 and HCC827 cells stably transfected with mCherry-EGFP-LC3B and treated with Osi for 72 h. Representative images ($n = 4$ technical replicates; 3 independent experiments). Scale bar, 10 µm. **g** Immunofluorescence using an anti-Ferritin antibody (Cyan, false color) and anti-LAMP1 antibody (Red) was performed in NCOA4 sg-ctrl, sg-#1 or sg-#2 cells treated with Osi for 72 h. Representative images ($n = 4$ technical replicates; 3 independent experiments). Scale bar, 10 µm. **h** Flow cytometry analysis of NCOA4 sg-ctrl, sg-#1 or sg-#2 H1975 and HCC827 cells treated with Osi for 0 h or 72 h and stained with Mito-FerroGreen. Representative experiment ($n = 3$ technical replicates; 3 independent experiments). **i–n** Immunoblotting of five electron transport chain (ETC) proteins in NCOA4 sg-ctrl, sg-#1 or sg-#2 H1975 and HCC827 cells treated with DMSO or Osi for 0 h, 24 h, or 72 h (Representative images from 3 independent experiments). Data are shown as mean ± SD and were analyzed by a one-way ANOVA **d, e** or a two-way ANOVA **c, j–n** followed by a Dunnett-t test. NS no significance, ***$p < 0.001$. For box plots, the boxes extend from the 25th to 75th percentiles, with the median depicted by a horizontal line. Source data are provided as a Source Data file.

with Osi, we also observed instances of inhibited tumor growth in vivo. These findings underscore the significance of NCOA4-mediated ferritinophagy, not only in conferring resistance to Osi but also in its indispensable role in regulating iron homeostasis and metabolism in normal cells.

Adaptive-resistant cells also show sensitivity to 3-MA and HCQ, the autophagy inhibitors. Our previous studies and others have revealed that mitophagy autophagy mediates the development of drug resistance, and combining autophagy inhibitors can increase the efficacy of EGFR-TKIs plus IGF1R inhibitor or the KRAS inhibitors[53,54]. Additionally, our findings indicate that the targeted knockout of NCOA4, intended to suppress ferritinophagy, escalates apoptosis in adaptive-resistant cells. These cells display apoptotic morphology, coupled with an upstream autophagy inhibition. However, further investigations are required to substantiate the crosstalk between autophagy and apoptosis. Our study does not test the efficacy of Osi in combination with autophagy inhibitors as first-line agents in vivo, but it does suggest the possibility of combining targeted therapy with autophagy inhibition in cancer.

Understanding the mechanisms of adaptive resistance of tumor cells to EGFR-TKIs allows us to consider actionable adaptive resistance for optimal therapeutic outcomes and identify and target vulnerabilities arising from stress-relieving pathways in adaptive resistance. Our study shows that adaptive resistance cells highly dependent on OXPHOS not only show high sensitivity to OXPHOS inhibition but also exhibit vulnerability to copper ions. Elesclomol, a copper ionophore, has been used in clinical trials to increase intracellular copper ion levels and induce cell death[49–51]. Our study shows that the combination of Osi and elesclomol can significantly inhibit tumor adaptive resistance in vivo. The combination of low concentrations of elesclomol did not cause additional toxicity and significantly increased survival in the mice.

In conclusion, our findings suggest that NCOA4 mediates the mechanism by which ferritinophagy and OXPHOS lead to adaptive resistance to EGFR-TKIs. The combination of copper ionophores can significantly improve the initial efficacy of EGFR-TKIs. These data support the exploration of combining EGFR-TKIs and copper ionophores as a strategy to improve initial response rates and expand benefits in NSCLC patients with EGFR mutations.

## Methods

Our research adheres to all relevant ethical regulations. The animal experiments were conducted under the approval of the Nanjing Medical University Institutional Animal Care and Use Committee (IACUC) with approval number 2311034. Human tissue samples were obtained from the Department of Pathology at Jiangsu Cancer Hospital, and ethical approval was granted by Jiangsu Cancer Hospital Medical Ethics Committee of Nanjing Medical University.

### Compounds
All compounds used in this study are listed in Supplementary Table 1.

### Cell culture
The EGFR-mutant human NSCLC cell lines, including H1975, HCC827, PC9, HCC4006, H1650, and A431 were obtained from the American Type Culture Collection (ATCC). All cell lines were cultured in 1640 with 10% fetal bovine serum (FBS, Corning) at 37 °C in a humidified 5% $CO_2$ atmosphere. Before the experiment, the cells were screened for mycoplasma contamination, cross-contamination between species, and authenticity. The cell lines utilized in the experiments were cultured for a maximum of 20 passages.

### Tissue samples
The research conducted in this manuscript adhered to all relevant ethical regulations. In our study, the research involving human participants did not consider gender. The experimental design was based on the lack of reported efficacy, resistance mechanisms, or other gender-related considerations regarding Osi at present. We affirm that human research participants provided informed consent to participate in the study and for publication of their data in Supplementary Table 2.

### Animals
For animal experiments, four-week-old female BALB/c nude mice were obtained from GemPharmatech. All animals were housed in a pathogen-free environment at Nanjing Medical University, and were provided with adequate food and water, subjected to a 12-h light-dark cycle, and maintained at 22 °C with humidity levels between 55% to 70%. The IACUC allows a maximum tumor size of 2 cm, none of the trials exceeded this restriction.

**Orthotopic xenograft mouse model.** For the orthotopic xenograft mouse model in Fig. 1a, luciferase labeled H1975 or HCC827 cells ($5 \times 10^5$) with 50% Matrigel (Corning, 356234) were intrapulmonary injected into the mice. Lung orthotopic xenografts were evaluated by bioluminescence imaging (BLI) after five days of injection. The mice were randomly divided into two groups (five mice in each group) and given Vehicle control or Osi (10 mg/kg, once a day), respectively. Lung tumors were monitored every five days through BLI. After 40 days (H1975) or 50 days (HCC827), the mice were sacrificed, and their lungs were extracted, weighed, and fixed in 4% paraformaldehyde for Hematoxylin-eosin (H&E) and IHC analyses.

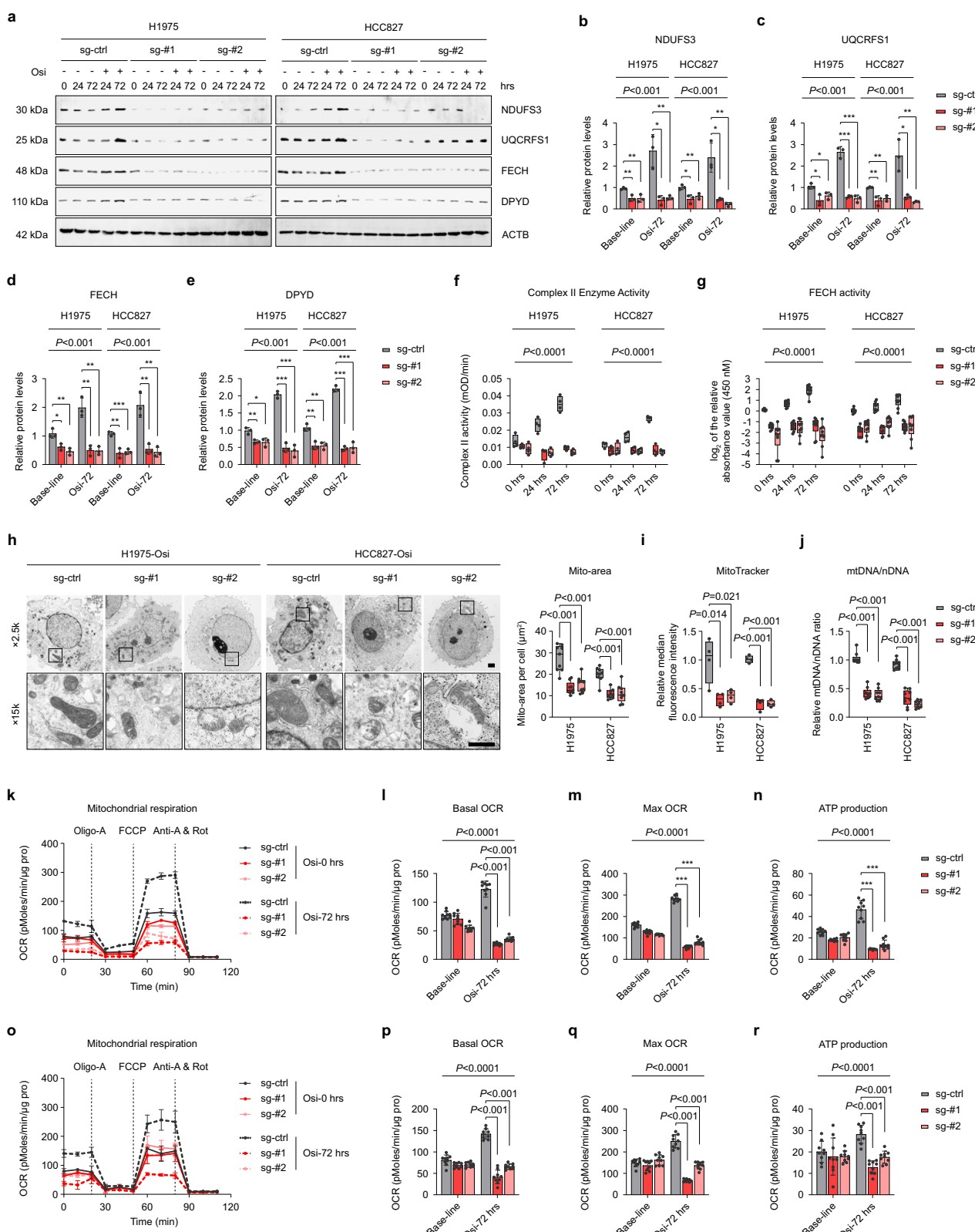

For the orthotopic xenograft mouse model in Fig. 8h, luciferase labeled NCOA4-wildtype or NCOA4-knockout H1975 cells ($5 \times 10^5$) with 50% Matrigel were intrapulmonary injected into the mice. Lung orthotopic xenografts were evaluated by BLI after ten days of injection. The mice were randomly divided into two groups (five mice in each group) and given Vehicle control, Osi (10 mg/kg, once a day), or Osi plus elesclomol (5 mg/kg), respectively. Lung tumors

and each mouse's weight were recorded every 10 days. After 40 days, the mice were sacrificed, and their lungs were extracted and weighed.

**Subcutaneous xenograft model.** For the subcutaneous xenograft model in Fig. 1f, H1975 or HCC827 cells ($1 \times 10^7$) with 50% Matrigel were injected subcutaneously into the mice. When the xenografts reached

**Fig. 5 | Knockout of NCOA4 significantly reduces the synthesis of iron-sulfur cluster (ISC) proteins and OXPHOS under Osi stress. a–e** Immunoblotting revealed ISC protein levels in the lysates of NCOA4 sg-ctrl, sg-#1 or sg-#2 H1975 and HCC827 cells treated with DMSO or Osi for 0 h, 24 h, and 72 h (Representative images from 3 independent experiments). ACTB is internal control. **f** Mitochondrial ETC complex II activity was detected in the lysates of NCOA4 sg-ctrl, sg-#1 or sg-#2 H1975 and HCC827 cells treated with DMSO or Osi for 0 h, 24 h, and 72 h. Representative experiment ($n = 4$ technical replicates; 3 independent experiments). **g** FECH activity was detected in the lysates of NCOA4 sg-ctrl, sg-#1 or sg-#2 H1975 and HCC827 cells treated with DMSO or Osi for 0 h, 24 h, and 72 h. Representative experiment ($n = 9$ technical replicates; 3 independent experiments). **h** NCOA4 sg-ctrl, sg-#1 or sg-#2 H1975 and HCC827 cells were harvested after 72 h of Osi treatment and analyzed by electron microscopy for mitochondrial morphology ($n = 3$ technical replicates; 3 independent experiments). Scale bar, 1 μm. **i** Specific mean fluorescence intensity (MFI) of MitoTracker in NCOA4 sg-ctrl, sg-#1 or sg-#2 H1975 and HCC827 cells. Representative experiment ($n = 4$ technical replicates; 3 independent experiments). **j** The ratio of mitochondrial DNA to nuclear DNA in NCOA4 sg-ctrl, sg-#1 or sg-#2 H1975 and HCC827 cells. Representative experiment ($n = 9$ technical replicates; 3 independent experiments). **k**, **o** Oxygen consumption rate (OCR) of NCOA4 sg-ctrl, sg-#1 or sg-#2 H1975 cells **k** and HCC827 cells **o** after 0 h or 72 h of Osi treatment. Representative experiment ($n = 3$ technical replicates; 3 independent experiments). **l–m**, **p–r** Basal OCR, Maximal OCR, and Mitochondrial ATP content in **k**, **o** normalized to total protein levels. Representative experiment ($n = 3$ technical replicates; 3 independent experiments). Data are shown as mean ± SD and were analyzed by a one-way ANOVA **h–j** or a two-way ANOVA **b–g**, **k–r** followed by a Dunnett-t test. NS no significance, *$p < 0.05$, **$p < 0.01$, ***$p < 0.001$. For box plots, the boxes extend from the 25th to 75th percentiles, with the median depicted by a horizontal line. Source data are provided as a Source Data file.

approximately 100 mm³, the mice were randomly divided into two groups (nine mice in each group) and administered Vehicle control or Osi (10 mg/kg, once a day), respectively. The size of subcutaneous tumors was measured every five days, and one mouse from each group was randomly sacrificed every week for Ki-67 analysis by IHC. After 40 days, the mice were sacrificed, and their subcutaneous tumors were extracted and weighed.

For the subcutaneous xenograft model in Fig. 6f, H1975 or HCC827 cells ($1 \times 10^7$) with 50% Matrigel were injected subcutaneously into the mice. When the xenografts reached approximately 100 mm³, the mice were randomly divided into two groups (five mice in each group) and administered Vehicle control or Osi (10 mg/kg, once a day), respectively. The size of subcutaneous tumors was measured every five days, and after 40 days, the mice were sacrificed, and their subcutaneous tumors were extracted and weighed.

For the subcutaneous xenograft model in Fig. 6i, a mixture of EGFP-labeled NCOA4 wildtype H1975 cells ($5 \times 10^6$) and mCherry-labeled NCOA4 knockout H1975 cells ($5 \times 10^6$) with 50% Matrigel were subcutaneously injected into the mice. When the xenografts reached approximately 100 mm³, the mice were randomly divided into two groups (five mice in each group) and given Vehicle control or Osi (10 mg/kg, once a day), respectively. The size of subcutaneous tumors was measured every five days, and after 40 days, the mice were sacrificed, and their subcutaneous tumors were extracted and prepared into a single-cell suspension for the analysis of each cell's proportion by flow cytometry analysis. To get a suspension of tumor cells, fresh tumors were mechanically and enzymatically disaggregated in a dissociation buffer consisting of 1640 medium containing 10% FBS (Corning), 100 U/mL collagenase type IV (Life Technologies), and 50 mg/mL DNase I (Roche). Suspension was incubated at 37 °C for 45 min and then further mechanically dissociated. Red blood cells were removed from samples using red blood cell lysis buffer (BioLegend).

For the subcutaneous xenograft model in Fig. 8a, H1975 or HCC827 cells ($1 \times 10^7$) with 50% Matrigel were injected subcutaneously into the mice. When the xenografts reached approximately 100 mm³, the mice were randomly divided into three groups (five mice in each group) and given Vehicle control, Osi (10 mg/kg, once a day), or Osi plus elesclomol (5 mg/kg), respectively. The size of subcutaneous tumors and each mouse's weight were measured every five days. After 35 days, the mice were sacrificed, and their subcutaneous tumors were extracted and weighed.

## Lentivirus and cell transfection
Lentivirus vectors expressing blasticidin-resistant luciferase, puromycin-resistant EGFP, puromycin-resistant mCherry, puromycin-resistant mVenus-p27K (purchased from Addgene), or puromycin-resistant mCherry-C1-EGFP-LC3B were used for lentivirus packaging after transfection into HEK-293T cells using Lipofectamine 3000

(Thermo). After 48 h, supernatants containing lentivirus particles were collected. H1975 and HCC827 cells were infected with lentiviruses in the presence of 2 μg/ml polybrene (MedChemExpress). To generate stable transfected monoclonal cell lines, cells were subjected to puromycin (2 μg/ml) or blasticidin (10 μg/ml) selection for one-week post-transfection. Resistant cells were resuspended, and single-cell plated into 96-well plates.

## H&E and IHC
H&E and IHC analyses were conducted on the formalin-fixed, paraffin-embedded mouse tumor tissue sections using hematoxylin-eosin stain solution or an anti-Ki-67 (Proteintech, 27309-1-AP) antibody. IHC was performed using an automated protocol designed for the BenchMark XT automated slide-staining system (Ventana Medical Systems) and detected with an ultraView Universal DAB detection kit (Ventana Medical Systems). Hematoxylin II (Ventana-Roche) was utilized as a counterstain.

## 3-D culture
To perform the 3-D cell culture, a 3-D Cell Culture Hydrogel Kit (Biozellen, Catalog: B-P-00002) was utilized in accordance with the manufacturer's instructions. Briefly, H1975 or HCC827 cells stably transfected with mVenus-p27K ($1 \times 10^5$) were suspended in 500 μL 1640 medium (containing DMSO or Osi at a final concentration of 1 μM) with 10% FBS and 500 μL gel. The resulting cell spheres were monitored daily for diameter measurement under a fluorescence microscope (Carl Zeiss).

## EdU assay
H1975, HCC827, PC9, or HCC4006 cells were inoculated into a 96-well template and treated with DMSO or Osi (1 μM). The culture medium was removed, and the cells were washed with ice-cold phosphate buffered saline (PBS) 3 times, and then replaced with fresh culture medium containing drugs every 24 h. 0 h, 24 h, or 72 h later, cells were incubated with 50 mM EdU for 2 h, fixed with 4% paraformaldehyde, and incubated with Apollo dye solution to label proliferative cells. Cell nuclei were counterstained by DAPI. The proliferating cells with green signal were observed by a fluorescent inverted microscope (Carl Zeiss).

## Cell cycle analysis
H1975, HCC827, PC9, and HCC4006 cells were subjected to Trypsin (Gibco) treatment, as described above, and subsequently harvested. The cells were washed with ice-cold PBS and then fixed overnight at −20 °C in 75% ice-cold ethanol before being stained with propidium iodide (PI) for 30 min at 37 °C. The DNA content of the cells was then measured using a BD Biosciences flow cytometer. The gating strategy for flow cytometry analysis is provided in Supplementary Fig. 10.

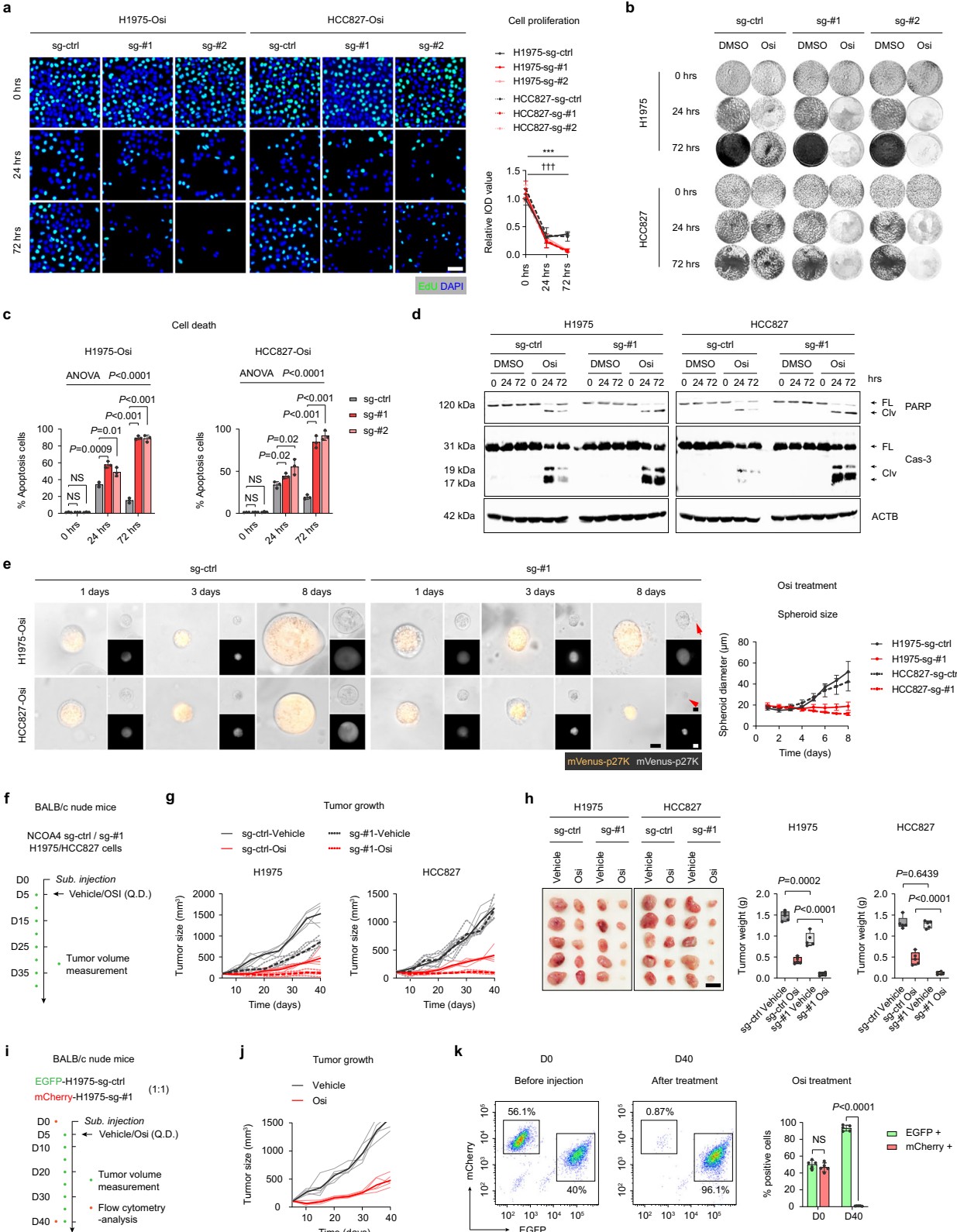

## PI/Annexin V apoptosis assay

Following the procedure outlined in the EdU assay, both floating and adherent H1975 and HCC827 cells were trypsinized and washed with PBS. To detect apoptotic cells, the Annexin V-FITC Apoptosis Detection Kit I (BD Biosciences) was employed, which uses Annexin V-FITC and PI to stain the cells. The manufacturer's instructions were closely followed for the staining process. The cells were then analyzed for the presence of apoptosis using a BD Biosciences flow cytometer.

## Cellular ROS analysis

Following the treatment procedure described above, the cells were incubated with 2 μM CellRox Reagent (Thermo) for 30 min at 37 °C in

**Fig. 6 | Knockout of NCOA4 inhibits the formation of adaptive resistance by Osi in vitro and in vivo. a** NCOA4 sg-ctrl, sg-#1 or sg-#2 H1975 and HCC827 cells treated with Osi for 0 h, 24 h, or 72 h and incubated with EdU for 2 h. Representative experiment ($n = 3$ technical replicates; 3 independent experiments). Scale bar, 50 μm. **b** Colony formation assays of NCOA4 sg-ctrl, sg-#1 or sg-#2 H1975 and HCC827 cells treated with DMSO or Osi for 0 h, 24 h, and 72 h. Representative experiment ($n = 3$ technical replicates; 3 independent experiments). **c** Flow cytometry analysis of cell apoptosis of NCOA4 sg-ctrl, sg-#1 or sg-#2 H1975 and HCC827 cells treated with Osi for 0 h, 24 h, or 72 h. Representative experiment ($n = 3$ technical replicates; 3 independent experiments). **d** Immunoblotting analysis of apoptosis-related proteins poly ADP-ribose polymerase (PARP) and Capase-3 (Cas-3) in NCOA4 sg-ctrl or sg-#1 H1975 and HCC827 cells treated with DMSO or Osi for 0 h, 24 h, or 72 h (Representative images from 3 independent experiments). **e** Left, Representative images of NCOA4 sg-ctrl or sg-#1 H1975 and HCC827 cells treated with Osi in 3-D culture on day 1, day 3 and day 8. The red arrow shows broken cell membranes. Scale bar, 10 μm. Right, the growth curves of tumor cells. Representative

experiment ($n = 3$ technical replicates; 3 independent experiments). **f**, **g** BALB/c nude mice were subcutaneous inoculated with NCOA4 sg-ctrl or sg-#1 H1975 and HCC827 cells and were given Vehicle (DMSO + saline) or Osi (nasal feeding, once a day) and follow-up ($n = 5$ mice each group). **h** Tumor weight was measured for the Vehicle-treated group and Osi-treated group ($n = 5$ mice each group). Scale bar, 1 cm. **i**, **j** BALB/c nude mice were subcutaneous inoculated with EGFP-labeled NCOA4 sg-ctrl or mCherry-labeled NCOA4 sg-#1 H1975 (mix in a ratio of 1: 1) cells and were given Vehicle (DMSO + saline) or Osi (nasal feeding, once a day) and follow-up (n = 5 mice each group). **k** Flow cytometry analysis of mice tumors from **j** at day 0 (before inoculating the mice) or at day 40 ($n = 5$ samples). Data are shown as mean ± SD and were analyzed by a two-way ANOVA **a**, **c**, **h**, **k** followed by a Dunnett-t test. *NS* no significance, *$p < 0.05$, **$p < 0.01$, ***$p < 0.001$. †††$p < 0.001$, which was employed to specifically demonstrate the distinction in HCC827 cells. For box plots, the boxes extend from the 25th to 75th percentiles, with the median depicted by a horizontal line. Source data are provided as a Source Data file.

the absence of light. The level of ROS in the cells was then quantified using a flow cytometer (BD Biosciences).

### RNA extraction and qRT-PCR
Total cellular RNA was isolated using the TRIzol reagent (Thermo) and then subjected to reverse transcription using the RNA-to-cDNA kit (Takara). To quantify RNA expression, qRT-PCR analysis with SYBR Green Premix (Vazyme) was performed using the QuantStudio 6 Flex system (ABI) with ACTB serving as an internal control. The primer sequences for qRT-PCR are specified in Supplementary Table 3.

### MtDNA/nDNA quantitative analysis
The Ratio of mtDNA/nDNA in parental and adapted cells was determined using the NovaQUANT™ Human Mitochondrial to Nuclear DNA Ratio Kit (Sigma-Aldrich, 72620).

### Protein extraction and immunoblotting
The cells were washed three times with ice-cold PBS, then harvested by centrifugation at 1000 rpm for 5 min, and lysed using RIPA buffer (Thermo) containing phosphatase and protease inhibitor cocktail (Thermo) for 30 min at 4 °C. Following lysing, the supernatant was collected by centrifugation at 15,000 g for 15 min at 4 °C. Protein concentration was determined using the BCA Protein Assay (Beyotime).

For the immunoblotting assay, the denatured lysates were separated into 10% or 15% SDS-PAGE gels and then transferred to polyvinylidene fluoride (PVDF) membranes (Millipore). Subsequently, the membranes were blocked using 5% non-fat dry milk for 2 h at room temperature and then incubated with the primary antibody overnight at 4 °C. The membranes were subjected to standard immunoblotting procedures using Alexa 680 and 800-conjugated species-specific secondary antibodies (Rockland). The membranes were visualized using an infrared scanner (LI-COR). The primary antibodies employed were as follows: anti-AKT (Cell Signaling, 9272 S), anti-phospho-AKT Ser473 (Cell Signaling, 4060 S), anti-ERK1/2 (Cell Signaling, 4695 S), anti-phospho-ERK1/2 Thr202/Tyr204 (Cell Signaling, 4370 S), anti-S6 (Cell Signaling, 2217 S), anti-phospho-S6 Ser235/236 (Cell Signaling, 4858 S), anti-PARP (Cell Signaling, 9542 S), anti-Caspase-3 (Cell Signaling, 9662 S), anti-NDUFS1 (Abcam, ab169540), anti-NDUFB8 (Abcam, ab192878), anti-SDHB (Abcam, ab175225), anti-UQCRC2 (Abcam, ab203832), anti-MTCO2 (Abcam, ab79393), anti-ATP5A (Abcam, ab14748), anti-NCOA4 (Cell Signaling, #66849), anti-FTH1 (Abcam, ab183781), anti-NDUFS3 (Abcam, ab177471), anti-UQCRFS1 (Abcam, ab191078), anti-FECH (Abcam, ab137042), anti-DPYD (Cell Signaling, #4654), anti-IREB2 (Abcam, ab181153), anti-DLAT (Cell Signaling, #12362), anti-DLST (Cell Signaling, # 5556), anti-Tubulin (Proteintech, 10068-1-AP) and anti-ACTB (Proteintech, 81115-1-RR). The antibodies were employed according to the manufacturer's recommended dilutions.

### Metabolites extraction and LC-MS analysis
H1975 cells were treated with Osi for 0, 24 or 72 h and were subsequently frozen in liquid nitrogen. The LC-MS system for metabolomics analysis is composed of Waters Acquity I-Class PLUS ultra-high performance liquid tandem Waters Xevo G2-XS QTof high-resolution mass spectrometer. The column used is purchased from Waters Acquity UPLC HSS T3 column (1.8 um, 2.1 × 100 mm). Positive ion mode: mobile phase A: 0.1% formic acid aqueous solution; mobile phase B: 0.1% formic acid acetonitrile. Negative ion mode: mobile phase A: 0.1% formic acid aqueous solution; mobile phase B: 0.1% formic acid acetonitrile. Injection volume was 1 μL. The Waters Xevo G2-XS QTof high-resolution mass spectrometer is capable of primary and secondary mass spectrometry data acquisition in MSE mode controlled by acquisition software (MassLynx v4.2, Waters). In each data acquisition cycle, dual-channel data acquisition can be performed for both low and high collision energies. The low-impact energy is 2 V, the high-impact energy range is 10–40 V and the scanning frequency is 0.2 s. ESI ion source parameters are as follows [3]: Capillary voltage, 2500 V (positive mode) or −2000 V (negative mode); cone hole voltage, 30 V; ion source temperature, 100 °C; desolvent temperature, 500 °C; back blowing gas flow rate, 50 L/h; desolvent gas flow rate, 800 L/h; plastic-nucleus ratio collection range, 50–1200 m/z.

The raw data collected using MassLynx V4.2 is processed by Progenesis QI software for peak extraction, peak alignment, and other data processing operations, based on the Progenesis QI software online METLIN database and Biomark's self-built library for identification, and at the same time, theoretical fragment identification and mass deviation. All are within 100 ppm. After normalizing the original peak area information with the total peak area, the follow-up analysis was performed. Principal component analysis and Spearman correlation analysis were used to judge the repeatability of the samples within group and the quality control samples. The identified compounds are searched for classification and pathway information in KEGG, HMDB, and lipidmaps databases. According to the grouping information, calculate and compare the difference multiples, a t-test was used to calculate the difference significance p-value of each compound. The R language package ropls was used to perform OPLS-DA modeling, and 200 times permutation tests were performed to verify the reliability of the model. The VIP value of the model was calculated using multiple cross-validation. The method of combining the difference multiple, the p-value and the VIP value of the OPLS-DA model was adopted to screen the differential metabolites. The screening criteria are FC > 1, *p*-value < 0.05 and VIP > 1. The difference metabolites of KEGG pathway enrichment significance were calculated using a hypergeometric distribution test. A list of the expression of metabolites is provided in Supplementary Data 1.

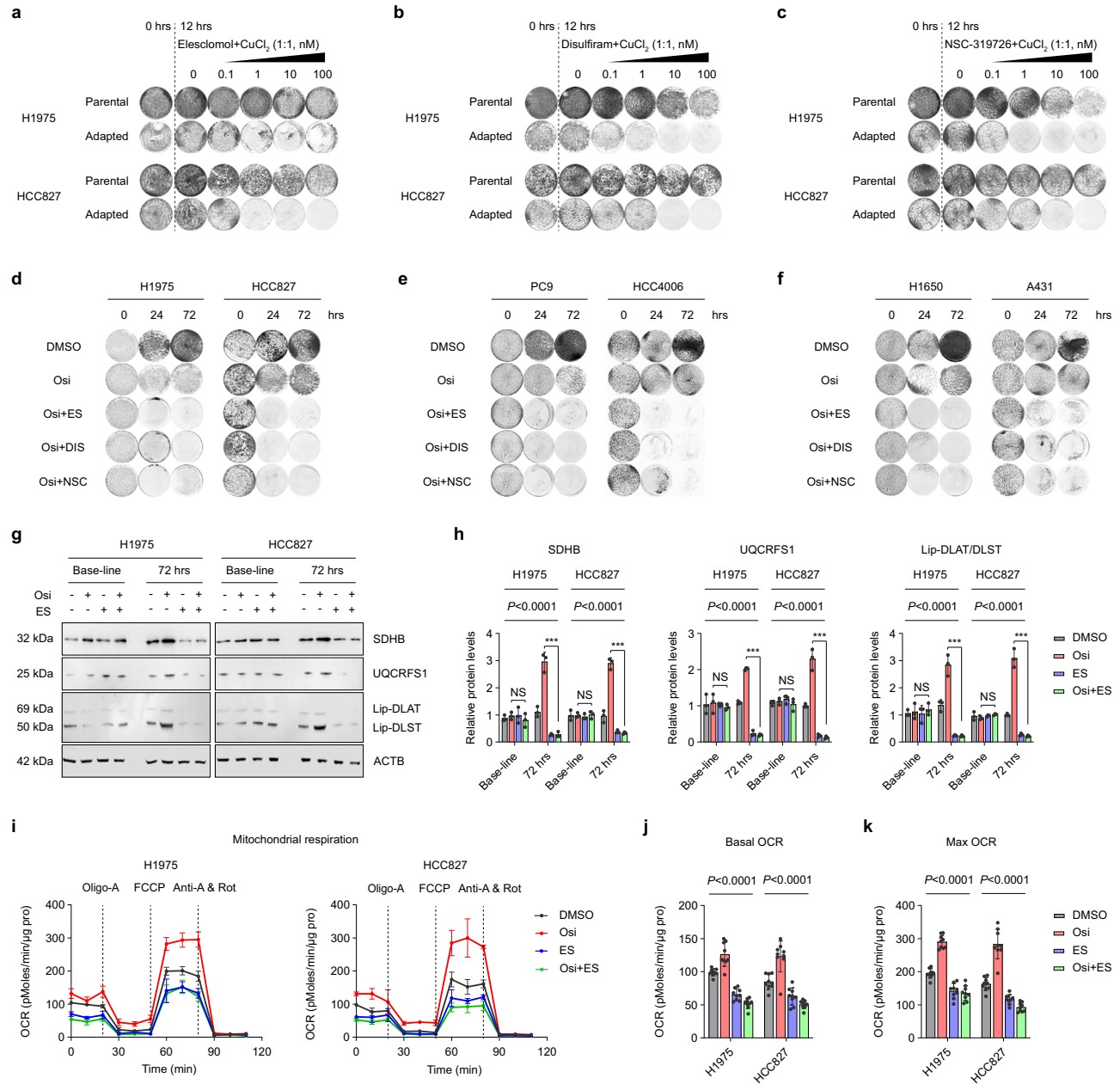

**Fig. 7 | Active Fe-S protein synthesis in Osi adaptive-resistant tumor cells significantly reduced the threshold of Cuproptosis. a**–**c** Colony formation assays of parental and adapted H1975 or HCC827 cells at 0 h, treated with DMSO for 12 h, and treated with an increasing concentration of elesclomol plus $CuCl_2$ **a**), an increasing concentration of disulfiram plus $CuCl_2$ **b** or an increasing concentration of NSC-319726 plus $CuCl_2$ **c** for 12 h. Representative experiment (*n* = 3 technical replicates; 3 independent experiments). **d**–**f** Colony formation assays of H1975, HCC827, PC9, HCC4006, H1650 and A431 cells treated with DMSO, Osi, Osi plus elesclomol and $CuCl_2$ (Osi + ES), Osi plus disulfiram and $CuCl_2$ (Osi + DIS), Osi plus NSC-319726 and $CuCl_2$ (Osi + NSC) for 0 h, 24 h, and 72 h. Representative experiment (*n* = 3 technical replicates; 3 independent experiments). **g, h** Immunoblotting of iron-sulfur cluster (ISC) proteins SDHB, UQCRFS1 and mitochondrial acylated proteins Lip-DLAT, Lip-DLST in H1975 and HCC827 cells treated with DMSO, Osi, ES, or Osi combined with

ES for 0 h (Baseline) or 72 h (Representative images from 3 independent experiments). ACTB is internal control. **i** Oxygen consumption rate (OCR) of H1975 cells (left) and HCC827 cells (right) after 72 h of treatment of DMSO, Osi, ES, or Osi combined with ES. Oligomycin A (Oligo-A), carbonyl cyanide-4-(trifluoromethoxy) phenylhydrazone (FCCP), antimycin A (Anti A), and rotenone (Rot) were added to measure basal OCR, ATP content, maximal OCR, and non-mitochondrial OCR. Representative experiment (*n* = 3 technical replicates; 3 independent experiments). **j, k** Basal OCR and maximal OCR in **i** normalized to total protein levels. Representative experiment (*n* = 3 technical replicates; 3 independent experiments). Data are shown as mean ± SD and were analyzed by a two-way ANOVA. *NS* no significance, *$p < 0.05$, **$p < 0.01$, ***$p < 0.001$. Source data are provided as a Source Data file.

## Seahorse assay

The cells were seeded into XFe96 Cell Culture Microplates (Bioscience) and treated as described previously. Before the experiment, the culture medium was aspirated from each well and replaced with 175 μL of serum-free unbuffered Seahorse XF Base Medium base pH 7.4 containing glucose (10 mM), glutamine (2 mM), and Pyruvate (1 mM) (For OCR) or with 0 mM glucose (for ECAR), all pre-warmed to 37 °C. OCR and ECAR were assessed over a specific period before and after administration of the following compounds: For OCR measurement, the compounds used were 1 mM oligomycin,

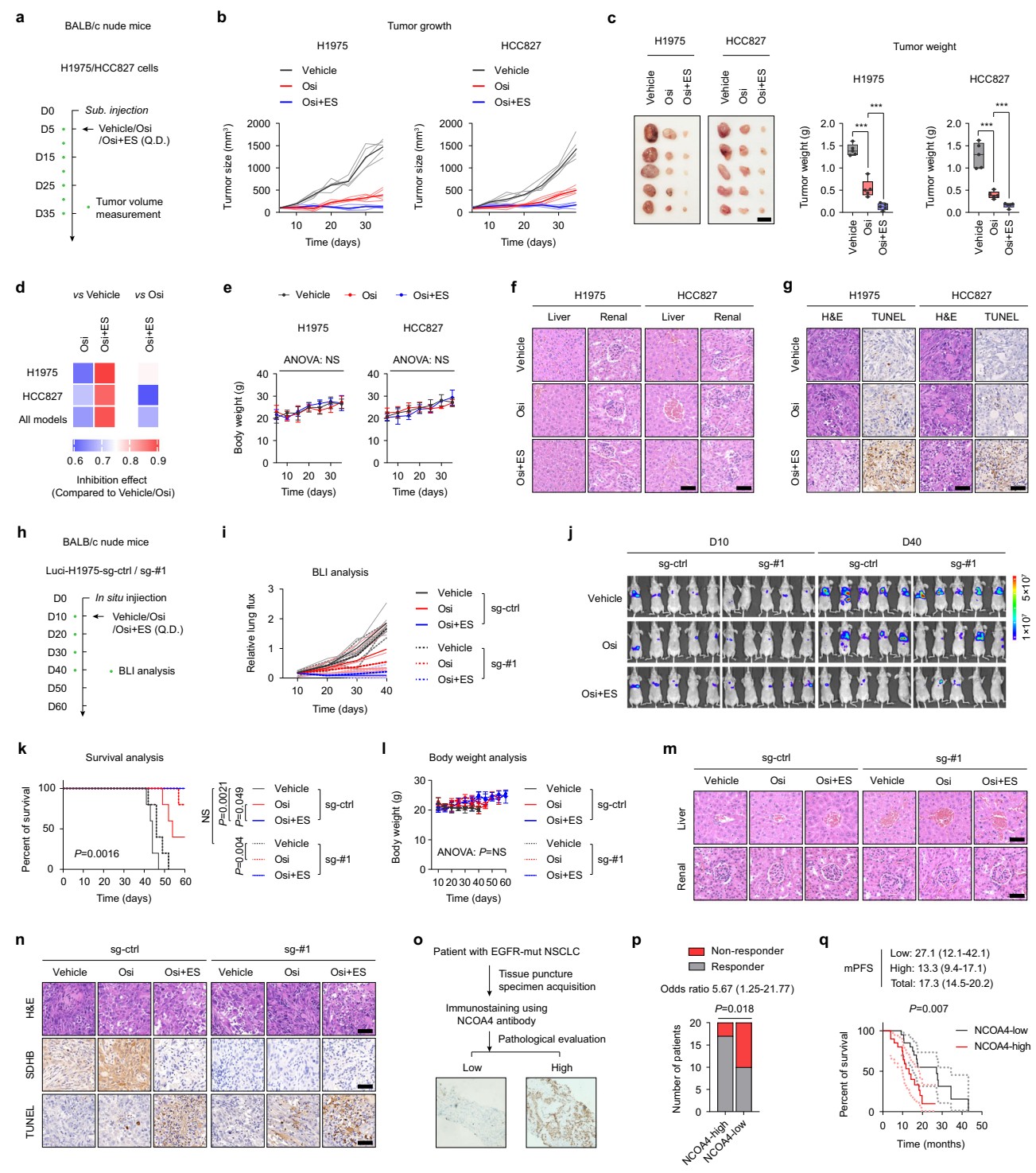

0.5 mM FCCP, and 0.5 mM antimycin A + rotenone. For ECAR measurement, the compounds used were 10 mM glucose, 1 mM oligomycin, and 50 mM 2-deoxyglucose (2-DG). Each injection port received a volume of 25 µL of the compound, and three baseline measurements were taken before the addition of any compounds. After a 3-min wait, three response measurements were taken after each compound addition. The OCR and ECAR values were normalized to the total amount of protein per well. The OCR and ECAR data points represent the average rates during the measurement cycles and were reported as absolute rates (pMoles/min for OCR, mpH/min for ECAR).

## Colony formation assay

To evaluate the responses of H1975 and HCC827 cells to various treatments, the cells were exposed to Osi for 0 h (Parental) or 72 h (Adapted) and then washed with PBS and cultured with a drug-containing medium for 12 h. The colonies were washed gently with PBS, fixed in 4% paraformaldehyde for 20 min at room temperature, and stained with 0.5% crystal violet.

## RNA-sequencing and analysis

Total RNA was extracted from H1975 cells treated with Osi for 0 or 72 h using standard procedures. RNA sequencing analysis was conducted

**Fig. 8 | Combination therapy with copper ionophore increases the efficacy of Osi. a, b** BALB/c nude mice were subcutaneous inoculated with H1975 or HCC827 cells and were given Vehicle, Osi, or Osi plus elesclomol (Osi + ES) and follow-up (*n* = 5 mice each group). **c** Tumor weight was measured for the vehicle-treated group, Osi-treated group and Osi plus ES group (*n* = 5 mice each group). Scale bar, 1 cm. **d** Generalized linear models to test the association of change in tumor volume over time, either between Osi treatment groups and vehicle (left) or between combination therapy and Osi alone (right) (*n* = 5 mice each group). **e** Mouse weight was measured from **b** as a surrogate for treatment toxicity (*n* = 5 mice each group). **f** Representative H&E-stained images of mouse liver and kidney from **b** were used as a surrogate for assessing treatment toxicity (*n* = 5 samples). Scale bar, 50 μm. **g** Representative H&E and TUNEL images by staining consecutive sections of mouse tumor tissues from **a** (*n* = 5 samples). Scale bar, 50 μm. **h** BALB/c nude mouse lungs were orthotopically transplanted with luciferase labeled NCOA4 sg-ctrl or NCOA4 sg-#1 H1975 cells and were given Vehicle, Osi, or Osi plus ES and follow-up using BLI every 10 days (*n* = 5 mice each group). **i** Quantification of the relative fluorescence activity of the ROI in mice by BLI reflects the growth of mice lung tumors (n = 5 mice each group). **j** Representative BLI images of mice after Vehicle, Osi or Osi plus ES treatment at day 10 and day 40, respectively (*n* = 5 mice each group). **k** Kaplan-Meier survival analysis of BALB/c nude mice in **h** (n = 5 mice each group). **l** Mouse weight was measured from **h** as a surrogate for treatment toxicity (*n* = 5 mice each group). **m** Representative H&E-stained images of mouse liver and kidney from **h** were used as a surrogate for assessing treatment toxicity (*n* = 5 samples). Scale bar, 50 μm. **n** Representative H&E-stained, IHC for SDHB and TUNEL images by staining consecutive sections of mouse tumor tissues from **h** (*n* = 5 samples). Scale bar, 50 μm. **o** Schematic representation of the analysis of patient samples for NCOA4 expression in cancer cells. NCOA4 immunostaining was performed on tissue specimens from 40 lung cancer patients with a validated EGFR mutation that was obtained prior to first-line Osi treatment. **p** Analysis of the association between NCOA4 expression in the patient **o** and the response to Osi. **q** Kaplan-Meier plot for PFS of Osi-treated patients from **o**. Data are shown as mean ± SD and were analyzed by a two-tailed one-way ANOVA **c**, a two-tailed two-way ANOVA **e**, **l**, a log-rank test **k**, **q**, or a two-tailed chi-square test **p**. NS no significance, *$p < 0.05$, **$p < 0.01$, ***$p < 0.001$. For box plots, the boxes extend from the 25th to 75th percentiles, with the median depicted by a horizontal line. Source data are provided as a Source Data file.

following the manufacturer's guidelines. DESeq2 was used for differential expression analysis, and differentially expressed genes (FDR < 0.05) were submitted to KEGG analysis and GSEA. The differentially expressed genes are listed in Supplementary Data 2.

### Autophagy flux assay
H1975 and HCC827 cells were transfected with mCherry-C1-EGFP-LC3B, and then plated in confocal dishes. The cells were treated as illustrated in the figure, and LC3 puncta were visualized using fluorescence confocal microscopy (Carl Zeiss).

### Immunofluorescence
Immunofluorescence staining was carried out using a specific antibody against Ferritin (Abcam, ab75973). NCOA4 knockout and wildtype H1975 and HCC827 cells stably transfected with mCherry-EGFP-LC3B were fixed with 4% formaldehyde for 15 min and then blocked for 60 min at room temperature using 5% normal goat serum. Immunostaining was performed with Alexa Fluor 647-labeled secondary antibodies (Thermo). DAPI was used to counterstain the nuclei.

### CRISPR-cas9
The single guide RNAs (sgRNAs) targeting NCOA4 exons were created using the CRISPR Design Tool (crispr.mit.edu). The sequence of NCOA4-targeting sgRNAs is: NCOA4 KO #1: TAGCTGTCCCTTTCAGCGAA; NCOA4 KO #2: GCATGAGCCATCAAGTGCTC. The NCOA4-targeting sgRNAs were cloned into eSpCas9-2A-GFP plasmid (GenScript). Cells transfected with plasmids 36 h later were selected for GFP-positive clusters by flow cytometry analysis. GFP-positive cells were resuspended and planted into 96-well plates as single cells. Finally, NCOA4 knockout clones were tested by immunoblotting four weeks later.

### Mito-free iron ion detection
The Mito-Free iron ion levels were detected according to the manufacturer's guidelines using Mito-FerroGreen (MFG). The levels of mitochondrial free iron ions were measured using a flow cytometer (BD Biosciences).

### Complex II enzyme activity
The complex II enzyme activity analysis assay was carried out according to the manufacturer's guidelines using a complex II enzyme activity microplate assay kit (Abcam, ab109908).

### FECH activity
The FECH activity analysis assay was performed using a Sandwich ELISA Kit for FECH (antibodies-online, Catalog No. ABIN6966713), in accordance with the manufacturer's guidelines. To prepare a solid phase carrier, the FECH antibody was coated on a 96-well microplate. The micropores were then loaded with standard samples of H1975 and HCC827 cell lysates. FECH was combined with the antibody on the solid phase carrier, followed by the addition of biotinylated FECH antibodies. Enzyme-labeled avidin was then added, thoroughly washed and TMB substrate was added to develop color. The od value absorbance was measured at a 450 nm wavelength using an enzyme-labeler, and the sample concentration was calculated accordingly.

### Statistics & reproducibility
All data were obtained from independent experiments with independent technical replicates. No statistical method was used to predetermine sample size. Most results were qualitatively replicated in two different cell lines and provided consistent results using independent techniques. The Investigators were not blinded to allocation during experiments and outcome assessment. No data were excluded from the analyses. Experimental data are presented as mean and individual points, mean and standard deviation as indicated in the figure legend. Before statistical analysis, tests were performed with the assumption that values followed a normal distribution and had similar variances. To assess the differences between a control group and an experimental group, an unpaired two-tailed Student's t-test was employed. One-way or multivariate ANOVA was utilized to test for the statistical significance of a control group and multiple experimental groups, and the Dunnett-t test was employed to evaluate pairwise differences between means of multiple samples in experimental tests. For certain specialized statistical methods, the corresponding details can be found in the figure legends that correspond to the results. SPSS v.28.0 and Prism v.9.4 were used for statistical analysis.

### Reporting summary
Further information on research design is available in the Nature Portfolio Reporting Summary linked to this article.

## Data availability
The metabolomics data generated in this study has been deposited in the MetaboLights database under accession code MTBLS8108. The RNA-seq data generated in this study has been uploaded to the Gene Expression Omnibus (GEO) database and is associated with the accession number GSE236238. Source data are provided with this paper. All the other data are available within the article and its Supplementary Information. Source data are provided with this paper.

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

## Acknowledgements

Funding for this study was provided by the National Natural Science Foundation of China (Grant No. 82372762, F.J., 82002434, F.J., 82003106, 81702892), the Project of Invigorating Health Care through Science, Technology and Education, Jiangsu Provincial Medical Innovation Team (CXTDA2017002, F.J.), and the Project of Invigorating Health Care through Science, Technology and Education, Jiangsu Provincial Medical Outstanding Talent (JCRCA2016001, F.J.).

## Author contributions

F.J. and H.W. designed experiments and wrote the manuscript. Y.F. performed statistical analysis. H.W., Q.H., Y.C., R.L., X.Y., X.S., Y.L., T.Z., and Y.S. completed the basic experiment part. L.X. and Y.C. were responsible for clinical sample collection and subsequent sample delivery. X.H. is responsible for scoring the IHC. G.D. helped to revise the manuscript. All authors read and approved the final manuscript.

## Competing interests

The authors declare no competing interests.
