## [Peer Review File · Nature Communications]

Ferritinophagy mediates adaptive resistance to EGFR tyrosine kinase inhibitors in non-small cell lung cancerREVIEWER COMMENTS

Reviewer #1 (Remarks to the Author); expert in iron metabolism, ferritinophagy:

Wang and co-authors present a manuscript evaluating mechanisms of resistance to EGFR-TKI in non-small lung cancer and through a series of metabolomic, cell biologic, and in vivo analyses propose that NCOA4-mediated ferritinophagy is a targetable mechanism of resistance to EGFR-TKI inhibition. The findings are of significant interest; however, there are several points that require clarification:

Major

1. The observation that NCOA4 KO decreases global autophagic flux in response to OSI treatment is unexpected (Fig. 4e). NCOA4 is an autophagic adaptor for ferritin that is downstream of autophagy activation machinery. How do the authors propose that KO of NCOA4 impacts global autophagy activation as previously demonstrated in their figures (LC3 western, LC3 cherry/gfp flux experiments)?

2. Figure 4i and j: the baseline levels of ETC proteins appear to decrease in WT vs. KO cells (no OSI treatment). Are the quantifications in 4i based on relative amount at 0h separately in the WT and KO conditions? This might be misleading as the baseline level of the proteins (without OSI) appears to be different.

3. Fig. 5a: IREB2 does not contain an ISC. Rather its levels are controlled in response to the free iron levels in the cell. In that regard the results here are inconsistent with prior knowledge about IREB2 levels in response to NCOA4 KO. Specifically loss of NCOA4 decreases cellular free iron thereby leading to FBXL5 degradation and stabilization of IREB2 (which is a target of FBXL5). The authors should take another look at the IREB2 results and include a positive control such as DFO to ensure they are seeing the expected changes in IREB2. Regardless - IREB2 is not an ISC containing protein and the changes they are seeing here are inconsistent with prior published data.

4. The premise of Figure 7 is that elesclomol will induce degradation of Fe-S proteins and that OSI-adaptive/resistant cells will be especially sensitive to elesclomol because of the importance of Fe-S proteins in supporting oxphos. Figure 7 shows synergy between OSI and elesclomol but there are no experiments to demonstrate the on-target efficacy of elesclomol with OSI to induce degradation of Fe-S proteins. Can the authors more specifically show that elesclomol is working how they propose in their model? It would seem that the last experiment where OSI is combined with NCOA4 KO and elesclomol would not necessarily demonstrate increased response compared to OSI + NCOA4 KO as by their proposed mechanism, NCOA4 should be upstream of elesclomol activity. Can the authors show that NCOA4 KO decreases Fe-S proteins and that elesclomol on top of NCOA4 KO further decreases Fe-S proteins as a justification of why there is additive effect of elesclomol on top of NCOA4 KO?

As it relates to the discussion, this statement "We identified that adaptive-resistant cells were sensitive to copper ions, which cause the mitochondria's degradation of Fe-S cluster proteins, leading to Cuproptosis." Is not supported by the presented experiments (no mitochondria or Fe-S cluster protein degradation experiments presented with the last figure about copper metabolism).

Minor

1. "Our results confirmed a significant increase in mVenus-p27K in quiescent tumor cells after 107 OSI treatment, which demonstrated that OSI treatment could rapidly induce adaptive resistance of tumor cells (Fig. 1g). " Representative images are shown but no quantitation is presented to this end. If the authors are noting something 'significant' some statistical test should be applied to a quantitative analysis. Further - it is unclear how p27 positivity suggesting a 'resting state' suggests adaptive resistance? Is it because otherwise cells are dying (as in 1k,l)? The authors should clarify what they mean here regarding evidence of adaptive resistance.

2. Fig. 2b: what are the numbers referring to on the graph and what is the x-axis scale?

3. "Analysis of OXPHOS-associated metabolites revealed that (S)-Hydroxydecanoyl-CoA,

135 Phosphate, Orthophosphate, FAD, and Citric acid were significantly increased in adapted
136 cells (Fig. 2c-e). These results suggest a significant enhancement of OXPHOS in adaptive-
137 resistant cells." It is often complicated to interpret metabolomic data as the increases in these
metabolites could indicate increases in oxphos or they could indicate blockage at some point along
in the pathway. I think a better statement would be there are metabolites in the oxphos pathway
that are elevated. The better evaluation of oxphos-level would be with the seahorse data presented
in fig. 2h.

4. The appearance of 3b is confusing with bars going opposite directions related to different
GO/KEGG/Hallmark GSEA evaluations. I believe all of these are unregulated pathways and the x-
axis is just a measure of p-value. I think would be better in that case that all the bars / x-axis are
pointed in the same direction.

5. Fig. 3j - this autophagic flux experiment is properly designed with +/- Bafilomycin conditions;
however, ideally the +/- Baf conditions would be run side by side on the same gel/blot for proper
comparison of -Baf to +Baf conditions (to control for exposure of western, etc.).

Reviewer #2 (Remarks to the Author); expert in mitochondria:

Wang et al. investigate downstream effects of long-term OSI treatment in a small number of
NSCLC cell lines, in order to identify ferritinophagy and OXPHOS as processes affected in resistant
cells. The authors make some interesting observations that implicate NCOA4 and cuproptosis as
selectively important in OSI-resistant cells, and it seems clear that NCOA4 knockout and
cuproptosis-inhibiting drugs impair tumor growth, and have synergistic effects with OSI treatment.
While these observations are interesting, the underlying mechanisms are not evaluated in this
manuscript. The importance of these results is also not clear: the most fundamental issue is that it
is not well-established that the authors are truly assessing adaptive resistant cells (described more
below), and if the resistance in these experimental/in vitro settings are related to resistance that
occurs in vivo and/or in human patients treated with OSI. In addition, most of the data are
collected from in vitro treatments, which dampens the enthusiasm for these results. Lastly, there
are some major concerns related to data presentation and statistical analysis that require
correction.

1) It is not quite clear that the protocol being used here (Fig. 1a) actually leads to adaptive
resistance. The authors claim that "short-term treatment ... within one week ... significantly
inhibited tumor growth" (lines 90-92); however there is no statistical analysis in Fig. 1a-d that
indicates the drug significantly inhibits growth. The Ki-67 analysis in Fig 1e,f would need to be
performed at early timepoints to indicate that the drug is having an effect that is overcome at later
timepoints. The authors do provide some in vitro data in Fig. 1e-l suggesting that some
parameters are changing at 24 vs. 72hr - however, these data aren't definitive of drug resistance
(see comment 4 below), and don't necessarily implicate that resistance is occurring in the xenograft
protocol (Fig. 1a-d). Thus, while the later results related to cuproptosis and NCOA4 are interesting,
their importance to drug resistance is unclear.

2) The link to cuproptosis is potentially interesting, but it is not mechanistically clear how the
selective cell death of "adapted" cells is achieved. While adapted cells might have more OXPHOS,
this doesn't necessarily imply they are reliant on OXPHOS.

3) There are some statistical issues with the majority of the figures. The figure legends largely
indicate that two-tailed t-tests are exclusively used, while several of the figures investigate
parameters collected from two cell lines, two treatments, and multiple timepoints. These
experiments would need to be assessed by multi-way ANOVA or another appropriate test.

4) With respect to the in vitro resistance examined in Fig. 1g-l: Fig 1g is not quantified, and the
images provided are largely interpretable. I am not sure exactly what the reader is supposed to
observe or conclude; to my eye, the proliferative cells do not change in vens intensity. I would
suggest removing this experiment from the manuscript. In fig 1K, the axes are unlabeled, and too
small to be interpretable. In fig. 1L, I don't see sufficient evidence of resistance at 72 hours ...
there appears to be significant cleaved caspase at this timepoint.

5) The metabolomics analysis (Fig. 2a-c) would be better performed by comparing a vehicle vs.

OSI-treated group at each timepoint. As presented, it is not possible to distinguish the effect of the drug from the effect of time in culture. There is a similar issue related to the RNAseq analysis (Fig. 3a).

6) lines 136-7: The authors attribute increases in "oxphos-related metabolites" to an increase in OXPHOS activity. However, it is not possible to attribute changes in activity with changes in metabolite levels – these metabolites may accumulate due to oxphos inhibition. Please revise.

7) Fig 2f: the authors group quantitation of several ETC-related genes together without a clear justification of this method. The quantitation should instead be separated to see if any of these genes are increasing in OSI-treated conditions. The statistics used here (2-tailed t-test) are not appropriate for similar reasons as described above.

8) Fig. 2g: only a single image of a single cell is provided, with no quantitation. This experiment should be repeated for several mitochondria from several individual cells, with quantitation of parameters.

9) Fig. 2h-k; the y-axes are not appropriately labeled (pmol/min) which do not reflect normalization to total protein as stated in the figure legend. There is no description of how basal or maximal OCR is calculated. It is not clear if the ATP production rate is experimentally determined, or calculated from the OCR data. If the ATP production rate is calculated from the OCR data alone, I would suggest removing this panel, as it is not an experimental value, and does not add significant information relative to the OCR data.

It would be important to know how the ECAR data look in these experiments, to see if glycolytic flux is adjusted in response to OSI treatment.

10) Fig. 2m,n – the data should be quantitated and statistically analyzed.

11) With respect to Fig. 3h-k: My understanding of autophagic flux determination is that it requires a co-treatment of bafilomycin to understand if changes in the LC3 ratio or GFP/mCherry ratio are due to increased or decreased autophagic flux. It is not clear how the steady state measurements provided allow a determination of autophagic flux.

13) The drugs used in Fig. 3l-n are not specific to ferritinophagy, but inhibit general autophagy and probably have other off-target effects. Based on these data, it is not possible to comment on the importance of ferritinophagy (lines 199-202).

14) Fig. 4 – although two knockout lines are generated in each cell line, only one is analyzed. It would be important to analyze the other one, as well as provide a rescue experiment, to determine if the observed effects are specific to NCOA4 deletion.

15) In a number of figures, the authors make claims about drug toxicity solely based on the weight of the animals. A more thorough analysis (histology, serum markers) would be appropriate in this setting, or the manuscript could be revised to avoid eliminate this claim.

16) Supplementary tables 3 and 4 are not clearly labeled; in particular, it is not clear what the values are for each metabolite/gene?

Reviewer #3 (Remarks to the Author); expert in copper metabolism and mitochondria:

The manuscript by Wang et al describes the potential mechanism of cancer resistance linked OXPHOS via increased iron availability. The induced adaptive resistance is achieved via reduced apoptosis and increased proliferation that is correlated to increased OXPHOS activity and improved mitochondrial morphology. The adapted cells have increased NCOA4 which is required for the adaptive response as mutation prevents the increase in OXPHOS. This implies that the release of iron is required to facilitate the increased synthesis of iron sulfur clusters. They further speculate that elesclomol would be an effective block in this synthesis or lead to enhanced degradation of iron sulfur thereby preventing the transition to adapted state. Overall, the experiments are well executed, and necessary controls are present. The hypothesis could be further supported by additional experimental evidence that would strengthen/support the claims.

Points to address:

1) The metabolites listed as evidence of increased OXPHOS are weak. However, the western blots and activities confirm an increase that strengthens the claims but additional measures such as mtDNA/nuclear DNA ratio or citrate synthase would help bolster the claims of total increase in mitochondrial volume.

2) It is not explained why citrate levels would increase if the global mechanism in increased

availability of iron sulfur, as that would presumably lead to increased aconitase activity and therefore citrate levels would be decreased. Was aconitase activity tested? Further explanation is needed then explain whether selected targets are enhanced or all targets?

3) Does additional iron supplementation affect the adaptation to OSI? Are there any changes in mitoferrin levels?

4) The claim that elesclomol works on inhibiting the iron sulfur cluster formation needs additional evidence. Cuproptosis is characterized by changes in stability of lipoylated proteins mediated by ferredoxin levels. What is the status of ferredoxin in these cells? What is the status of lipoylated protein in tumor and adapted cells? These experiments would add to the mechanisms of action.

Reviewer #4 (Remarks to the Author); expert in lung cancer and resistance:

Osimertinib is a third-generation tyrosine kinase inhibitor (TKI) that is standard of care for patients with EGFR mutant non-small cell lung cancer (NSCLC). Resistance to this TKI inevitably develops in the clinic and therefore the overall scope of the study is significant. The manuscript itself first characterizes steady state metabolite differences in cells that have "adapted" to short term osimertinib treatment. Specifically, the authors found that cells that have "adapted" to osimertinib showed elevated levels OXPHOS related metabolites. This is attributed to increased nuclear receptor coactivator 4 (NCOA4) expression in treated cancer cells. NCOA4 is a known regulator of iron metabolism and ferritinophagy (as it regulates the synthesis of proteins of the electron transport chain (ETC) and OXPHOS) and the pharmacological induction of cuproptosis could sensitize "adapted" cancer cells to osimertinib. Thus, the main conclusion of this study is that targeting ferritinophagy in EGFR mutant may increase the depth of osimertinib response and potentially delay resistance.

The quality of data and figures are in general appropriate. The authors provided both in vitro and in vivo experiments using genetic and pharmacological agents in their functional experiments across two major cell line models. The manuscript is also, for the most part, clearly written. Nevertheless, several weaknesses diminish the impact and novelty of the study.

First, conceptually, the authors seem to confound concepts of drug persistence and acquired resistance, as well as genetic vs non-genetic mechanism that arise as a consequence of TKI treatment (presumably the authors are trying to distinguish "adaptive resistance following TKI treatment vs "intrinsic resistance" or resistance due to secondary mutations?). Also, the biology of drug persisters or tolerant cells (slow cycling) are an established concept which should be referred to with greater clarity so as to enable readers to better interpret the manuscript's findings in relation to the field (see additional comments below).

Second, while the potential of connecting ferritinophagy to TKI persistence or resistance is appreciated, there has already been several published studies examining metabolic mechanisms (including but not limited to autophagy) during persistence and acquired resistance to EGFR inhibitors (e.g. PMID: 27861144). Certain experiments done in this study only demonstrate the induction of overall autophagy in osimertinib-treated cells without fundamentally ruling out lipid peroxidation (which is a hallmark of ferroptosis and typically coupled to ferritinophagy). Moreover, even though the study shows the upregulation of OXPHOS in treated cells, the authors do not provide an explanation for how ferritinophagy-promoted OXPHOS contributes to the survival advantage of osimertinib-adapted cell populations.

Third, given the multitude of pre-clinical options already suggested by the literature (to target osimertinib resistance) and many translational studies on EGFR mutant NSCLC, it is difficult to evaluate the significance of the findings presented herein without some evaluation of NCOA4 across human samples and/or clinically relevant models (e.g., PDXs).

Additional comments and suggestions:

- Consider using a different abbreviation for osimertinib (e.g., Fig 1, 2) since OSI was/is a company that has generated earlier generation TKIs (which might confuse readers).
- Most in vitro experiments define 72 hours of treatment as the time point of adaptive resistance, which needs to be reconsidered. 72 hours of treatment serves more as an acute response rather than induction of "adaptive resistance". It is also worth extending the osimertinib treatment time

to see if metabolic rewiring is maintained in cells that survive long-term osimertinib stress, which is likely to be contributing to adaptive resistance (also see comment related to figure 1 below).

- Figure 1g-i: The expression of p27 at 72hrs of treatment is an indicator of cell cycle arrest which is a signature of slow cycling persister cells. It is confusing that elevated p27 expression is accompanied by higher cell proliferation as shown in 1i.
- Figure 2m: Is there a reason why the parental and adapted cells were seeded at different densities?
- Figure 3e: LDHA, a major regulator of glycolysis, is also upregulated in adapted cells. The conclusion that the adapted cells switch to OXPHOS may not be accurate since it is possible for both OXPHOS and glycolysis to be concurrently upregulated in this context (as has been demonstrated in other tumor models as well).
- Figure 3l: It cannot be concluded that inhibiting ferritinophagy is the major cause of apoptosis in adapted cells since HCQ and 3-MA are autophagy inhibitors that do not specifically inhibit ferritinophagy. It is known that autophagy plays a protective role in cancer cells to decrease targeted therapy toxicity.
- Figure 4a: Key experiments should be validated using the second guide RNAs/knockout model (KO#2) given the comparable level of NCOA4 reduction (most if not all of the genetic experiments are performed with only one sgRNA line).
- Figure 4h: What is the baseline Mito-free iron ion level before osimertinib treatment?
- Figure 7j-m: It's not well reasoned why NCOA3 KO tumors are treated with OSI+ES, given that, in the proposed pathway, NCOA4 is upstream of cuproptosis. Moreover, although the authors show that combining copper inonphore+CuCl₂ with osimertinib inhibits the outgrowth of "resistant/persistent cells, it is not shown whether inonphore+CuCl₂ alone can induce cell apoptosis and if cell death is caused by the degradation of Fe-S protein and cuproptosis specifically. The design and time frame of the experiment are relatively short term and do not model regression followed by acquired resistance, which is more physiologically relevant. Finally, confirmation of tumor cell proliferation vs death (or cuproptosis) should be evaluated in the treated tumors to confirm the central hypothesis in vivo.

RESPONSE TO REVIEWERS' COMMENTS

We sincerely thank the four reviewers for their constructive and detailed comments on our manuscript and for their recognition of our work. The reviewers raised several major concerns and a series of specific comments that were very helpful in improving our work. We revised the manuscript extensively to address the reviewer's concerns, and the following point-by-point responses to the reviewer's specific comments were made.

The reviewers have collectively emphasized several key points: 1) the reliability of the adaptive resistance model, 2) a more accurate understanding of certain metabolites, metabolic enzymes, and changes in OXPHOS and glycolysis, 3) more specific phenotypes related to ferritinophagy, 4) replicates of another NCOA4 knockout cell line, 5) clarifying whether the sensitivity of adaptive cells to copper ionophores is based on the mechanism of cuproptosis, 6) the necessary validation in patients and *in vivo*. These are excellent points. As you will see below, we did extensive re-analyses based on your suggestions.

In our revised manuscript, we have made the following noteworthy changes to address these points:

1. We have incorporated a time-serial *in vivo* Ki-67 analysis model, further reinforcing our proposed assertion of adaptive resistance.
2. We have provided a more accurate interpretation of the metabolite profiling, along with additional analysis of both oxidative respiration and glycolysis.
3. We have included more specific phenotypes related to ferritinophagy, such as analysis of ferritin and lysosome localization.
4. All relevant experiments have been conducted using both the NCOA4-wildtype (sg-ctrl) and two NCOA4-knockout (sg-#1 and sg-#2) cell lines.
5. Further analysis of the downstream of cuproptosis to confirm our proposed cuproptosis mechanism.
6. We have validated the sensitivity of patients to Osimertinib (Osi) in patients with different NCOA4 expression levels.

In response to the major concerns and other detailed comments, we have performed additional analyses and substantially revised the manuscript. These supplementary analyzes and corresponding results have been described and provided in detail below. Overall, the reviewer's comments led to a much-improved manuscript, thank you for that again.

Please find below our point-by-point responses to each of the reviewer's comments.

Reviewer comments are shown in **black**; our responses are shown in blue.

Reviewer #1 (Remarks to the Author); expert in iron metabolism, ferritinophagy:

Wang and co-authors present a manuscript evaluating mechanisms of resistance to EGFR-TKI in non-small lung cancer and through a series of metabolomic, cell biologic, and in vivo analyses propose that NCOA4-mediated ferritinophagy is a targetable mechanism of resistance to EGFR-TKI inhibition. The findings are of significant interest; however, there are several points that require clarification:

Major

1. The observation that NCOA4 KO decreases global autophagic flux in response to OSI treatment is unexpected (Fig. 4e). NCOA4 is an autophagic adaptor for ferritin that is downstream of autophagy activation machinery. How do the authors propose that KO of NCOA4 impacts global autophagy activation as previously demonstrated in their figures (LC3 western, LC3 cherry/gfp flux experiments)?

--Response: Thank you for your insightful comment. Our research demonstrates that NCOA4-KO inhibits ferritinophagy, leading to a significant reduction in ferritinophagy in Osi-responsive cells (Fig. 4b, c, f-h). Additionally, it causes a decrease in certain autophagy-related markers associated with global autophagy (Fig. 4b-e), which aligns with the findings of previous studies¹⁻³.

Ferritinophagy, a form of selective autophagy, relies on the association between NCOA4 and both the heavy and light chains of ferritin, as well as with LC3 proteins¹. The selective interaction between LC3 and NCOA4-ferritin is crucial for the extension and closure of the engulfing membrane during autophagosome maturation. NCOA4 has also been shown to interact with TAX1BP1, facilitating the delivery of ferritin to lysosomes^{4,5}. It is hypothesized that the loss of NCOA4 impairs the progression of ferritinophagy, resulting in decreased binding between downstream autophagosomes and lysosomes observed in Fig. 4e (now updated as Fig. 4d).

In addition, we emphasize that the knockdown of NCOA4 in adapted cells under Osi-induced stress resulted in a significant increase in apoptosis (Fig. 4e, Fig. 6c-e), thereby inhibiting autophagy^{4,6}. The absence of NCOA4 inhibits upstream ferritinophagy signaling, LC3 lipidation, and specific recruitment (Fig. 4b, e).

Fig. 4 Knockout of NCOA4 significantly reduces ferritinophagy under Osi stress.

b, c IB analysis of cell lysates was performed to determine the levels of NCOA4, FTH1 and LC3B in NCOA4 sg-ctrl or sg-#1 H1975 and HCC827 cells treated with Osi for 0, 24 or 72 hours ($n = 3$). **d** Representative images of NCOA4 sg-ctrl, sg-#1 or sg-#2 H1975 and HCC827 cells stably transfected with mCherry-EGFP-LC3B under the treatments of Osi for 72 hours ($n = 12$). Scale bar, 20 μm . **e** NCOA4 sg-ctrl, sg-#1 or sg-#2 H1975 and HCC827 cells were harvested after 72 hours of Osi treatment and analyzed by electron microscopy, the red arrow marks the autophagy vesicles ($n = 12$). The red and blue dashed lines show broken cell membranes and crumpled nucleus of apoptotic cells. Scale bar, 2 μm . **f** IF using anti-Ferritin antibody (Cyan, false color) was performed in NCOA4 sg-ctrl, sg-#1 or sg-#2 H1975 and HCC827 cells stably transfected with mCherry-EGFP-LC3B and treated as **e** ($n = 12$). Scale bar, 10 μm . **g** Immunofluorescence using anti-Ferritin antibody (Cyan, false color) and anti-LAMP1 antibody (Red) was performed in cells treated as **f** ($n = 12$). Scale bar, 10 μm . **h** Flow cytometry analysis of NCOA4 sg-ctrl, sg-#1 or sg-#2 H1975 and HCC827 cells treated with Osi for 0 hours or 72 hours and stained with Mito-FerroGreen ($n = 3$). Data are shown as mean \pm SD.

2. Figure 4i and j: the baseline levels of ETC proteins appear to decrease in WT vs. KO cells (no OSI treatment). Are the quantifications in 4i based on relative amount at 0h separately in the WT and KO conditions? This might be misleading as the baseline level of the proteins (without OSI) appears to be different.

--Response to Figure 4i and j (Now updated to Fig. 4i-n): Thank you for your suggestion. We conducted a series of experiments to assess the protein expression levels in NCOA4-WT cells and two NCOA4-KO cell lines. We measured the protein expression levels under two conditions: baseline (without Osi treatment) and after 72 hours of treatment with either DMSO or Osi. Our results indicate that the knockdown of NCOA4 did not have a significant impact on the expression levels of NDUFB8, UQCRC2, MT-CO2, and ATP5A at baseline (0 hours) or after 72 hours Osi treatment, which are subunits of complexes I, III, IV, and V, respectively. However, we did observe a decrease in the expression level of SDHB, a subunit of complex II, at baseline (Fig. 4i-n).

Fig. 4 Knockout of NCOA4 significantly reduces ferritinophagy under Osi stress.

i-n Immunoblotting of five electron transport chain (ETC) proteins in NCOA4 sg-ctrl, sg-#1 or sg-#2 H1975 and HCC827 cells treated with DMSO or Osi for 0 hours, 24 hours or 72 hours (n = 3). ACTB is internal control. Data are shown as mean ± SD and were analyzed by a two-way ANOVA followed by a Dunnett-t test. NS = no significance, ***p < 0.001.

3. Fig. 5a: IREB2 does not contain an ISC. Rather its levels are controlled in response to the free iron levels in the cell. In that regard the results here are inconsistent with prior knowledge about IREB2 levels in response to NCOA4 KO. Specifically loss of NCOA4 decreases cellular free iron thereby leading to FBXL5 degradation and stabilization of IREB2 (which is a target of FBXL5). The authors should take another look at the IREB2 results and include a positive control such as DFO to ensure they are seeing the expected changes in IREB2. Regardless - IREB2 is not an ISC containing protein and the changes they are seeing here are inconsistent with prior published data.

--Response: Thank you for your thorough review of our manuscript and for this comment. We sincerely appreciate your attention to detail. We sincerely apologize for the mistake in our data collation and any inconvenience caused to the reviewers. As pointed out by the reviewer, it is important to clarify that IREB2 does not contain ISC and its levels are actually negatively regulated by free irons. We presented the accurate findings in the revised manuscript, confirming that baseline IREB2 levels were increased and remained elevated 72 hours after Osi treatment following NCOA4 knockout (Supplementary Fig. 6c). Thank you for bringing this to our attention. We have made the necessary corrections to this statement in the manuscript.

Supplementary Fig. 6 Knockout of NCOA4 inhibits the ferritinophagy of Osi adapted cells.

c The expression levels of IREB2 were assessed using immunoblotting analysis in NCOA4 sg-ctrl, sg-#1, sg-#2 H1975, and HCC827 cells after a 72-hour treatment with DMSO/Osi. The addition of the iron chelator Deferasirox (DFO) served as a positive control. ACTB is internal control. Data are shown as mean \pm SD and were analyzed by a two-way ANOVA. Source data are provided.

4. The premise of Figure 7 is that elesclomol will induce degradation of Fe-S proteins and that OSI-adaptive/resistant cells will be especially sensitive to elesclomol because of the importance of Fe-S proteins in supporting oxphos. Figure 7 shows synergy between OSI and elesclomol but there are no experiments to demonstrate the on-target efficacy of elesclomol with OSI to induce degradation of Fe-S proteins. Can the authors more specifically show that elesclomol is working how they propose in their model? It would seem that the last experiment where OSI is combined with NCOA4 KO and elesclomol would not necessarily demonstrate increased response compared to OSI + NCOA4 KO as by their proposed mechanism, NCOA4 should be upstream of elesclomol activity. Can the authors show that NCOA4 KO decreases Fe-S proteins and that elesclomol on top of NCOA4 KO further decreases Fe-S proteins as a justification of why there is additive effect of elesclomol on top of NCOA4 KO?

As it relates to the discussion, this statement "We identified that adaptive-resistant cells were sensitive to copper ions, which cause the mitochondria's degradation of Fe-S cluster proteins, leading to Cuproptosis." Is not supported by the presented experiments (no mitochondria or Fe-S cluster protein degradation experiments presented with the last figure about copper metabolism).

--Response: Thank you very much for this insightful point that was also mentioned by the other three reviewers. We greatly appreciate your comment. We have included additional experiments to further validate our claims. Specifically, we analyzed the expression levels of Fe-S proteins and downstream molecules related to cuproptosis, such as mitochondrial

lipoacylation proteins, in cells treated with Osi, elesclomol, and a combination of Osi and elesclomol *in vitro* and *in vivo*.

Our results suggest that the expression levels of ISC proteins, as well as mitochondrial acyltransferase proteins, measured using a lipoic acid-specific antibody, were significantly reduced in adapted cells following co-treatment with copper ionophores (Fig. 7g, h). Furthermore, the combined treatment of Osi and copper ionophores effectively suppressed the activity of OXPPOS in adaptive-resistant cells (Fig. 7i-k).

Additionally, in response to the comment on the last experiment (now updated as Fig. 8h-n): We conducted an analysis to investigate the differences between monotherapy with Osi, combination therapy of Osi with NCOA4 knockout, and combined treatment of Osi after NCOA4 knockout with supplementation of elesclomol (ES) to validate the core mechanism of NCOA4-mediated ferritinophagy. Importantly, the survival results of mice further demonstrated the superior efficacy of Osi when combined with ES. IHC analysis showed a significant decrease in intracellular ISC levels after continuous treatment with Osi. However, additional treatment with ES did not see the further decrease of intracellular ISC levels in NCOA4-knockout tumor cells (Fig. 8n). The results indicated that the effectiveness of ES is dependent on the core mechanism of NCOA4- ferritinophagy-ISC.

Fig.7 Active Fe-S protein synthesis in Osi adaptive-resistant tumor cells significantly reduced the threshold of Cuproptosis.

g, h Immunoblotting of iron-sulfur cluster (ISC) proteins SDHB, UQCRCFS1 and mitochondrial acylated proteins Lip-DLAT, Lip-DLST in H1975 and HCC827 cells treated with DMSO, Osi, elesclomol (ES), or Osi combined with ES for 0 hours (Baseline) or 72 hours (n = 3). ACTB is internal control. **i** Oxygen consumption rate (OCR) of H1975 cells (left) and HCC827 cells (right) after 72 hours treatment of DMSO, Osi, ES, or Osi combined with ES. Oligomycin A (Oligo-A), carbonyl cyanide-4-(trifluoromethoxy) phenylhydrazine (FCCP), antimycin A (Anti A), and rotenone (Rot) were added to measure basal OCR, ATP content, maximal OCR, and non-mitochondrial OCR (n = 3). **j, k** Basal OCR and maximal OCR in **i** normalized to total protein levels (n = 9). Data are shown as mean \pm SD and were analyzed by a two-way ANOVA. NS = no significance, *p < 0.05, **p < 0.01, ***p < 0.001.

Fig.8 Combination therapy with copper ionophore increases the efficacy of Osi.

n NCOA4 sg-ctrl or NCOA4 sg-#1 H1975 cells were orthotopically implanted into the lungs of BALB/c nude mice. The mice were then treated with either Vehicle (DMSO + saline), Osi (nasal administration, once daily), or Osi + ES (n=5). H&E staining, SDHB, and TUNEL images of the mouse tumors were obtained by staining consecutive sections of mouse tumor tissues. Scale bar, 50 μ m.

Minor

1. “Our results confirmed a significant increase in mVenus-p27K in quiescent tumor cells after

107 OSI treatment, which demonstrated that OSI treatment could rapidly induce adaptive
108 resistance of tumor cells (Fig. 1g). “ Representative images are shown but no quantitation is presented to this end. If the authors are noting something ‘significant’ some statistical test should be applied to a quantitative analysis. Further - it is unclear how p27 positivity suggesting a ‘resting state’ suggests adaptive resistance? Is it because otherwise cells are dying (as in 1k,l)? The authors should clarify what they mean here regarding evidence of adaptive resistance.

--Response: Thank you very much for this insightful point. We apologize for omitting important details in our previous description. The p27 is a cell cycle marker that is highly expressed in cells in the resting state (cells in a state of cell cycle arrest, but not necessarily undergoing apoptosis)⁷. Our findings revealed that a brief exposure to Osi for three days resulted in notable tumor cell cycle arrest, as indicated by the elevated activity of p27K. However, following an extended duration of eight days, the treatment induced adaptive resistance in tumor cells, enabling them to resume their proliferative capacity (Fig. 1i). It is suggested that on the eighth day of Osi treatment, cells develop adaptive resistance. This resistance is characterized by the ability to overcome cell cycle arrest induced by Osi and regain proliferative capacity, which is also in line with recent research findings on “adaptive resistance”^{8,9}. We performed the necessary quantification of p27K fluorescence intensity concurrently, as depicted in the representative image to the right. We have also revised

and refined our claims in the manuscript.

Fig. 1 Osi treatment rapidly induces adaptive resistance in tumor cells.

i Left, Representative images of H1975 and HCC827 cells treated with Osi in 3-D culture in day 1, day 3 and day 8 (n = 3). Scale bar, 10 μ m. Right, relative fluorescence intensity of p27K and the growth of tumor cells treated with DMSO or Osi in 3-D culture (n = 3). Data are shown as mean \pm SD and were analyzed by a two-way ANOVA. ***p < 0.001, which was employed to specifically demonstrate the distinction in H1975 cells. †††p < 0.001, which was employed to specifically demonstrate the distinction in HCC827 cells.

2. Fig. 2b: what are the numbers referring to on the graph and what is the x-axis scale?

--Response: Thank you for this comment. The numbers on the graph represent the quantities of differential metabolites in the respective pathways. The percentage scale on the x-axis represents the proportion of differential metabolites in this signaling pathway to all differential metabolites. We have added a legend to the axes of the graph (Fig. 2b).

3. "Analysis of OXPHOS-associated metabolites revealed that (S)-Hydroxydecanoyl-CoA, 135 Phosphate, Orthophosphate, FAD, and Citric acid were significantly increased in adapted

136 cells (Fig. 2c-e). These results suggest a significant enhancement of OXPHOS in adaptive-

137 resistant cells." It is often complicated to interpret metabolomic data as the increases in these metabolites could indicate increases in oxphos or they could indicate blockage at some point along in the pathway. I think a better statement would be there are metabolites in the oxphos pathway that are elevated. The better evaluation of oxphos level would be with the seahorse data presented in fig. 2h.

--Response: Thank you for this insightful suggestion. We have corrected this statement in the manuscript, as follows: "Moreover, there are metabolites in the OXPHOS pathway that are elevated in adapted cells (Fig. 2c, d, Supplementary Fig. 2d). These results suggest a significant alteration of OXPHOS in adaptive-resistant cells. "

4. The appearance of 3b is confusing with bars going opposite directions related to different GO/KEGG/Hallmark GSEA evaluations. I believe all of these are unregulated pathways and the x-axis is just a measure of p-value. I think would be better in that case that all the bars / x-axis are pointed in the same direction.

--Response: Thank you for this suggestion. To minimize any potential misreading, we have made the decision to standardize the direction of the x-axis (Fig. 3b).

5. Fig. 3j - this autophagic flux experiment is properly designed with +/- Bafilomycin conditions; however, ideally the +/- Baf conditions would be run side by side on the same gel/blot for proper comparison of -Baf to +Baf conditions (to control for exposure of western, etc.).

--Response: Thank you for this suggestion. We concur with the reviewer's suggestion and subsequently repeated the relevant immunoblot experiments on a single gel block (Fig. 3j).

Fig. 3 NCOA4 mediates ferritinophagy and adaptative resistance to Osi.

j Immunoblotting analysis of cell lysates was performed to determine the levels of LC3B and FTH1 in H1975 and HCC827 cells treated with Osi for 0 hours, 24 hours or 72 hours followed by two hours of treatment with DMSO or Bafilomycin A1 (Baf A1) (n = 3). Tubulin is internal control.

Reviewer #2 (Remarks to the Author); expert in mitochondria:

Wang et al. investigate downstream effects of long-term OSI treatment in a small number of NSCLC cell lines, in order to identify ferritinophagy and OXPHOS as processes affected in resistant cells. The authors make some interesting observations that implicate NCOA4 and cuproptosis as selectively important in OSI-resistant cells, and it seems clear that NCOA4 knockout and cuproptosis-inhibiting drugs impair tumor growth, and have synergistic effects with OSI treatment. While these observations are interesting, the underlying mechanisms are not evaluated in this manuscript. The importance of these results is also not clear: the most fundamental issue is that it is not well-established that the authors are truly assessing adaptive resistant cells (described more below), and if the resistance in these experimental/in vitro settings are related to resistance that occurs in vivo and/or in human patients treated with OSI. In addition, most of the data are collected from in vitro treatments, which dampens the enthusiasm for these results. Lastly, there are some major concerns related to data presentation and statistical analysis that require correction.

--Response: Thank you for your thorough analysis of our manuscript and your constructive feedback. We have incorporated your suggestions to further elucidate the underlying mechanisms of cuproptosis, specifically focusing on the impact of Osimertinib (Osi) in conjunction with copper ionophore on OXPHOS, adaptive resistance, and downstream cuproptosis processes. The detailed results pertaining to this topic will be discussed in the following Comment 2.

Furthermore, we have conducted additional experiments to validate the correlation between NCOA4 expression and the reduced objective response rate (ORR) and shortened progression-free survival (PFS) observed in lung cancer patients undergoing Osi treatment. To clinically validate our findings, we performed NCOA4 immunostaining on tissue specimens obtained from 40 EGFR-mut lung cancer patients prior to Osi treatment (Fig. 8o). An independent, blinded pathological examination revealed a statistically significant association between NCOA4 expression and a decrease in ORR (Fig. 8p). Moreover, a higher expression of NCOA4 in cancer cells from pre-Osimertinib (Osi) treatment samples correlated significantly with worse PFS during Osi treatment (Fig. 8q).

To strengthen our claims, we have also included additional *in vivo* experiments (Fig. 1f-h, Supplementary Fig. 7c-e, Supplementary Fig. 8). Additionally, we acknowledge previous statistical errors and have undertaken a comprehensive revalidation of all data to ensure the appropriate application of statistical methods. Detailed descriptions of the statistical methods employed can be found in the figure legends.

Fig. 8 **o** Schematic representation of the analysis of patient samples for NCOA4 expression in cancer cells. NCOA4 immunostaining was performed on tissue specimens (biopsies/resected material) from 40 lung cancer patients with a validated EGFR mutation that were obtained prior to first-line Osi treatment. The samples were scored by independent pathologists as either NCOA4-high (expression levels higher than the median) or NCOA4-low (expression levels lower than the median). **p** Analysis of the association between NCOA4 expression in the patient (**o**) and the response to Osi. Responders were defined as those whose best response was complete response or partial response. **q** Kaplan-Meier plot for progression-free survival (PFS) of Osi-treated patients from **o**. Data are shown as mean \pm SD and were analyzed by a chi-square test (**p**) or a log-rank test (**q**).

1) It is not quite clear that the protocol being used here (Fig. 1a) actually leads to adaptive resistance. The authors claim that “short-term treatment ... within one week ... significantly inhibited tumor growth” (lines 90-92); however there is no statistical analysis in Fig. 1a-d that indicates the drug significantly inhibits growth. The Ki-67 analysis in Fig 1e,f would need to be performed at early timepoints to indicate that the drug is having an effect that is overcome at later timepoints. The authors do provide some *in vitro* data in Fig. 1e-l suggesting that some parameters are changing at 24 vs. 72hr – however, these data aren’t definitive of drug resistance (see comment 4 below), and don’t necessarily implicate that resistance is occurring in the xenograft protocol (Fig. 1a-d). Thus, while the later results related to cuproptosis and NCOA4 are interesting, their importance to drug resistance is unclear.

--Response: We would like to express our sincere appreciation to the reviewers for their insightful evaluation and excellent suggestions on our resistance model. In our initial *in vivo* model, we analyzed the tumor growth curve and measured the doubling time of tumor growth. The results obtained from the appropriate statistical methods used for quantitative analysis revealed that short-term Osi pressure (within one week) significantly inhibited tumor growth. However, long-term Osi pressure (after one week of treatment) restored tumor growth (Fig. 1a-d). To address the limitations of our *in vivo* model in quantifying the real-time proliferation ability of tumor cells, we have incorporated a time-serial *in vivo* Ki-67 analysis model as suggested by the reviewer. Specifically, we subcutaneously inoculated H1975 or HCC827 cells into BALB/c nude mice and administered either Vehicle

or Osi treatment. A subset of mice from each group were sacrificed, and their subcutaneous tumors were subjected to IHC analysis (Fig. 1f). These results further support our observation that tumor cell proliferation is initially suppressed following treatment but undergoes rapid recovery thereafter (Fig. 1g, h, Supplementary Fig. 1a, b). Our *in vitro* resistance data also support the same conclusion (Fig. 1i-k). This finding suggests that continuous Osi treatment leads to the acquisition of “adaptive resistance” in tumor cells, but this adaptive resistance does not indicate acquired drug resistance.

In our “adaptive resistance” model, we have proposed the term 'adaptive resistance' to describe the blunting of the therapeutic response caused by adaptive changes in the signaling network following treatment with selective inhibitors of the oncoprotein-activated pathway. Adaptation occurs in hours (or days) and diminishes the initial effectiveness of the therapy (Nat Med, 2013)¹⁰. As such, adaptive resistance may facilitate acquired resistance as *de novo* resistance can arise from drug-tolerant persister cells (Nat Med, 2016)¹¹. In our adaptive resistance model, we hypothesize that adaptive resistance arises at the initiation of Osi treatment. This process is characterized by a rapid adaptation of cells to Osi, which is distinct from the conventional notion of “drug resistance”. This adaptation is largely due to epigenetic changes in cells⁹, including the metabolic alterations in this study. This is different from the genomic instability of tumor cells under long-term drug stress which leads to the development of irreversible “drug resistance”.

Fig. 1 Osi treatment rapidly induces adaptive resistance in tumor cells.

f-h BALB/c nude mouse subcutaneously inoculated with H1975 or HCC827 cells were given Vehicle or Osi treatment. A subset of mice from each group were sacrificed, and their subcutaneous tumors were excised for IHC analysis (n = 9). Data are shown as mean ± SD and were analyzed by a two-way ANOVA (g-l). ***p < 0.001, which was employed to specifically demonstrate the distinction in H1975 cells. †††p < 0.001, which was employed to specifically demonstrate the distinction in HCC827 cells.

2) The link to cuproptosis is potentially interesting, but it is not mechanistically clear how the selective cell death of “adapted” cells is achieved. While adapted cells might have more OXPHOS, this doesn’t necessarily imply they are reliant on OXPHOS.

--Response: Thank you very much for this insightful point that was also mentioned by the other three reviewers. We greatly appreciate your comment. We have included additional experiments to further validate our claims about cuproptosis. Specifically, we analyzed the

expression levels of Fe-S proteins and downstream molecules related to cuproptosis, such as mitochondrial lipoacylation proteins, in cells treated with Osi, elesclomol, and a combination of Osi and elesclomol *in vitro* and *in vivo*. Our results suggest that the expression levels of ISC proteins, as well as mitochondrial acyltransferase proteins were significantly reduced in adapted cells following co-treatment with copper ionophores (Fig. 7g, h). Furthermore, the combined treatment of Osi and copper ionophores effectively suppressed the activity of OXPHOS in adaptive-resistant cells (Fig. 7i-k). These indicate that the combination therapy of Osi with low concentrations of copper ionophores leads to cuproptosis in adapted cells. Our previous results showed that inhibition of OXPHOS induced more cell death in the adapted cells than in the parental cells (Fig. 2o, p, Supplementary Fig. 3a-d), indicating that active OXPHOS is crucial for adapted cells to survive.

In summary, we demonstrate that the active ferritinophagy-ISC synthesis-OXPHOS in adapted cells maintain a competitive advantage to tumor cells in nutrient competition under drug pressure, while also increasing sensitivity to cuproptosis.

Fig.7 Active Fe-S protein synthesis in Osi adaptive-resistant tumor cells significantly reduced the threshold of Cuproptosis.

g, h Immunoblotting of iron-sulfur cluster (ISC) proteins SDHB, UQCRCF1 and mitochondrial acylated proteins Lip-DLAT, Lip-DLST in H1975 and HCC827 cells treated with DMSO, Osi, elesclomol (ES), or Osi combined with ES for 0 hours (Baseline) or 72 hours (n = 3). ACTB is internal control. **i** Oxygen consumption rate (OCR) of H1975 cells (left) and HCC827 cells (right) after 72 hours treatment of DMSO, Osi, ES, or Osi combined with ES. Oligomycin A (Oligo-A), carbonyl cyanide-4-(trifluoromethoxy) phenylhydrazone (FCCP), antimycin A (Anti A), and rotenone (Rot) were added to measure basal OCR, ATP content, maximal OCR, and non-mitochondrial OCR (n = 3). **j, k** Basal OCR and maximal OCR in **i** normalized to total protein levels (n = 9). Data are shown as mean ± SD and were analyzed by a two-way ANOVA. NS = no significance, *p < 0.05, **p < 0.01, ***p < 0.001.

3) There are some statistical issues with the majority of the figures. The figure legends largely indicate that two-tailed t-tests are exclusively used, while several of the figures investigate parameters collected from two cell lines, two treatments, and multiple timepoints. These experiments would need to be assessed by multi-way ANOVA or another appropriate test.

--Response: Thank you very much for your suggestion. We acknowledge that there were some inappropriate statistical methods used previously, and we subsequently revalidated all the data to ensure the application of appropriate statistical methods. The use of the new statistical methods did not affect the original conclusions. The detailed descriptions of the statistical methods can be found in the figure legends.

4) With respect to the in vitro resistance examined in Fig. 1g-l: Fig 1g is not quantified, and the images provided are largely interpretable. I am not sure exactly what the reader is supposed to observe or conclude; to my eye, the proliferative cells do not change in venus intensity. I would suggest removing this experiment from the manuscript. In fig 1K, the axes are unlabeled, and too small to be interpretable. In fig. 1L, I don't see sufficient evidence of resistance at 72 hours ... there appears to be significant cleaved caspase at this timepoint.

--Response: Thank you for this comment.

In response to the comment on Fig. 1g, h (now updated as Fig. 1i and Supplementary Fig. 1c): We apologize for omitting important details in our previous description. The p27 is a marker that is highly expressed in cells in the resting state⁷. Our findings revealed that a brief exposure to Osi for three days resulted in notable tumor cell cycle arrest, as indicated by the elevated activity of p27K. However, following an extended duration of eight days, the treatment induced adaptive resistance in tumor cells, enabling them to resume their proliferative capacity (Fig. 1i). It is suggested that on the eighth day of Osi treatment, cells develop adaptive resistance. This resistance is characterized by the ability to overcome cell cycle arrest induced by Osi and regain proliferative capacity. We performed the necessary quantification of p27K fluorescence intensity concurrently, as depicted in the representative image to the right. We have also revised and refined our claims in the manuscript.

In response to the comment on Fig. 1k (now updated as Fig. 1l): The axes were properly labeled, and the numbers in each quadrant were magnified in Fig. 1l. The results demonstrate a significant decrease in the level of apoptosis in cells treated with Osi for 72 hours compared to cells treated for 24 hours, indicating the development of adaptive resistance to Osi in the 72-hour Osi-treated group.

In response to the comment on Fig. 1l (now updated as Fig. 1m): The expression levels of full-length PARP and Caspase-3 were significantly restored after 72 hours of Osi treatment, compared to 24 hours of treatment. Likewise, the levels of cleaved PARP decreased after 72 hours of Osi treatment. Although changes in the expression of cleaved Caspase-3 were not visually apparent in the figure, further quantitative analysis revealed a decrease in its

expression levels after 72 hours of Osi treatment compared to 24 hours of treatment (Supplementary Fig. 1f).

Fig. 1 Osi treatment rapidly induces adaptive resistance in tumor cells.

i Left, Representative images of H1975 and HCC827 cells treated with Osi in 3-D culture in day 1, day 3 and day 8 (n = 3). Scale bar, 10 µm. Right, relative fluorescence intensity of p27K and the growth of tumor cells treated with DMSO or Osi in 3-D culture (n = 3). Data are shown as mean ± SD and were analyzed by a two-way ANOVA. ***p < 0.001, which was employed to specifically demonstrate the distinction in H1975 cells. †††p < 0.001, which was employed to specifically demonstrate the distinction in HCC827 cells.

5) The metabolomics analysis (Fig. 2a-c) would be better performed by comparing a vehicle vs. OSI-treated group at each timepoint. As presented, it is not possible to distinguish the effect of the drug from the effect of time in culture. There is a similar issue related to the RNAseq analysis (Fig. 3a).

--Response: Thank you for this comment. We agree with the suggestions made by the reviewer. In our manuscript, we conducted sequencing and analysis of Osi at different time points, comparing the changes in cellular adaptation with the duration of Osi treatment, as well as the trends in metabolite and gene expression under Osi pressure. Additionally, we included control groups treated with the vehicle (DMSO) at different time points to ensure that the observed cellular adaptive resistance was not influenced by the treatment duration (Fig. 1).

We also followed the reviewer's recommendation and included detailed control groups at each time point, including a vehicle control in all subsequent relevant experiments. The data obtained in these experiments are consistent with the sequencing results (Fig. 2i, Fig. 3g, j). Therefore, we believe that the methods suggested by the reviewer and the ones employed in our manuscript are reasonable.

6) lines 136-7: The authors attribute increases in "oxphos-related metabolites" to an increase in OXPHOS activity. However, it is not possible to attribute changes in activity with changes in metabolite levels – these metabolites may accumulate due to oxphos inhibition. Please revise.

--Response: Thank you for this insightful suggestion, which has also been mentioned by other reviewers. We have corrected this statement in the manuscript, which now reads as follows: *"Moreover, there are metabolites in the OXPHOS pathway that are elevated in adapted cells (Fig. 2c, d, Supplementary Fig. 2d). These results suggest a significant alteration of OXPHOS in adaptive-resistant cells."*

7) Fig 2f: the authors group quantitation of several ETC-related genes together without a clear justification of this method. The quantitation should instead be separated to see if any of these genes are increasing in OSI-treated conditions. The statistics used here (2-tailed t-test) are not appropriate for similar reasons as described above.

--Response: Thank you for your comment. We quantified the expression levels of each complex subunit separately after 0 hours, 24 hours, and 72 hours, of DMSO/Osi treatment, which confirmed high-ETC protein levels in adapted cells (Fig. 2i, Supplementary Fig. 2e). Additionally, the statistical methods used have also been improved. The data were presented as mean \pm SD and were analyzed using a two-way ANOVA.

Supplementary Fig. 2 Metabolomics sequencing analyses metabolic characteristics in adaptive-resistant cells.

e Representative immunoblotting of electron transport chain (ETC) proteins in H1975 and HCC827 cells treated with DMSO or Osi for 0 hours, 24 hours, and 72 hours (n = 3). β -Actin (ACTB) is internal control. Data are shown as mean \pm SD and were analyzed by a two-way ANOVA. **p < 0.01, ***p < 0.001, ****p < 0.0001. Source data are provided.

8) Fig. 2g: only a single image of a single cell is provided, with no quantitation. This experiment should be repeated for several mitochondria from several individual cells, with quantitation of parameters.

--Response to the comment of Fig. 2g (now updated as Fig. 2e): Thank you for your comment. In the original manuscript, we examined mitochondrial morphology through electron microscopy in a randomly selected sample of 12 individual cells. Additionally, as recommended by the reviewer, we conducted a quantitative analysis of the data (Fig. 2e). Below are images of the other independent fields of view (Fig. R1).

Fig R1 H1975 and HCC827 cells were harvested after 0 hours (Parental) or 72 hours (Adapted) Osi treatment and analyzed by electron microscopy for mitochondrial morphology (n = 12). Scale bar, 1 μ m.

9) Fig. 2h-k; the y-axes are not appropriately labeled (pmol/min) which do not reflect normalization to total protein as stated in the figure legend. There is no description of how basal or maximal OCR is calculated. It is not clear if the ATP production rate is experimentally determined, or calculated from the OCR data. If the ATP production rate is calculated from the OCR data alone, I would suggest removing this panel, as it is not an experimental value, and does not add significant information relative to the OCR data.

It would be important to know how the ECAR data look in these experiments, to see if glycolytic flux is adjusted in response to OSI treatment.

--Response: Thank you for your insightful comment and suggestion. We have revised the Y-axis label as suggested by the reviewer to include the specific units of "pMoles/min/ μ g protein". We also provide a more detailed description in the Methods section. Based on the reviewers' suggestions, we have also removed the results panel for ATP production rate.

Additionally, we also assessed the levels of ECAR in both the parental and adapted cells. The results revealed that the baseline ECAR level did not exhibit significant variation between the parental and adapted cells (Fig. 2m). However, a slight decrease in the maximal ECAR level was observed in the adapted cells, suggesting a possible shift in the preferred energy metabolism pathway employed by these cells (Fig. 2n). A recent study also demonstrates that the enhancement of OXPHOS through MAPK inhibition does not negatively impact glycolysis (Cancer Res, 2023)¹². Therefore, it is possible for both OXPHOS and glycolysis to be upregulated simultaneously in the process of drug resistance. However, our adaptive resistance model showed a more pronounced increase in OXPHOS activity, rather than glycolysis. In this regard, we sincerely appreciate the reviewer's suggestion to analyze the adapted cell glycolytic capacity in more detail.

Fig. 2 Adaptive resistance cells active OXPHOS.

m Basal EACR in control conditions or in presence of 10 mM glucose (Glc) in Parental and Adapted cells (n = 3). Bar plots show means ± SEM for each OXPHOS subgroup. **n** Cell Energy Phenotype analyses of the Parental and Adapted cells through real-time quantifications of ECAR and OCR at baseline or stressed with Oligo-A/FCCP. Data are shown as mean ± SD and were analyzed by a two-way ANOVA. NS = no significance.

10) Fig. 2m,n – the data should be quantitated and statistically analyzed.

--Response: Thank you for your suggestion. We performed quantification and statistical analysis on the data presented in Fig. 2m, n (now updated as Fig. 2o, p). The results showed that inhibition of OXPHOS induced more cell death in the adapted cells than in the parental cells.

11) With respect to Fig. 3h-k: My understanding of autophagic flux determination is that it requires a co-treatment of bafilomycin to understand if changes in the LC3 ratio or GFP/mCherry ratio are due to increased or decreased autophagic flux. It is not clear how the steady state measurements provided allow a determination of autophagic flux.

--Response: Thank you for your suggestions. Based on the reviewer's suggestion, we conducted further experiments to assess the autophagic flux. Specifically, we performed a specific detection of ferritinophagy by adding bafilomycin A1. The presence of activated ferritinophagy was indicated in the adaptive-resistant tumor cells. (Supplementary Fig. 4a, Fig. 3j-l).

Supplementary Fig. 4 a Representative image of H1975 and HCC827 cells stably transfected with mCherry-EGFP-LC3B under the treatments of Osi for 0 hours (Parental) or 72 hours (Adapted) followed by two hours of treatment with DMSO or Bafilomycin A1 (n = 12). Scale bar, 20 μm.

Fig. 3 NCOA4 mediates ferritinophagy and adaptive resistance to Osi.

j Immunoblotting analysis of cell lysates was performed to determine the levels of LC3B and FTH1 in H1975 and HCC827 cells treated with Osi for 0 hours, 24 hours or 72 hours followed by two hours of treatment with DMSO or Bafilomycin A1 (Baf A1) (n = 3). Tubulin is internal control. **k** Immunofluorescence using anti-Ferritin antibody (Cyan, false color) was performed in Parental and Adapted cells stably transfected with mCherry-EGFP-LC3B followed by two hours of treatment with DMSO or Baf A1 (n = 12). Scale bar, 10 μ m. **l** Immunofluorescence using anti-Ferritin antibody (Cyan, false color) and anti-LAMP1 antibody (Red) was performed Parental and Adapted cells followed by two hours of treatment with DMSO or Baf A1 (n = 12). Scale bar, 20 μ m.

13) The drugs used in Fig. 3l-n are not specific to ferritinophagy, but inhibit general autophagy and probably have other off-target effects. Based on these data, it is not possible to comment on the importance of ferritinophagy (lines 199-202).

--Response: Thank you for your insightful comment. We acknowledge the reviewer's viewpoint and agree that the use of a specific ferritinophagy inhibitor would be more appropriate. Due to the lack of specific inhibitors for ferritinophagy, global autophagy inhibitors are still important tools in current research for studying various specific autophagy processes, including ferritinophagy (Cancer Discov. 2022)¹³ and lipid autophagy (Nat Cancer, 2023)¹⁴.

To address potential off-target effects, we employed a strategy to inhibit the expression of the adaptor protein TAX1BP1 downstream of NCOA4⁵. This intervention led to comparable outcomes, specifically through a selective increase in apoptosis in adapted cells (Supplementary Fig. 5e). Furthermore, we propose that subsequent knockout of NCOA4, which specifically inhibits ferritinophagy (Fig. 4, Fig. 5) and greatly attenuates the survival

of adapted cells (Fig. 6). These results also confirm the importance of ferritinophagy in adapted cells.

Supplementary Fig. 5 Osi adapted cells are sensitive to inhibition of ferritinophagy.

e Apoptosis levels of H1975 or HCC827 cells transfected with si-ctrl or si-TAX1BP1 for 24 hours followed by 0 hours (Parental) or 72 hours Osi (Adapted) treatment (n = 3). Data are shown as mean ± SD and were analyzed by a two-way ANOVA.

14) Fig. 4 – although two knockout lines are generated in each cell line, only one is analyzed. It would be important to analyze the other one, as well as provide a rescue experiment, to determine if the observed effects are specific to NCOA4 deletion.

--Response: Thank you for your valuable feedback, which is also raised by other reviewers. We conducted experiments using two NCOA4-knockout cell lines, and the results consistently demonstrated that downregulating NCOA4 specifically impeded ferritinophagy, synthesis of ISC proteins, OXPPOS, and cell survival in adapted cells (Fig. 4, Fig. 5, Fig. 6, Supplementary Fig. 6, and Supplementary Fig. 7). Rescue experiments have also demonstrated that overexpression of NCOA4 significantly mitigated the apoptosis induced by NCOA4 knockout in adapted cells (Fig. R2).

Fig. R2 Apoptosis levels of wildtype or NCOA4-knockout H1975 or HCC827 cells transfected with EV or NCOA4 overexpression plasmid for 24 hours followed by 72 hours Osi treatment (n = 3). Data are shown as mean ± SD and were analyzed by a one-way ANOVA.

15) In a number of figures, the authors make claims about drug toxicity solely based on the weight of the animals. A more thorough analysis (histology, serum markers) would be appropriate in this setting, or the manuscript could be revised to avoid eliminate this claim.

Response: Thank you for your valuable comments and suggestions. Relevant histological analyses, including mouse liver and kidney, were conducted following the recommendations of the reviewers. The results demonstrated that treatment with the combination of elesclomol did not result in an increase in hepatorenal toxicity. These findings further support the assertion that there was no significant drug toxicity associated with combination therapy (Fig. 8f, m).

Fig.8 Combination therapy with copper ionophore increases the efficacy of Osi.

f Representative H&E images of the liver and kidney were obtained from subcutaneous tumor-bearing mice and used as surrogate markers to assess treatment toxicity. Scale bar, 50 μ m.

m Representative H&E images of the liver and kidney were obtained from mice with lung orthotopic implantation models and used as surrogate markers to assess treatment toxicity. Scale bar, 50 μ m.

16) Supplementary tables 3 and 4 are not clearly labeled; in particular, it is not clear what the values are for each metabolite/gene?

--Response: Thank you for your comment. We have provided more detailed annotations for Supplementary Tables 3 and 4, specifically specifying the values for each metabolite/gene (now updated as Supplementary Tables 4 and 5).

Reviewer #3 (Remarks to the Author); expert in copper metabolism and mitochondria:

The manuscript by Wang et al describes the potential mechanism of cancer resistance linked OXPHOS via increased iron availability. The induced adaptive resistance is achieved via reduced apoptosis and increased proliferation that is correlated to increased OXPHOS activity and improved mitochondrial morphology. The adapted cells have increased NCOA4 which is required for the adaptive response as mutation prevents the increase in OXPHOS. This implies that the release of iron is required to facilitate the increased synthesis of iron sulfur clusters. They further speculate that elesclomol would be an effective block in this synthesis or lead to enhanced degradation of iron sulfur thereby preventing the transition to adapted state. Overall, the experiments are well executed, and necessary controls are present. The hypothesis could be further supported by additional experimental evidence that would strengthen/support the claims.

Points to address:

1) The metabolites listed as evidence of increased OXPHOS are weak. However, the western blots and activities confirm an increase that strengthens the claims but additional measures such as mtDNA/nuclear DNA ratio or citrate synthase would help bolster the claims of total increase in mitochondrial volume.

--Response: Thank you for your valuable feedback, which is also raised by other reviewers. We have revised the statement regarding the elevation of OXPHOS inferred from metabolites in the manuscript. It now reads as follows: *"Moreover, there are metabolites in the OXPHOS pathway that are elevated in adapted cells (Fig. 2c, d, Supplementary Fig. 2d). These results suggest a significant alteration of OXPHOS in adaptive-resistant cells"*.

Additionally, in accordance with the recommendations of the reviewers, we assessed the mtDNA/nuclear DNA ratio and citrate synthase (CS) activity. The results indicate that both the ratio of mitochondrial DNA to nuclear DNA and citrate synthase activity were significantly increased in the adapted cells (Fig. 2g, h). These results suggest an active involvement of OXPHOS in the adapted cells.

Fig. 2 Adaptive resistance cells active oxidative phosphorylation (OXPHOS).

g The ratio of mitochondrial DNA to nuclear DNA in Parental and Adapted cells (n = 9). **h** Relative citrate synthase activity in Parental and Adapted cells (n = 9). Data are shown as mean \pm SD and were analyzed by a two-tailed unpaired t-test. ***p < 0.001.

2) It is not explained why citrate levels would increase if the global mechanism in increased availability of iron sulfur, as that would presumably lead to increased aconitase activity and therefore citrate levels would be decreased. Was aconitase activity tested? Further explanation is needed then explain whether selected targets are enhanced or all targets?

--Response: Thank you for this insightful comment. The increase in iron-sulfur synthesis and high OXPHOS enhance the production rate of the tricarboxylic acid (TCA) cycle, resulting in high activity of citrate synthase (CS) in adapted cells (Fig. 2h). This, in turn, leads to an increased generation of citrate. On the other hand, we propose although an increase in iron-sulfur content leads to an increase in the activity of aconitase, which promotes the utilization of citrate, this process is reversible¹⁵. It is important to note that in a high-energy environment, the activity of the key rate-limiting enzyme, isocitrate dehydrogenase (IDH), in the TCA cycle may be inhibited, causing the accumulation of isocitrate, and subsequently increasing the levels of citrate. We also acknowledge that analyzing the levels of metabolites at a single time point has limitations. Therefore, in future studies, it would be advisable to select multiple closely spaced time points to reflect more accurate levels and changes in metabolites. Thank you again for your insightful comments.

3) Does additional iron supplementation affect the adaption to OSI? Are there any changes in mitoferrin levels?

--Response: Thank you for your insightful comment and suggestion. In response to the recommendations of the reviewers, we conducted an experiment to examine the impact of supplemental iron on the adaptability of tumor cells to Osi, as well as the expression of the Fe-S protein in tumor cells under Osi pressure. Tumor cells were treated with or without iron ions for 24 hours, after which we measured the levels of apoptosis and cellular Fe-S protein expression. The results indicated that the addition of iron ions resulted in an acceleration of adaptive resistance in tumor cells to Osi, accompanied by a significant increase in Fe-S expression (Fig. R3).

Fig. R3 a, b H1975 or HCC827 cells were treated with Osi for 24 hours with or without the addition of iron ions. The levels of apoptosis (**a**) (n = 3) and the expression levels of Fe-S protein (**b**) were measured. Data are shown as mean ± SD and were analyzed by a two-way ANOVA.

4) The claim the elesclomol works on inhibiting the iron sulfur cluster formation needs additional evidence. Cuproptosis is characterized by changes in stability of lipoylated

proteins mediated by ferredoxin levels. What is the status of ferredoxin in these cells? What is the status of lipoylated protein in tumor and adapted cells? These experiments would add to the mechanisms of action.

--Response: Thank you very much for this insightful point that was also mentioned by the other three reviewers. We greatly appreciate your comment. We have included additional experiments to further validate our claims. Specifically, we analyzed the expression levels of Fe-S proteins and downstream molecules related to cuproptosis, such as mitochondrial lipoylated proteins, in cells treated with Osi, elesclomol, and a combination of Osi and elesclomol *in vitro* and *in vivo*.

Our results suggest that the expression levels of ISC proteins (SDHB, UQCRCF1), as well as mitochondrial acyltransferase proteins (Lip-DLAT, Lip-DLST), measured using a lipoyc acid-specific antibody, were significantly reduced in adapted cells following co-treatment with copper ionophores (Fig. 7g, h). Furthermore, the combined treatment of Osi and copper ionophores effectively suppressed the activity of OXPHOS in adaptive-resistant cells (Fig. 7i-k). Additionally, immunohistochemistry (IHC) analysis of the tumors in mice revealed that the combined treatment of Osi and elesclomol led to a significant reduction in intracellular ISC levels in the tumor cells (Fig. 8n).

Fig.7 Active Fe-S protein synthesis in Osi adaptive-resistant tumor cells significantly reduced the threshold of Cuproptosis.

g, h Immunoblotting of iron-sulfur cluster (ISC) proteins SDHB, UQCRCF1 and mitochondrial acylated proteins Lip-DLAT, Lip-DLST in H1975 and HCC827 cells treated with DMSO, Osi, elesclomol (ES), or Osi combined with ES for 0 hours (Baseline) or 72 hours (n = 3). ACTB is internal control. **i** Oxygen consumption rate (OCR) of H1975 cells (left) and HCC827 cells (right) after 72 hours treatment of DMSO, Osi, ES, or Osi combined with ES. Oligomycin A (Oligo-A), carbonyl cyanide-4-(trifluoromethoxy) phenylhydrazone (FCCP), antimycin A (Anti A), and rotenone (Rot) were added to measure basal OCR, ATP content, maximal OCR, and non-mitochondrial OCR (n = 3). **j, k** Basal OCR and maximal OCR in **i** normalized to total protein levels (n = 9). Data are shown as mean ± SD and were analyzed by a two-way ANOVA. NS = no significance, *p < 0.05, **p < 0.01, ***p < 0.001.

Fig.8 Combination therapy with copper ionophore increases the efficacy of Osi.

n NCOA4 sg-ctrl or NCOA4 sg-#1 H1975 cells were orthotopically implanted into the lungs of BALB/c nude mice. The mice were then treated with either Vehicle (DMSO + saline), Osi (nasal administration, once daily), or Osi + ES (n=5). H&E staining, SDHB, and TUNEL images of the mouse tumors were obtained by staining consecutive sections of mouse tumor tissues. Scale bar, 50 μ m.

Reviewer #4 (Remarks to the Author); expert in lung cancer and resistance:

Osimertinib is a third-generation tyrosine kinase inhibitor (TKI) that is standard of care for patients with EGFR mutant non-small cell lung cancer (NSCLC). Resistance to this TKI inevitably develops in the clinic and therefore the overall scope of the study is significant. The manuscript itself first characterizes steady state metabolites differences in cells that have “adapted” to short term osimertinib treatment. Specifically, the authors found that cells that have “adapted” to osimertinib showed elevated levels OXPHOS related metabolites. This is attributed to increased nuclear receptor coactivator 4 (NCOA4) expression in treated cancer cells. NCOA4 is a known regulator of iron metabolism and ferritinophagy (as it regulates the synthesis of proteins of the electron transport chain (ETC) and OXPHOS) and the pharmacological induction of cuproptosis could sensitize “adapted” cancer cells to osimertinib. Thus, the main conclusion of this study is that targeting ferritinophagy in EGFR mutant may increase the depth of osimertinib response and potentially delay resistance.

The quality of data and figures are in general appropriate. The authors provided both in vitro and in vivo experiments using genetic and pharmacological agents in their functional experiments across two major cell line models. The manuscript is also, for the most part, clearly written. Nevertheless, several weaknesses diminish the impact and novelty of the study.

First, conceptually, the authors seem to confound concepts of drug persistence and acquired resistance, as well as genetic vs non-genetic mechanism that arise as a consequence of TKi treatment (presumably the authors are trying to distinguish “adaptive resistance following TKi treatment vs “intrinsic resistance” or resistance due to secondary mutations?). Also, the biology of drug persister or tolerant cells (slow cycling) are an established concepts which should be referred to with greater clarity so as to enable readers to better interpret the manuscript’s findings in relation to the field (see additional comments below).

--Response: Thank you very much for this insightful comment. We agree with the reviewer's perspective and apologize for not presenting our views more clearly in the manuscript. We propose that under sustained EGFR-TKI pressure, tumor cells initially rapidly adapt to the drug, and then either re-connect to cell cycle and continue growing or enter a state of sustained drug tolerance (DTP)^{9,16}. This may depend on the magnitude of drug pressure, including drug concentration (JCI, 2019; Nature 2017) and whether combination therapy is employed (Cancer Cell, 2020)¹⁷⁻¹⁹. Over time, persistent drug exposure may lead to the development of irreversible acquired resistance through epigenetic changes or second mutations (or both)^{20,21}.

In our “adaptive resistance” model, we have proposed the term 'adaptive resistance' to describe the blunting of the therapeutic response caused by adaptive changes in the signaling network following treatment with selective inhibitors of the oncoprotein-activated pathway. Adaptation occurs in hours and diminishes the initial effectiveness of the therapy (Nat Med, 2013)¹⁰. As such, adaptive resistance may facilitate acquired resistance as de novo resistance can arise from drug-tolerant persister cells (Nat Med, 2016)¹¹.

Additionally, in response to the reviewer's suggestion, we have provided a more detailed description of the concept of DTP cells in the Introduction section.

“Studies have shown that the adaptive resistance of tumor cells to drugs at the initial treatment^{9,22}, the drug-tolerant persister (DTP)^{17,23,24}, and the genomic instability of tumor cells under the long-term drug stress are the bridges mediating the emergence of irreversible drug resistance^{25,26}. The characteristics of DTP cells include their ability to evade cell death during initial chemotherapy or EGFR-TKIs-based combination therapy, which represents a subset of cancer cells. These cells serve as a reservoir from which drug-resistant tumors emerge^{16,17,27}. Although the adaptive resistance of tumor cells to drugs is a crucial step in the acquisition of drug resistance, little is known about how this process relates to EGFR-TKIs.”

Second, while the potential of connecting ferritinophagy to TKI persistence or resistance is appreciated, there has already several published studies examining metabolic mechanisms (including but not limited to autophagy) during persistence and acquired resistance EGFR inhibitors (e.g. PMID: 27861144). Certain experiments done in this study only demonstrate the induction of overall autophagy in osimertinib-treated cells without fundamentally ruling out lipid peroxidation (which is a hallmark of ferroptosis and typically coupled to ferritinophagy). Moreover, even though the study shows the upregulation of OXPHOS in treated cells, the authors do not provide an explanation for how ferritinophagy-promoted OXPHOS contributes to the survival advantage of osimertinib-adapted cell populations.

--Response: Thank you for your comments and suggestions. Indeed, several studies have demonstrated the relationship between autophagy and resistance to targeted therapy, not only in the EGFR-TKI resistance process in NSCLC but also in the resistance process to MAPK inhibition in various tumors (Nat Med, 2019)^{28,29}. Our study further emphasizes the role of ferritinophagy in the adaptive resistance to initial drug treatment. To highlight this specificity, we have incorporated additional experiments specifically focusing on ferritinophagy and have excluded the potential influence of ferroptosis, which is highly related to ferritinophagy, in the adaptive resistance cells. Our findings indicate that the initial treatment with Osimertinib (Osi) leads to a significant increase in ferritinophagy in the adapted cells characterized by elevated OXPHOS. Moreover, the inhibition of the key factor in ferritinophagy, NCOA4, significantly reduces the autophagic activity in the adapted cells, including a notable decrease in the expression of markers for global autophagy (Fig. 3k, l, Fig.4f, g and Supplementary Fig. 4b-g).

We appreciate the reviewer's insightful comment on how elevated OXPHOS enhances the survival advantage of tumor cells under Osi pressure. Drug-tolerant tumor cells prefer OXPHOS, which takes place in the mitochondria and produces large amounts of ATP (Cancer Res, 2023; Cell, 2020)^{12,16}. We hypothesize that OXPHOS, as a process with higher efficiency in utilizing nutrients compared to aerobic glycolysis, may provide a competitive advantage to tumor cells in nutrient competition under drug stress. This hypothesis is also supported by other studies (Nat Commun 2022; Blood, 2023; Nature,

2023)³⁰⁻³². However, it is worth further investigating the mechanism underlying the survival advantage promoted by OXPPOS under drug pressure in our subsequent research.

Fig. 3 NCOA4 mediates ferritinophagy and adaptive resistance to Osi.

k Immunofluorescence using anti-Ferritin antibody (Cyan, false color) was performed in Parental and Adapted cells stably transfected with mCherry-EGFP-LC3B followed by two hours of treatment with DMSO or Baf A1 (n = 12). Scale bar, 10 μm. **l** Immunofluorescence using anti-Ferritin antibody (Cyan, false color) and anti-LAMP1 antibody (Red) was performed Parental and Adapted cells followed by two hours of treatment with DMSO or Baf A1 (n = 12). Scale bar, 20 μm.

Fig. 4 Knockout of NCOA4 significantly reduces ferritinophagy under Osi stress.

f Immunofluorescence using anti-Ferritin antibody (Cyan, false color) was performed in NCOA4 sg-ctrl, sg-#1 or sg-#2 H1975 and HCC827 cells stably transfected with mCherry-EGFP-LC3B and treated with Osi for 72 hours (n = 12). Scale bar, 10 μm. **g** Immunofluorescence using anti-Ferritin antibody (Cyan, false color) and anti-LAMP1 antibody (Red) was performed in NCOA4 sg-ctrl, sg-#1 or sg-#2 cells treated with Osi for 72 hours (n = 12). Scale bar, 10 μm.

Supplementary Fig. 4 Activated ferritinophagy in adaptive resistance does not lead to Ferroptosis.

b The expression of GPX4 in H1975 and HCC827 cells was detected by Enzyme-Linked Immunosorbent Assay (ELISA) after 0 hours (Parental) or 72 hours (Adapted) Osi treatment (n = 3). **c** PTGS2 mRNA expression levels was measured by qRT-PCR in H1975 and HCC827 cells treated with DMSO or Osi for 0 hours (Parental) or 72 hours (Adapted) (n = 3). **d** H1975 and HCC827 cells were collected after 0 hours (parental) or 72 hours (adapted) Osi treatment and incubated with DCFH-DA for 20 minutes, and Reactive oxygen species (ROS) levels was analyzed by FCA (n = 3). **e** Representative image of H1975 and HCC827 cells treated with Osi for 0 hours (Parental) or 72 hours (Adapted) followed by 2 hours of C11-BODIPY probe staining (n = 4). Scale bar, 50 µm. **f** Lipid Peroxidation (MDA) levels of H1975 and HCC827 cells was detected after 0 hours (Parental) or 72 hours (Adapted) Osi treatment (n = 9). **g** FCA of cell apoptosis in adapted H1975 and HCC827 cells after 12 hours of Osi, Osi combined with Deferoxamine (DFO), Osi combined with Ferrostatin-1 (Fer-1), or Osi combined with Trolox (TRO) treatment (n = 3). Data are shown as mean ± SD and were analyzed by a two-tailed unpaired t-test (**b-f**) or a one-way ANOVA (**g**). NS = no significance, **p < 0.01, ***p < 0.001, ****p < 0.0001. Source data are provided.

Third, given the multitude of pre-clinical options already suggested by the literature (to target osimertinib resistance) and many translational studies on EGFR mut NSCLC, it is difficult to evaluate the significance of the findings presented herein without some evaluation of NCOA4 across human samples and/or clinically relevant models (e.g., PDXs).

--Response: Thank you for raising this point and your suggestions. We strongly agree with

the reviewer's perspective. We validated the correlation between NCOA4 expression and decreased objective response rate (ORR) and shortened progression-free survival (PFS) in lung cancer patients receiving Osi treatment. To clinically validate our experimental findings, we performed NCOA4 immunostaining on tissue specimens that were obtained prior to Osi treatment from 40 EGFR-mut lung cancer patients (Fig. 8o). An independent blinded pathological examination revealed a statistically significant association between NCOA4 expression and the decrease in ORR (Fig. 8p). Moreover, a high expression of NCOA4 in cancer cells from pre-Osi-treatment samples was significantly correlated with a worse PFS on Osi treatment (Fig. 8q).

Fig. 8 o Schematic representation of the analysis of patient samples for NCOA4 expression in cancer cells. NCOA4 immunostaining was performed on tissue specimens (biopsies/resected material) from 40 lung cancer patients with a validated EGFR mutation that were obtained prior to first-line Osi treatment. The samples were scored by independent pathologists as either NCOA4-high (expression levels higher than the median) or NCOA4-low (expression levels lower than the median). **p** Analysis of the association between NCOA4 expression in the patient (**o**) and the response to Osi. Responders were defined as those whose best response was complete response or partial response. **q** Kaplan-Meier plot for progression-free survival (PFS) of Osi-treated patients from **o**. Data are shown as mean \pm SD and were analyzed by a chi-square test (**p**) or a log-rank test (**q**).

Additional comments and suggestions:

- Consider using a different abbreviation for osimertinib (e.g., Fig 1, 2) since OSI was/is a company that has generated earlier generation TKIs (which might confuse readers).

--Response: Thank you for raising this point. We have made a modification to the abbreviation of Osimertinib, and it is now referred to as "Osi". This abbreviation is commonly used in many research studies, for example, "patients treated with Osimertinib (ESMO Open, 2023, Cancer Cell, 2023)"^{33,34}. This modification was implemented to avoid confusion with the abbreviation of the company name.

- Most *in vitro* experiments define 72 hours of treatment as the time point of adaptive resistance, which needs to be reconsidered. 72 hours of treatment serves more as an acute response rather than induction of “adaptive resistance”. It is also worth extending the osimertinib treatment time to see if metabolic rewiring is maintained in cells that survive long-term osimertinib stress, which is likely to be contributing to adaptive resistance (also see comment related to figure 1 below).

--Response: Thank you for your insightful comment. As we previously mentioned, we propose that the acute response of tumor cells after 72 hours of treatment reflects their adaptation to the drug. This adaptation leads to resistance in tumor cells, manifested by their ability to undergo renewed proliferation. In some studies, tumor cells may enter a state of sustained tolerance (slow cycle) after adaptation to the drug, depending on factors such as the drug concentration, and the use of combination therapy. In summary, we consider this rapid adaptation to the drug as a form of adaptive resistance, which aligns with recent research on adaptive resistance to KRAS inhibitors (Nat Cancer, 2023)⁸.

Following the reviewer's suggestion, we conducted a 50-day Osi treatment in an *in vivo* model and assessed the expression of proteins involved in OXPHOS in tumors. The results revealed that, after prolonged exposure to Osi treatment, adaptive cells maintained a high level of OXPHOS characteristics (Fig. 8n).

- Figure 1g-i: The expression of p27 at 72hrs of treatment is an indicator of cell cycle arrest which is a signature of slow cycling persister cells. It is confusing that elevated p27 expression is accompanied by higher cell proliferation as shown in 1i.

--Response to the comment on Fig. 1i (now updated as Fig. 1j): Thank you for this comment. We sincerely apologize for any confusion arising from our unclear description. The figure presented illustrates another experiment, demonstrating the results of the 5-ethynyl-2-deoxyuridine (EdU) experiment. The intensity of the EdU dye serves as an indicator of DNA replication, with higher activity indicating vigorous cell proliferation, which is contrary to p27K. The results of this experiment indicate that cells treated with Osi for 72 hours exhibited higher cell proliferation activity compared to cells treated with Osi for 24 hours (Fig. 1j).

- Figure 2m: Is there a reason why the parental and adapted cells were seeded at different densities?

--Response to the comment on Fig. 2m (now updated as Fig. 2o, p): Thank you for your comment. The adaptive cells underwent a 72-hour treatment with Osi, resulting in a significant reduction in cell density. Detailed experimental protocols have also been described in the Methods section of the manuscript. Additionally, quantification and statistical analysis were conducted (Fig. 2o, p). The results clearly demonstrated that inhibition of OXPHOS led to a greater induction of cell death in the adapted cells as compared to the parental cells.

Fig. 2 Adaptive resistance cells active OXPHOS.

o, p Colony formation assays of Parental and Adapted H1975 or HCC827 cells at 0 hours, treated with DMSO for 12 hours, and treated with an increasing concentration of oligo-A or rotenone for 12 hours (n = 3). Data are shown as mean ± SD and were analyzed by a two-way ANOVA.

- Figure 3e: LDHA, a major regulator of glycolysis, is also upregulated in adapted cells. The conclusion that the adapted cells switch to OXPHOS may not be accurate since it is possible for both OXPHOS and glycolysis to be concurrently upregulated in this context (as has been demonstrated in other tumor models as well).

--Response: Thank you very much for this insightful comment. We simultaneously assessed changes in glycolysis in adaptive resistant cells. The results indicated that there was no significant variation in the baseline extracellular acidification rate (ECAR) level between the parental and adaptive cells (Fig. 2m). However, a slight decrease in the maximal ECAR level was observed in the adaptive cells, implying a potential shift in the preferred energy metabolism pathway employed by these cells (Fig. 2n). A recent study also demonstrates that the enhancement of OXPHOS through MAPK inhibition does not negatively impact glycolysis (Cancer Res, 2023)¹². Therefore, it is possible for both oxidative phosphorylation (OXPHOS) and glycolysis to be upregulated simultaneously in the process of drug resistance. However, our adaptive resistance model showed a more pronounced increase in OXPHOS activity, rather than glycolysis.

Fig. 2 Adaptive resistance cells active OXPHOS.

m Basal EACR in control conditions or in presence of 10 mM glucose (Glc) in Parental and Adapted cells (n = 3). Bar plots show means ± SEM for each OXPHOS subgroup. **n** Cell Energy Phenotype analyses of the Parental and Adapted cells through real-time quantifications of ECAR and OCR at baseline or stressed with Oligo-A/FCCP. Data are shown as mean ± SD and were analyzed by a two-way ANOVA. NS = no significance.

- Figure 3I: It cannot be concluded that inhibiting ferritinophagy is the major cause of apoptosis in adapted cells since HCQ and 3-MA are autophagy inhibitors that do not specifically inhibit ferritinophagy. It is known that autophagy plays a protective role in cancer cells to decrease targeted therapy toxicity.

--Response: Thank you for your insightful comment. We acknowledge the reviewer's viewpoint and agree that the use of a specific ferritinophagy inhibitor would be more appropriate. Due to the lack of specific inhibitors for ferritinophagy, global autophagy inhibitors are still important tools in current research for studying various specific autophagy processes, including ferritinophagy (Cancer Discov. 2022)¹³ and lipid autophagy (Nat Cancer, 2023)¹⁴.

To address potential off-target effects, we employed a strategy to inhibit the expression of the adaptor protein TAX1BP1 downstream of NCOA4⁵. This intervention led to comparable outcomes, specifically through a selective increase in apoptosis in adapted cells (Supplementary Fig. 5e). Furthermore, we propose that subsequent knockout of NCOA4, which specifically inhibits ferritinophagy (Fig. 4, Fig. 5) and greatly attenuates the survival of adapted cells (Fig. 6). These results also confirm the importance of ferritinophagy in adapted cells.

Supplementary Fig. 5 Osi adapted cells are sensitive to inhibition of ferritinophagy.

e Apoptosis levels of H1975 or HCC827 cells transfected with si-ctrl or si-TAX1BP1 for 24 hours followed by 0 hours (Parental) or 72 hours Osi (Adapted) treatment (n = 3). Data are shown as mean ± SD and were analyzed by a two-way ANOVA.

- Figure 4a: Key experiments should be validated using the second guide RNAs/knockout model (KO#2) given the comparable level of NCOA4 reduction (most if not all of the genetic experiments are performed with only one sgRNA line).

--Response: Thank you for your valuable feedback, which is also raised by other reviewers. We conducted experiments using two NCOA4-knockout cell lines, and the results consistently demonstrated that downregulating NCOA4 specifically impeded ferritinophagy, OXPHOS, and cell survival in adapted cells (Fig. 4, Fig. 5, Fig. 6, Supplementary Fig. 6, and Supplementary Fig. 7).

- Figure 4h: What is the baseline Mito-free iron ion level before osimertinib treatment?

-Response: Thank you for raising this point. Based on the reviewer's suggestion, we conducted additional measurements of the levels of free iron ions in mitochondria at baseline and after 72 hours of treatment with Osi. The results demonstrated that the knockout of NCOA4 led to a reduction in mitochondrial free iron levels both at baseline and after 72 hours of treatment with Osi (Fig. 4h).

- Figure 7j-m: It's not well reasoned why NCOA3 KO tumors are treated with OSI+ES, given that, in the proposed pathway, NCOA4 is upstream of cuproptosis. Moreover, although the authors show that combining copper ionophore+CuCl₂ with osimertinib inhibits the outgrowth of "resistant/persistent cells, it is not shown whether ionophore+CuCl₂ alone can induce cell apoptosis and if cell death is caused by the degradation of Fe-S protein and cuproptosis specifically. The design and time frame of the experiment are relatively short term and do not model regression followed by acquired resistance, which is more physiologically relevant. Finally, confirmation of tumor cell proliferation vs death (or cuproptosis) should be evaluated in the treated tumors to confirm the central hypothesis *in vivo*.

-Response: Thank you very much for this insightful point that was also mentioned by the other three reviewers. We greatly appreciate your comment. In response to the comment of Fig.7j-m (now updated as Fig. 8h-n): We conducted an analysis to investigate the differences between monotherapy with Osi, combination therapy of Osi with NCOA4 knockout, and combined treatment of Osi after NCOA4 knockout with supplementation of elesclomol (ES) to validate the core mechanism of NCOA4-mediated ferritinophagy. Importantly, the survival results of mice further demonstrated the superior efficacy of Osi when combined with ES. IHC analysis showed a significant decrease in intracellular ISC levels after continuous treatment with Osi. However, additional treatment with ES did not see the further decrease of intracellular ISC levels in NCOA4-knockout tumor cells (Fig. 8n). The results indicated that the effectiveness of ES is dependent on the core mechanism of NCOA4- ferritinophagy-ISC.

In response to the reviewer's suggestion, we conducted an analysis to assess the potential of ionophore+CuCl₂ to induce cell apoptosis. Additionally, we explored the expression levels of Fe-S proteins and downstream molecules that are associated with cuproptosis in cells that were subjected to Osi, ES, and a combination of Osi and ES, both *in vitro* and *in vivo*. Based on our findings, it appears that the use of ES alone does not significantly inhibit tumor growth both *in vitro* and *in vivo* (Fig. 7a-c, Supplementary Fig. 8a, b).

The expression levels of Fe-S proteins, as well as mitochondrial acyltransferase proteins, measured using a lipoic acid-specific antibody, were significantly reduced in adapted cells following co-treatment with copper ionophores (Fig. 7g, h). Furthermore, the combined treatment of Osi and copper ionophores effectively suppressed the activity of OXPHOS in adaptive-resistant cells (Fig. 7i-k). Additionally, IHC analysis of the subcutaneous tumors

in mice revealed that the combined treatment of Osi and ES led to a significant reduction in intracellular Fe-S levels in the tumor cells (Fig. 8n).

Due to limitations with the mouse model, we assessed the impact of NCOA4 expression levels on the efficacy of Osi treatment in clinical patients, as described in the third comment.

Fig.7 Active Fe-S protein synthesis in Osi adaptive-resistant tumor cells significantly reduced the threshold of Cuproptosis.

a-c Colony formation assays of parental and adapted H1975 or HCC827 cells at 0 hours, treated with DMSO for 12 hours, and treated with an increasing concentration of elesclomol plus CuCl₂ (a), an increasing concentration of disulfiram plus CuCl₂ (b) or an increasing concentration of NSC-319726 plus CuCl₂ (c) for 12 hours (n = 3).

Supplementary Fig. 8 Elesclomol alone does not inhibit tumor growth.

a BALB/c nude mice were subcutaneous inoculated with H1975 cells and were given Vehicle (DMSO + saline) or elesclomol (ES) and follow-up (n = 5). **b** Tumor weight was measured for Vehicle-treated group and ES-treated (n = 5). Scale bar, 1 cm. Data are shown as mean ± SD and were analyzed by a two-tailed unpaired t-test. NS = no significance. Source data are provided.

Fig.8 Combination therapy with copper ionophore increases the efficacy of Osi.

n NCOA4 sg-ctrl or NCOA4 sg-#1 H1975 cells were orthotopically implanted into the lungs of BALB/c nude mice. The mice were then treated with either Vehicle (DMSO + saline), Osi (nasal administration, once daily), or Osi + ES (n=5). H&E staining, SDHB, and TUNEL images of the mouse tumors were obtained by staining consecutive sections of mouse tumor tissues. Scale bar, 50 μ m.

Fig.7 Active Fe-S protein synthesis in Osi adaptive-resistant tumor cells significantly reduced the threshold of Cuproptosis.

g, h Immunoblotting of iron-sulfur cluster (ISC) proteins SDHB, UQCRCFS1 and mitochondrial acylated proteins Lip-DLAT, Lip-DLST in H1975 and HCC827 cells treated with DMSO, Osi, elesclomol (ES), or Osi combined with ES for 0 hours (Baseline) or 72 hours (n = 3). ACTB is internal control. i Oxygen consumption rate (OCR) of H1975 cells (left) and HCC827 cells (right) after 72 hours treatment of DMSO, Osi, ES, or Osi combined with ES. Oligomycin A (Oligo-A), carbonyl cyanide-4-(trifluoromethoxy) phenylhydrazone (FCCP), antimycin A (Anti A), and rotenone (Rot) were added to measure basal OCR, ATP content, maximal OCR, and non-mitochondrial OCR (n = 3). j, k Basal OCR and maximal OCR in i normalized to total protein levels (n = 9). Data are shown as mean \pm SD and were analyzed by a two-way ANOVA. NS = no significance, *p < 0.05, **p < 0.01, ***p < 0.001.

References:

- 1 Mancias, J. D., Wang, X., Gygi, S. P., Harper, J. W. & Kimmelman, A. C. Quantitative proteomics identifies NCOA4 as the cargo receptor mediating ferritinophagy. *Nature* **509**, 105-109, doi:10.1038/nature13148 (2014).
- 2 Guo, W., Zhao, Y., Li, H. & Lei, L. NCOA4-mediated ferritinophagy promoted inflammatory responses in periodontitis. *J Periodontal Res* **56**, 523-534, doi:10.1111/jre.12852 (2021).
- 3 Lee, J., You, J. H. & Roh, J. L. Poly(rC)-binding protein 1 represses ferritinophagy-mediated ferroptosis in head and neck cancer. *Redox Biol* **51**, 102276, doi:10.1016/j.redox.2022.102276 (2022).
- 4 Mariño, G., Niso-Santano, M., Baehrecke, E. H. & Kroemer, G. Self-consumption: the interplay of autophagy and apoptosis. *Nat Rev Mol Cell Biol* **15**, 81-94, doi:10.1038/nrm3735 (2014).
- 5 Vargas, J. N. S., Hamasaki, M., Kawabata, T., Youle, R. J. & Yoshimori, T. The mechanisms and roles of selective autophagy in mammals. *Nat Rev Mol Cell Biol* **24**, 167-185, doi:10.1038/s41580-022-00542-2 (2023).
- 6 Maiuri, M. C., Zalckvar, E., Kimchi, A. & Kroemer, G. Self-eating and self-killing: crosstalk between autophagy and apoptosis. *Nat Rev Mol Cell Biol* **8**, 741-752, doi:10.1038/nrm2239 (2007).
- 7 Correia, A. L. *et al.* Hepatic stellate cells suppress NK cell-sustained breast cancer dormancy. *Nature* **594**, 566-571, doi:10.1038/s41586-021-03614-z (2021).
- 8 Adachi, Y. *et al.* Scribble mis-localization induces adaptive resistance to KRAS G12C inhibitors through feedback activation of MAPK signaling mediated by YAP-induced MRAS. *Nat Cancer* **4**, 829-843, doi:10.1038/s43018-023-00575-2 (2023).
- 9 Labrie, M., Brugge, J. S., Mills, G. B. & Zervantonakis, I. K. Therapy resistance: opportunities created by adaptive responses to targeted therapies in cancer. *Nat Rev Cancer* **22**, 323-339, doi:10.1038/s41568-022-00454-5 (2022).
- 10 Lito, P., Rosen, N. & Solit, D. B. Tumor adaptation and resistance to RAF inhibitors. *Nat Med* **19**, 1401-1409, doi:10.1038/nm.3392 (2013).
- 11 Hata, A. N. *et al.* Tumor cells can follow distinct evolutionary paths to become resistant to epidermal growth factor receptor inhibition. *Nat Med* **22**, 262-269, doi:10.1038/nm.4040 (2016).
- 12 Li, Y. *et al.* PINK1-Mediated Mitophagy Promotes Oxidative Phosphorylation and Redox Homeostasis to Induce Drug-Tolerant Persister Cancer Cells. *Cancer Res* **83**, 398-413, doi:10.1158/0008-5472.Can-22-2370 (2023).
- 13 Ravichandran, M. *et al.* Coordinated Transcriptional and Catabolic Programs Support Iron-Dependent Adaptation to RAS-MAPK Pathway Inhibition in Pancreatic Cancer. *Cancer Discov* **12**, 2198-2219, doi:10.1158/2159-8290.Cd-22-0044 (2022).
- 14 Bruedigam, C. *et al.* Imetelstat-mediated alterations in fatty acid metabolism to induce ferroptosis as a therapeutic strategy for acute myeloid leukemia. *Nat Cancer*, doi:10.1038/s43018-023-00653-5 (2023).
- 15 Anderson, N. M., Mucka, P., Kern, J. G. & Feng, H. The emerging role and targetability of the TCA cycle in cancer metabolism. *Protein Cell* **9**, 216-237, doi:10.1007/s13238-017-0451-1 (2018).

- 16 Shen, S., Vagner, S. & Robert, C. Persistent Cancer Cells: The Deadly Survivors. *Cell* **183**, 860-874, doi:10.1016/j.cell.2020.10.027 (2020).
- 17 Kurppa, K. J. *et al.* Treatment-Induced Tumor Dormancy through YAP-Mediated Transcriptional Reprogramming of the Apoptotic Pathway. *Cancer Cell* **37**, 104-122.e112, doi:10.1016/j.ccell.2019.12.006 (2020).
- 18 Cao, Y. Adipocyte and lipid metabolism in cancer drug resistance. *J Clin Invest* **129**, 3006-3017, doi:10.1172/jci127201 (2019).
- 19 Hangauer, M. J. *et al.* Drug-tolerant persister cancer cells are vulnerable to GPX4 inhibition. *Nature* **551**, 247-250, doi:10.1038/nature24297 (2017).
- 20 Cooper, A. J., Sequist, L. V. & Lin, J. J. Third-generation EGFR and ALK inhibitors: mechanisms of resistance and management. *Nat Rev Clin Oncol*, doi:10.1038/s41571-022-00639-9 (2022).
- 21 Marin-Bejar, O. *et al.* Evolutionary predictability of genetic versus nongenetic resistance to anticancer drugs in melanoma. *Cancer Cell* **39**, 1135-1149.e1138, doi:10.1016/j.ccell.2021.05.015 (2021).
- 22 Criscione, S. W. *et al.* The landscape of therapeutic vulnerabilities in EGFR inhibitor osimertinib drug tolerant persister cells. *NPJ Precis Oncol* **6**, 95, doi:10.1038/s41698-022-00337-w (2022).
- 23 Mikubo, M., Inoue, Y., Liu, G. & Tsao, M. S. Mechanism of Drug Tolerant Persister Cancer Cells: The Landscape and Clinical Implication for Therapy. *J Thorac Oncol* **16**, 1798-1809, doi:10.1016/j.jtho.2021.07.017 (2021).
- 24 Nie, M. *et al.* Targeting acetylcholine signaling modulates persistent drug tolerance in EGFR-mutant lung cancer and impedes tumor relapse. *J Clin Invest* **132**, doi:10.1172/jci160152 (2022).
- 25 Noronha, A. *et al.* AXL and Error-Prone DNA Replication Confer Drug Resistance and Offer Strategies to Treat EGFR-Mutant Lung Cancer. *Cancer Discov* **12**, 2666-2683, doi:10.1158/2159-8290.Cd-22-0111 (2022).
- 26 Thress, K. S. *et al.* Acquired EGFR C797S mutation mediates resistance to AZD9291 in non-small cell lung cancer harboring EGFR T790M. *Nat Med* **21**, 560-562, doi:10.1038/nm.3854 (2015).
- 27 Rehman, S. K. *et al.* Colorectal Cancer Cells Enter a Diapause-like DTP State to Survive Chemotherapy. *Cell* **184**, 226-242.e221, doi:10.1016/j.cell.2020.11.018 (2021).
- 28 Bryant, K. L. *et al.* Combination of ERK and autophagy inhibition as a treatment approach for pancreatic cancer. *Nat Med* **25**, 628-640, doi:10.1038/s41591-019-0368-8 (2019).
- 29 Kinsey, C. G. *et al.* Protective autophagy elicited by RAF→MEK→ERK inhibition suggests a treatment strategy for RAS-driven cancers. *Nat Med* **25**, 620-627, doi:10.1038/s41591-019-0367-9 (2019).
- 30 Baran, N. *et al.* Inhibition of mitochondrial complex I reverses NOTCH1-driven metabolic reprogramming in T-cell acute lymphoblastic leukemia. *Nat Commun* **13**, 2801, doi:10.1038/s41467-022-30396-3 (2022).
- 31 Shao, X. *et al.* The palmitoyltransferase ZDHHC21 regulates oxidative phosphorylation to induce differentiation block and stemness in AML. *Blood* **142**, 365-381, doi:10.1182/blood.2022019056 (2023).

- 32 Zhang, X. *et al.* Reprogramming tumour-associated macrophages to outcompete cancer cells. *Nature* **619**, 616-623, doi:10.1038/s41586-023-06256-5 (2023).
- 33 Tamura, K. *et al.* Comparison of clinical outcomes of osimertinib and first-generation EGFR-tyrosine kinase inhibitors (TKIs) in TKI-untreated EGFR-mutated non-small-cell lung cancer with leptomeningeal metastases. *ESMO Open* **8**, 101594, doi:10.1016/j.esmoop.2023.101594 (2023).
- 34 de Miguel, F. J. *et al.* Mammalian SWI/SNF chromatin remodeling complexes promote tyrosine kinase inhibitor resistance in EGFR-mutant lung cancer. *Cancer Cell* **41**, 1516-1534.e1519, doi:10.1016/j.ccell.2023.07.005 (2023).

REVIEWERS' COMMENTS

Reviewer #1 (Remarks to the Author):

The authors have responded adequately to the majority of the concerns.

There is one outstanding issue:

The authors propose in their rebuttal that the reason for a decrease in global autophagy with Osi + NCOA4 KO is because of apoptosis-autophagy crosstalk. This hypothesis is reasonable given the prior literature on crosstalk between apoptosis and autophagy; however, it is not explicitly tested in their experiments (e.g. inhibition of apoptosis restoring global autophagic flux). This reviewer would suggest that the authors reword the manuscript to suggest that this is the reason for a global autophagic flux block but that it is not formally tested. Further, the authors should remove any wording that suggests this is somehow specific to NCOA4 molecular function (rather what they are seeing may just be a downstream consequence of apoptosis induction).

Reviewer #2 (Remarks to the Author):

The manuscript is much improved, particularly with the addition of a second set of NCOA4-knockout lines and tumors to strengthen the confidence in the results. Since the NCOA4-knockouts sometimes inhibit tumor growth in both vehicle and OSI-treated animals, the authors should take care to explain the limitations of their findings with respect to a specific role for NCOA4 in developing Osi-resistance. This and other limitations should be addressed in the discussion.

Previous point 1:

The authors now provide a Ki57 timeserial analysis to support that Osi treatment initially blunts proliferation, but tumors recover over time, suggesting adaptive resistance and/or selection for drug-tolerant persister cells. It is not clear that the authors can distinguish between these two possibilities based on their data, and thus, I would suggest they modify their text to acknowledge this limitation.

Previous point 2: I see the authors' point that adapted cells are more sensitive to OXPHOS (Fig. 2), but this does not necessarily mean that the mechanism by which adapted cells are more sensitive to cuproptosis is because of their sensitivity to OXPHOS ... this appears to be a correlation, not a causal link based on the current data. Instead, the authors show that NCOA4 is required for generation of adapted cells, and the text should be adjusted to focus on this requirement for NCOA4.

Previous point 3:

In the statistical analysis, the authors indicate that they made an assumption that datasets were normally distributed with equal variances and therefore performed Student's t-test. I will defer to the editor as to whether this meets the journals' standards for statistical analysis.

Previous point 4: Thank you for the additional explanation regarding Figure 1i. I don't find the data in Figure 1m (western blots) to be compelling, there is certainly a significant amount of apoptosis still occurring at the 72 hr timepoint, and the Cas-3 blot is overexposed which precludes a precise quantitation. Presumably the differences are due to a decreased number of apoptotic cells, which is already shown in Figure 1L, but it could also be due to changes in apoptotic efficiency in cells, and it is not quite clear how to distinguish between these possibilities on a western blot. I would suggest removing this panel from the manuscript, and relying on figure 1L.

Previous point 5:

I am confused by the author's response, it does not appear that they have responded to this comment. In figures 2a-c, they are still comparing 0 vs. 24hr and 0 vs. 72hr OSI data, but not vehicle vs. OSI treated samples. Do these metabolite changes occur in vehicle treated cells at 24 and 72hr? I see their point that in select analyses they compare vehicle vs. treated (e.g., Figure

2i), but am still concerned as to whether these changes reflect time-dependent vs. OSI-dependent effects.

Previous point 6:

The authors have appropriately adjusted the text in response to this point.

Previous point 7:

The authors have appropriately adjusted their analysis in their response.

Previous point 8:

The authors have appropriately provide quantitation of these changes. It would be best to include Figure R1 within the supplemental information for the reader's interest.

Previous point 9: The additional quantitation and analysis is appropriate. In the revised methods surrounding the Seahorse experiment, the drug concentrations used for oligomycin, cccp, antimycin are extremely high and likely induce non-specific effects – the authors should consider repeating these experiments at more accepted doses (e.g., 0.1-2 micromolar).

Previous point 10-16:

The authors have appropriately responded to these comments; thank you!

Reviewer #3 (Remarks to the Author):

The article by Wang et al addresses the mechanism by which tumor cells become osimertinib resistance. The resistance is characterized by increased OXPHOS that is facilitated by increase iron availability via ferritinophagy. They show that the adapted cells become susceptible to copper ionophores which inhibit iron-sulfur enzymes and lead to depletion of lipoylated mitochondrial proteins. These characteristics are consistent with cuproptosis-mediated cell death. Overall, the new experiments and discussion enhance the manuscript and the authors have satisfactorily addressed my original queries.

Reviewer #4 (Remarks to the Author):

The manuscript has been significantly revised with new data and more rigor. Remaining comments and suggestions (minor) are summarized below.

First, regarding the concept of drug persistence and acquired resistance, the authors provided a more thorough explanation of drug-tolerant persister cells and adaptive-resistant cells, which aligns with their findings.

The authors provide sufficient data to support the decoupling of ferritinophagy and ferroptosis in osimertinib adaptive-resistant cells, which distinguishes this study from the previous ones. However, the mechanisms of how cells circumvent ferroptosis under high ferritinophagy induction and how high free iron ions are required for the survival of resistant cells remains questionable and should be mentioned in the text as areas in need of future studies.

The addition of clinical data (Fig 8p) on NCOA4 expression in NSCLC patients adds clinical relevance and impact to this study. However, this data suggest that a subset of tumors may express NCOA4 prior to osimertinib treatment. The authors should clarify their interpretation of the data (i.e, can NCOA4-overexpression occur independently of osimertinib treatment and could this lead to osimertinib resistance in a non-adaptive manner).

Additional comments and suggestions:

Figure 2o, p: It seems that the adapted cells were treated with OXPHOS inhibitors right after osimertinib treatment without being reseeded and osimertinib was removed after 72-hours of treatment. It should be confirmed if the cell density was the same and clarified if the adapted cells remain dependent on OXPHOS without sustained osimertinib treatment or not.

RESPONSE TO REVIEWERS' COMMENTS

Reviewer #1 (Remarks to the Author):

The authors have responded adequately to the majority of the concerns.

There is one outstanding issue:

The authors propose in their rebuttal that the reason for a decrease in global autophagy with Osi + NCOA4 KO is because of apoptosis-autophagy crosstalk. This hypothesis is reasonable given the prior literature on crosstalk between apoptosis and autophagy; however, it is not explicitly tested in their experiments (e.g. inhibition of apoptosis restoring global autophagic flux). This reviewer would suggest that the authors reword the manuscript to suggest that this is the reason for a global autophagic flux block but that it is not formally tested. Further, the authors should remove any wording that suggests this is somehow specific to NCOA4 molecular function (rather what they are seeing may just be a downstream consequence of apoptosis induction).

--Response: Thank you for your insightful comment and suggestions. We have made a more cautious statement about the findings of this section in the results section of the manuscript.

“As anticipated, the adapted cells show decreased interaction between downstream autophagosomes and lysosomes after NCOA4 knockout, which is reflected in the restoration of ferritin levels and the obstruction of autophagic flux (Fig. 4b,c, Supplementary Fig. 6a, b). Simultaneously, the NCOA4-knockout adapted cells display an apoptotic morphology, accompanied by inhibition upstream of autophagy (Fig. 4b,c,e, Supplementary Fig. 6a, b), consistent with the characteristics of crosstalk between apoptosis and autophagy”.

Reviewer #2 (Remarks to the Author):

The manuscript is much improved, particularly with the addition of a second set of NCOA4-knockout lines and tumors to strengthen the confidence in the results. Since the NCOA4-knockouts sometimes inhibit tumor growth in both vehicle and OSI-treated animals, the authors should take care to explain the limitations of their findings with respect to a specific role for NCOA4 in developing Osi-resistance. This and other limitations should be addressed in the discussion.

We express our heartfelt gratitude for the careful review and profound suggestions provided by the reviewers regarding our work. Below, we will individually address your further comments.

Previous point 1:

The authors now provide a Ki57 timeserial analysis to support that Osi treatment initially

blunts proliferation, but tumors recover over time, suggesting adaptive resistance and/or selection for drug-tolerant persister cells. It is not clear that the authors can distinguish between these two possibilities based on their data, and thus, I would suggest they modify their text to acknowledge this limitation.

--Response: Thank you for your comment and suggestion. Ki-67 time-course analysis revealed that Osi treatment resulted in the suppression of tumor cell proliferation, followed by a gradual recovery. We have more accurately described this data.

“Additional mouse models, incorporating sequential Ki-67 expression analysis, further supported the observation that tumor cell proliferation was temporarily suppressed after initial treatment and subsequently experienced a rapid recovery (Fig. 1f-h, Supplementary Fig. 1a, b). This finding supports that Osi treatment can rapidly facilitate tumor cells to transition from proliferation inhibition to re-entering the cell cycle and growing in vivo.”

Previous point 2: I see the authors' point that adapted cells are more sensitive to OXPHOS (Fig. 2), but this does not necessarily mean that the mechanism by which adapted cells are more sensitive to cuproprptosis is because of their sensitivity to OXPHOS ... this appears to be a correlation, not a causal link based on the current data. Instead, the author's show that NCOA4 is required for generation of adapted cells, and the text should be adjusted to focus on this requirement for NCOA4.

--Response: Thank you for your comment and suggestion. Our study demonstrates that adaptive-resistant cells possess an active NCOA4-mediated ferritinophagy-ISC synthesis-OXPHOS pathway. Given the molecular mechanism of cuproprptosis is inseparable from ISC, we have found that adaptive cells are more sensitive to copper ion-induced cuproprptosis. Indeed, it should not be assumed that the increased sensitivity to cuproprptosis in adaptive-resistant cells is solely due to their dependence on OXPHOS. We have carefully reviewed the manuscript and confirmed that there is no emphasis on a cause-and-effect relationship between OXPHOS activity and cuproprptosis in the article.

Previous point 3:

In the statistical analysis, the authors indicate that they made an assumption that datasets were normally distributed with equal variances and therefore performed Student's t-test. I will defer to the editor as to whether this meets the journals' standards for statistical analysis.

--Response: Thank you for your comment. We have refined our statistical methods description in accordance with the recommendations outlined in the Author Checklist.

Previous point 4: Thank you for the additional explanation regarding Figure 1i. I don't find the data in Figure 1m (western blots) to be compelling, there is certainly a significant amount of apoptosis still occurring at the 72 hr timepoint, and the Cas-3 blot is overexposed which precludes a precise quantitation. Presumable the differences are due to a decreased number of apoptotic cells, which is already shown in Figure 1L, but it could also be due to changes in apoptotic efficiency in cells, and it is not quite clear how to distinguish between these possibilities on a western blot. I would suggest removing this panel from the

manuscript, and relying on figure 1L.

--Response: Thank you for your comment. We think that data regarding the detection of apoptotic proteins play a vital role in reinforcing our model of adaptive resistance. It has come to our attention that the representative image of the Cas-3 blot presented in our prior manuscript might have seemed overly exposed. We have substituted this with images of lower exposure for clarity.

Previous point 5:

I am confused by the author's response, it does not appear that they have responded to this comment. In figures 2a-c, they are still comparing 0 vs. 24hr and 0 vs. 72hr OSI data, but not vehicle vs. OSI treated samples. Do these metabolite changes occur in vehicle treated cells at 24 and 72hr? I see their point that in select analyses they compare vehicle vs. treated (e.g., Figure 2i), but am still concerned as to whether these changes reflect time-dependent vs. OSI-dependent effects.

--Response: Thank you for this comment. In Fig. 2a-c, we agree with the reviewers that their suggestions, along with the analytical methods we adopted, are reasonable. Our analysis notably highlights the time-dependent nature of the response to Osi treatment.

Following the reviewers' recommendations, we conducted analyses in subsequent experiments on both the vehicle and the Osi-treated groups, as well as at different time points within each group.

Previous point 6:

The authors have appropriately adjusted the text in response to this point.

--Response: Thank you!

Previous point 7:

The authors have appropriately adjusted their analysis in their response.

--Response: Thank you!

Previous point 8:

The authors have appropriately provide quantitation of these changes. It would be best to include Figure R1 within the supplemental information for the reader's interest.

--Response: Thank you for your advice. We have included Fig. R1 in the Supplementary Information.

Previous point 9: The additional quantitation and analysis is appropriate. In the revised methods surrounding the Seahorse experiment, the drug concentrations used for oligomycin, cccp, antimycine are extremely high and likely induce non-specific effects – the authors should consider repeating these experiments at more accepted doses (e.g., 0.1-2 micromolar).

--Response: Thank you for your insightful comment and valuable suggestion. The

concentrations of the compounds (oligomycin, FCCP, and antimycin A) used in hippocampal experiments were partly derived from our previous experience and by referencing prior studies. We will take the reviewer's suggestion into account and employ varying concentration gradients in our subsequent experiments to replicate the study.

Previous point 10-16:

The authors have appropriately responded to these comments; thank you!

--Response: Thank you!

Reviewer #3 (Remarks to the Author):

The article by Wang et al addresses the mechanism by which tumor cells become osimertinib resistance. The resistance is characterized by increased OXPHOS that is facilitated by increase iron availability via ferritinophagy. They show that the adapted cells become susceptible to copper ionophores which inhibit iron-sulfur enzymes and lead to depletion of lipoylated mitochondrial proteins. These characteristics are consistent with cuproptosis-mediated cell death. Overall, the new experiments and discussion enhance the manuscript and the authors have satisfactorily addressed my original queries.

--Response: We would like to express our sincere gratitude for the reviewers' appreciation of our work.

Reviewer #4 (Remarks to the Author):

The manuscript has been significantly revised with new data and more rigor. Remaining comments and suggestions (minor) are summarized below.

First, regarding the concept of drug persistence and acquired resistance, the authors provided a more thorough explanation of drug-tolerant persister cells and adaptive-resistant cells, which aligns with their findings.

The authors provide sufficient data to support the decoupling of ferritinophagy and ferroptosis in osimertinib adaptive-resistant cells, which distinguishes this study from the previous ones. However, the mechanisms of how cells circumvent ferroptosis under high ferritinophagy induction and how high free iron ions are required for the survival of resistant cells remains questionable and should be mentioned in the text as areas in need of future studies.

--Response: Thank you for your suggestion. We have added this in-depth issue to the discussion section on the limitations of our paper.

“However, how to maintain an appropriate iron ion concentration in adaptive cells to ensure the balance between ISC synthesis and avoidance of ferroptosis is indeed unknown, requiring our further exploration.”

The addition of clinical data (Fig 8p) on NCOA4 expression in NSCLC patients adds clinical relevance and impact to this study. However, this data suggest that a subset of tumors may express NCOA4 prior to osimertinib treatment. The authors should clarify their interpretation of the data (i.e, can NCOA4-overexpression occur independently of osimertinib treatment and could this lead to osimertinib resistance in a non-adaptive manner).

--Response: Thank you for your insightful comment, we agree with this reviewer's perspective. We have made a more comprehensive statement on the data.

“Furthermore, a high expression of NCOA4 in cancer cells from pre-treatment samples was significantly correlated with a poorer PFS following Osi treatment, indicating that not only does NCOA4 expression induced by treatment matter, but also its high baseline expression prior to treatment can occur independently of Osi administration, potentially leading to resistance (Fig. 8q).”

Additional comments and suggestions:

Figure 2o, p: It seems that the adapted cells were treated with OXPHOS inhibitors right after osimertinib treatment without being reseeded and osimertinib was removed after 72-hours of treatment. It should be confirmed if the cell density was the same and clarified if the adapted cells remain dependent on OXPHOS without sustained osimertinib treatment or not.

--Response: Thank you for your comment. Upon Osi treatment, we promptly administered OXPHOS inhibitors to adapted cells to ensure the preservation of their adaptive characteristics at that time. Numerous studies have demonstrated that the resistance induced by the initial drug treatment, including drug-tolerant persister (DTP), is reversible. We were concerned that the adapted cells might partially regain the phenotype of their parental cells after re-seeding. We performed quantification and statistical analysis on the data presented in Fig. 2o, p. The results showed that inhibition of OXPHOS induced more cell death in the adapted cells than in the parental cells.